# Molecular layer interneurons in the cerebellum encode for valence in associative learning

Ming Ma [1,2], Gregory L. Futia [3], Fabio M. Simoes de Souza [2,4], Baris N. Ozbay [5], Isabel Llano [6], Emily A. Gibson [1,3] & Diego Restrepo [1,2 ✉]

The cerebellum plays a crucial role in sensorimotor and associative learning. However, the contribution of molecular layer interneurons (MLIs) to these processes is not well understood. We used two-photon microscopy to study the role of ensembles of cerebellar MLIs in a go-no go task where mice obtain a sugar water reward if they lick a spout in the presence of the rewarded odorant and avoid a timeout when they refrain from licking for the unrewarded odorant. In naive animals the MLI responses did not differ between the odorants. With learning, the rewarded odorant elicited a large increase in MLI calcium responses, and the identity of the odorant could be decoded from the differential response. Importantly, MLIs switched odorant responses when the valence of the stimuli was reversed. Finally, mice took a longer time to refrain from licking in the presence of the unrewarded odorant and had difficulty becoming proficient when MLIs were inhibited by chemogenetic intervention. Our findings support a role for MLIs in learning valence in the cerebellum.

[1] Neuroscience Graduate Program, University of Colorado Anschutz Medical Campus, Aurora, CO 80045, USA. [2] Department of Cell and Developmental Biology, University of Colorado Anschutz Medical Campus, Aurora, CO 80045, USA. [3] Department of Bioengineering, University of Colorado Anschutz Medical Campus, Aurora, CO 80045, USA. [4] Center for Mathematics, Computation and Cognition, Federal University of ABC, Sao Bernardo do Campo, SP, Brazil. [5] Intelligent Imaging Innovations, Denver, CO 80216, USA. [6] Saints Pères Paris Institute for Neurosciences, Université Paris Descartes, 75006 Paris, France. ✉email: diego.restrepo@cuanschutz.edu

The cerebellum plays a pivotal role in coordinating movements through sensorimotor integration. It receives massive input through mossy fiber synapses onto granule cells (GCs) to form a circuit efficient in complex pattern separation[1] (Supplementary Fig. 1). In addition, the cerebellum contributes to cognition and emotion and is associated with non-motor conditions such as autism spectrum disorders[2–5]. The Purkinje cells (PCs), the sole projection neurons of the cerebellar cortex, receive excitatory afferent input from the parallel fibers (PFs) of GCs and dense feedforward inhibitory inputs from the molecular layer interneurons (MLIs). Importantly, subtle changes in this PC excitatory–inhibitory balance generate robust, bidirectional changes in the output of PCs[6].

Plasticity in cerebellar circuit activity plays an important role in generation of adequate output. Indeed, long-term depression (LTD) mediated by dendritic increases in $Ca^{2+}$ in PCs elicited by motor error signals conveyed by climbing fibers (CFs) is a classical model of plasticity[7–10]. However, recent studies indicate that CFs also signal reward prediction[11,12] or decision-making errors[13], and the cerebellum modulates association pathways in the ventral tegmental area (VTA) contributing to reward-based learning and social behavior[14]. Furthermore, although LTD at the PF–PC synapse is often considered as the substrate for cerebellar dependent learning[7–10], such learning can occur in the absence of LTD and may therefore involve other forms of plasticity[15]. A potential substrate for plasticity is the PF-MLI synapse[16] where LTP can be induced in slices by pairing MLI depolarization with PF stimulation[17] and in vivo by conjunctive stimulation of PFs and CFs[18], believed to underlie changes in the size of cutaneous receptive fields[19,20]. In addition, high frequency stimulation of PFs alters subunit composition of AMPA receptors, rendering them calcium impermeable[21], a long-lasting change linked to behavioral modifications[22]. Furthermore, MLIs have been proposed to participate in cerebellar plasticity[20,23,24], and Rowan et al. found graded control of PC plasticity by MLI inhibition[25], suggesting that MLI inhibition is a gate for learning stimulus valence, which conveys information as to whether the stimulus is rewarded. However, whether there is a causal participation of MLIs in reward-associated learning is unknown.

Here we explored whether MLI activity plays a role in reward-associated learning in a go–no go task where mice learn to lick to obtain a water reward[26,27]. We applied two-photon microscopy[28] to record $Ca^{2+}$ changes in ensembles of MLIs and utilized chemogenetics to explore the functional role of MLI activity in learning.

## Results

In order to explore the role of MLIs in associative learning we employed in vivo two-photon microscopy to record neural activity reported by changes in fluorescence emitted by the $Ca^{2+}$ indicators GCaMP6/7. MLIs were imaged within the superficial 50 μm of the ML in head-fixed mice through a $2 \times 2$ mm glass window implanted above the cerebellar vermis (lobule VI, Fig. 1, Supplementary Fig. 2), where GCs acquire a predictive feedback signal or expectation reward[29,30]. We used the go–no go task where water-deprived mice initiated the trial by licking on the spout to elicit odorant delivery 1–1.5 s after the first lick. Mice received a water reward when they licked at least once in two lick segments during rewarded odorant delivery (1% isoamyl acetate, Iso, termed S+) (Fig. 1a, Hit trial, mouse movement shown in Supplementary Movie 1, quantified in Supplementary Fig. 3). Mice did not receive the reward if they failed to lick in one of the two lick segments (Miss trial). When the unrewarded odorant was presented (mineral oil, MO, S− trials) the animals did not receive a reward when they refrained from licking (Correct Rejection,

CR, Fig. 1a), and they received a timeout of 10 s if they licked in both segments (False Alarm, FA). A proficient mouse (percent correct ≥80%) licked at least once in each lick segment in the S+ trials and stopped licking during the segments for the S− trials, see lick trace at the bottom of Fig. 1f.

An example of an increase in MLI GCaMP6f fluorescence intensity during odorant application in a Hit trial is shown for a proficient mouse in Fig. 1b, c (Supplementary Movie 2). We extracted regions of interest (ROIs) of the components (Fig. 1d) and temporal traces of the normalized change in fluorescence ($\Delta F/F$, Fig. 1e, f) using constrained nonnegative matrix factorization analysis[31]. The average diameter of the ROIs in this field of view (FOV) was $10.5 \pm 4$ μm (mean ± SD, $n = 170$, Supplementary Fig. 4a), which falls within the range of diameters reported for MLIs[32]. When the animal was proficient, we observed that MLI $\Delta F/F$s increased when the rewarded odorant was presented (Fig. 1e (Hit) and f, vertical orange lines in 1f indicate odorant on and off times for S+). In contrast, the unrewarded odorant-elicited smaller transient increases in $\Delta F/F$ (Fig. 1e (CR) and f, vertical light blue lines and Supplementary Figs. 4b–d), with some exceptions where the increases for $\Delta F/F$ for S− were larger (arrow in Fig. 1f). Finally, the rewarded odorant elicited increases in $\Delta F/F$ in 89% of the ROIs among a total of 191 (Fig. 1e and Supplementary Fig. 4c, e).

**MLIs responses diverge as mice learn to differentiate odors**. The behavioral performance in a learning session where the animal reached proficient level after 70 trials is shown in Fig. 2a. When the animal was naive (≤65% correct) the time courses for $\Delta F/F$ overlapped between S+ and S− (Fig. 2b), and when the animal became proficient the time courses for $\Delta F/F$ diverged (Fig. 2d). For the proficient animal in S+ trials $\Delta F/F$ started increasing at trial initiation (1–1.5 s before odorant addition) and reached a peak when the animal was rewarded (Fig. 2d, orange trace), while for S− trials, the response returned to basal levels shortly after the odorant was applied (Fig. 2d, light blue trace, lower panel of Fig. 2d shows the time courses for lick rates (LRs)). Whereas $\Delta F/F$ did not differ between S+ and S− before odorant addition (Fig. 2e), it diverged after odorant addition as the animal learned (Fig. 2f) and $\Delta F/F$ per ROI increased when the animal became proficient (Fig. 2g). Generalized linear model (GLM) analysis yielded a statistically significant difference between S+ and S− and naive and proficient, $p < 0.001$, 5550 observations, 5538 d.f., one mouse, GLM F-statistic = 267, $p < 0.001$. Finally, the mean $\Delta F/F$ (±95% CI, four mice) for the last second of odorant application increased as a function of percent correct performance for the S+ condition, whereas it remained stable for the S− condition (Fig. 2i). A GLM analysis found a statistically significant difference between S+ and S− ($p < 0.01$, 47 observations, 44 d.f., $n = 4$ sessions, 4 mice, GLM F-statistic = 60, $p < 0.001$) and for the interaction between performance and the odorant (S+ vs. S−) ($p < 0.001$, 47 observations, 44 d.f., $n = 4$ sessions, 4 mice, Supplementary Fig. 5a–d shows learning curves for these mice). Therefore, as the animal learned, the magnitude and temporal dynamics of the change in $Ca^{2+}$ in MLIs diverged between S+ and S− odorants, suggesting a possible role for MLIs in associative learning. The majority of the ROIs responded similarly (Fig. 1e, Supplementary Fig. 4) and a dimensionality analysis indicated that the dimensionality ranged from 2 to 6 (see Supplementary Note 1 and Supplementary Fig. 6) indicating that the responses are highly redundant.

**Stimulus identity can be decoded from MLI activity**. The finding that the MLIs developed divergent responses for S+ vs. S− trials (Fig. 2i) raised the question whether ensemble neural

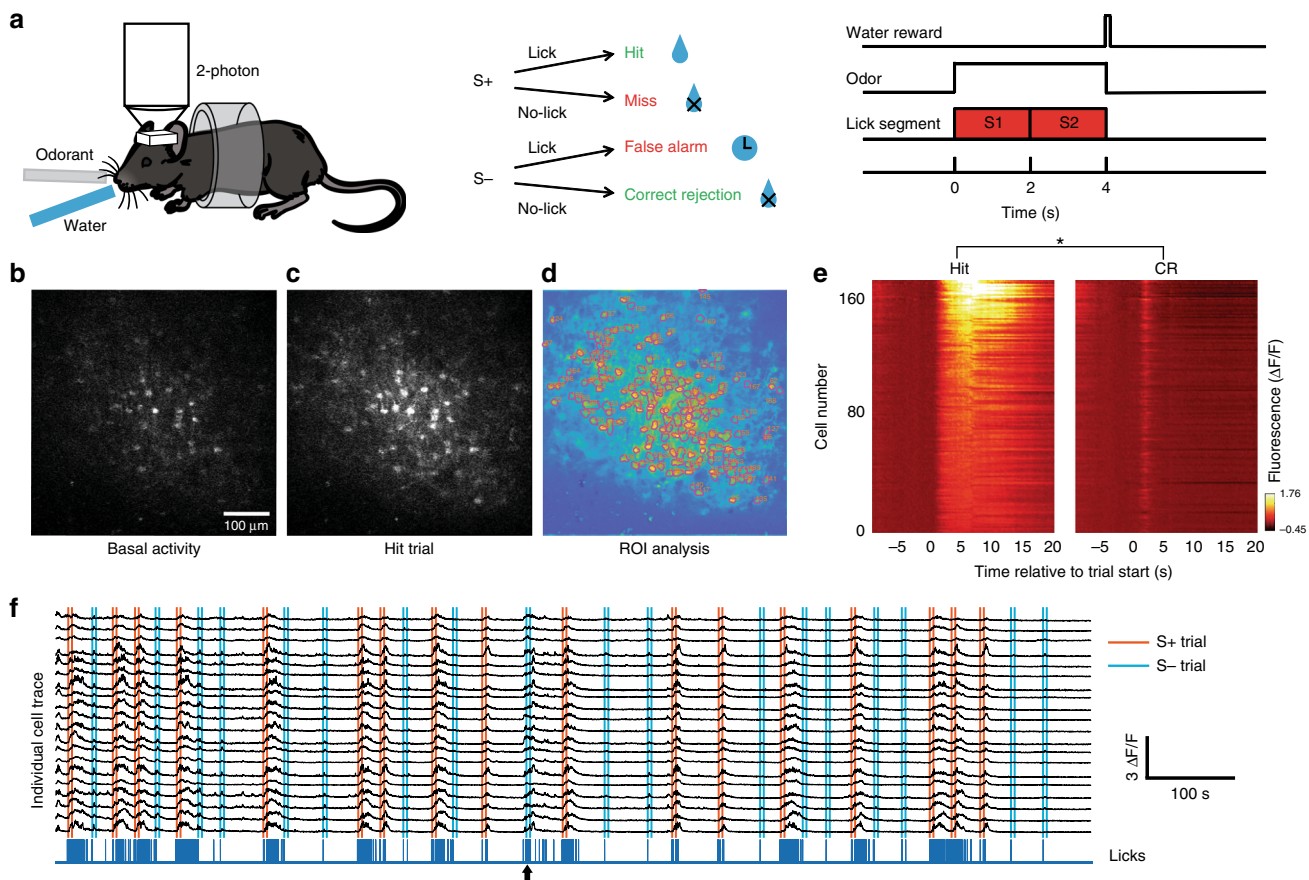

**Fig. 1 Two-photon Ca$^{2+}$ imaging of MLIs in head-fixed mice. a** go–no go task. Left: Two-photon imaging of a head-fixed mouse responding to odorants by licking on a water spout in response to the rewarded odorant in the go–no go task. Center: Scoring of decision-making. Right: time course of the trial. For water reward in Hit trials the animal must lick at least once in each of the two 2 s lick segments. **b–d** The panels in **b** and **c** show two-photon microscopy images of GCaMP6f fluorescence recorded from MLIs in a mouse proficient in the go–no go task (basal activity: before trial start, Hit trial: during reinforced odorant application). The color image in **d** shows the regions of interest identified using CalmAn software[31]. **e** Pseudocolor plots displaying the average per trial $\Delta F/F$ time course for Hit (left panel) and CR (right panel) odorants for all ROIs in this example. GLM analysis involving time periods pre-odorant (1 s before odorant onset), odorant (last second during odorant application) and reinforcement (1.5 s after reinforcement), and different events (Hits, Miss, CR and FA) yielded significant differences between reinforcement and pre-odorant ($p < 0.001$), between odorant and pre-odorant ($p < 0.001$), and between all interactions between these two period pairs and all events ($p < 0.01$, 2040 observations, 2028 degrees of freedom, $n = 170$ ROIs, 1 mouse, GLM F-statistic 234, $p < 0.001$). *Post-hoc ranksum $p < \text{pFDR} = 0.048$. **f** Neural ensemble activity. Black traces are the GCaMP6f fluorescence ($\Delta F/F$) time courses for a subset of the ROIs identified in the FOV in (**d**). This mouse was proficient (≥80% correct trials). Vertical lines: orange start and end of S+ odorant application, light blue is S−. The blue trace at the bottom shows the licks. All trials were Hits or CRs with the exception of the trial identified with the arrow that was a FA. Data shown in panels (**b–f**) are from one session (one mouse).

activity encodes for the identity of the stimulus. The learning curve for a session where the mouse achieved ≥80% in ~60 trials is shown in Fig. 3a. The first two principal components for a principal component analysis (PCA) of $\Delta F/F$ values for all MLIs in this session showed clear differences between S+ and S− trials for the reward and odorant periods for trials when the mouse was proficient (Fig. 3c), in contrast to the overlap observed between odorants for the naive mouse (Fig. 3b). This suggested that odorant identity (S+ vs. S−) could be decoded from $\Delta F/F$ values for proficient mice.

We utilized a linear discriminant analysis (LDA) to determine whether a hyperplane placed in the multidimensional space of $\Delta F/F$ values for all MLIs in the ensemble could decode the stimulus. The LDA was trained with $\Delta F/F$ for all trials minus one and was queried to identify the odorant in the remaining trial (see "Methods"). The bootstrapped 95% confidence interval (CI) for an LDA trained after shuffling stimulus was used as a control. The time course for LDA decoding accuracy for the session with

the learning curve in Fig. 3a is shown in Fig. 3d (Supplementary Fig. 7a shows the corresponding lick frequency time course). For trials with percent correct ≤65%, decoding accuracy raises above the shuffled 95% CI after water reinforcement (Fig. 3d, upper panel), while for proficient trials, decoding accuracy started rising above 95% CI shortly after odorant addition (Fig. 3d, lower panel). When tested in four mice LDA decoding accuracy for the odorant and reinforcement periods differed from shuffled LDA (Fig. 3e). A GLM analysis yielded statistically significant interactions between shuffled vs. reinforcement and for the interactions between (shuffled vs. odorant) × (proficient vs. naive) and (shuffled vs. reinforcement) × (proficient vs. naive) (GLM $p$ values <0.01 and <0.05, 24 observations, 18 d.f., $n = 4$ sessions, 4 mice, GLM F-statistic 16.4, $p < 0.001$). Post-hoc tests corrected for multiple comparison using false discovery rate (FDR)[33] yielded a statistically significant difference for decoding accuracy for either reinforcement or odorant vs. shuffled for proficient ($p < \text{pFDR} = 0.007$, $n = 4$ sessions, 4 mice). The LDA analysis

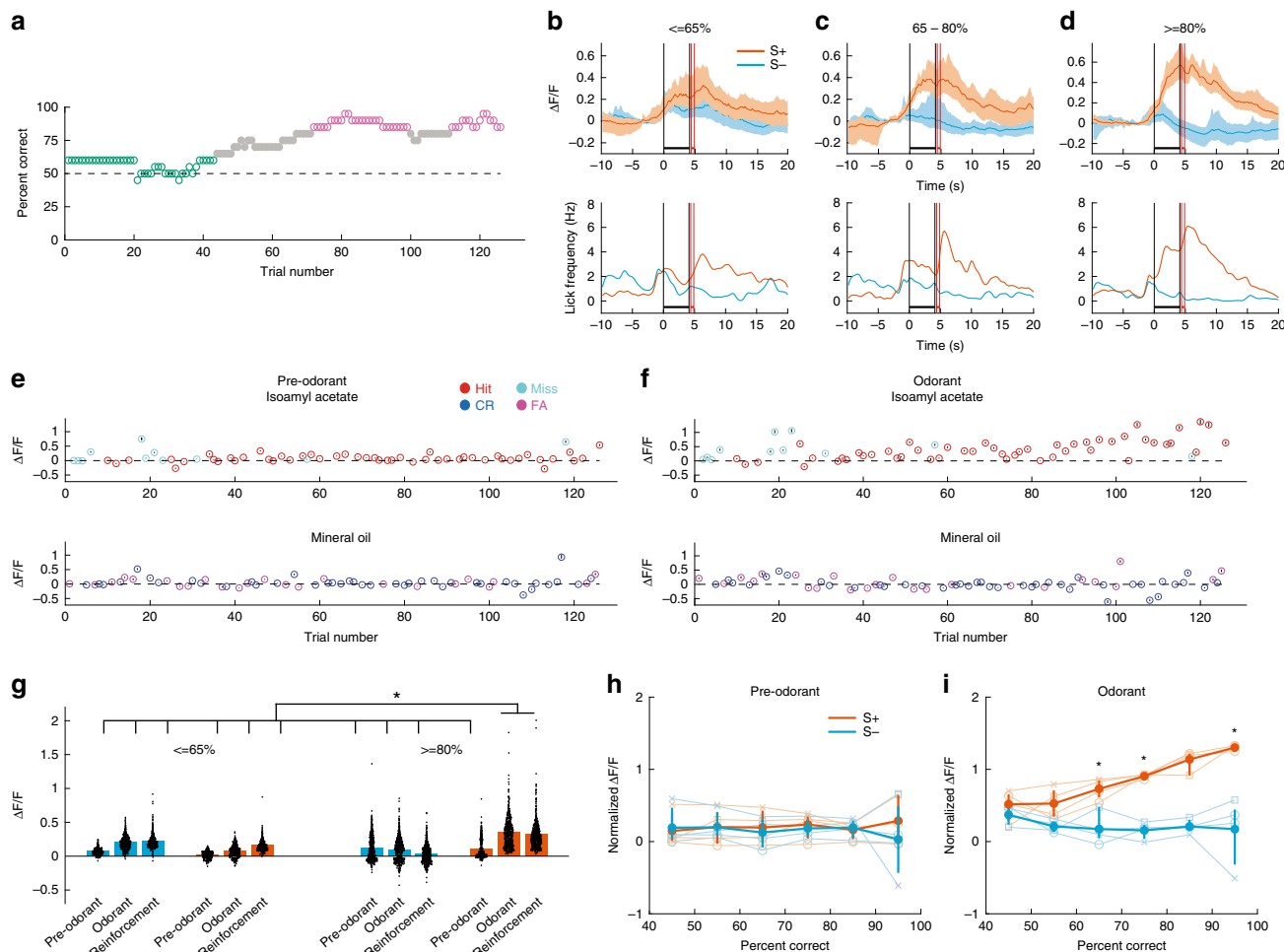

**Fig. 2 MLIs developed divergent responses during go–no go learning. a** Learning curve for a mouse discriminating 1% Iso from MO in the go–no go task. Magenta dots: proficient level (≥80% percent correct), green dots: naive level (percent correct ≤ 65%). $n = 20$ trials within a sliding window. **b**–**d** GCaMP6f fluorescence ($\Delta F/F$) time course averaged (mean ± 95% CIs) over all ROIs and trials falling within different proficiency windows shown for the session whose performance is shown in (**a**). **b** ≤65% (naive, 21 trials), **c** 65–80% (12 trials) and **d** ≥80% (proficient, 30 trials), 167 ROIs. Orange: S+, light blue: S− (the shaded area is the 95% CIs). The vertical black lines are odorant onset and removal and the red lines bound the reinforcement period. **e**, **f** Per trial $\Delta F/F$ averaged (mean ± 95% CIs) over all ROIs for 1 s before odorant application (**e** Pre-odorant) and in the last second during odorant application (**f** Odorant) shown for the session in (**a**). $n = 1$ mouse, 167 ROIs. **g** Violin plot showing per ROI $\Delta F/F$s for the session shown in (**a**) for the following time windows: pre-odorant (1 s before odorant application), odorant (the last second of odorant application) and reinforcement (one and a half seconds after onset of reinforcement). Per ROI $\Delta F/F$s are shown for trials falling within different proficiency windows: ≤65% (naive) and ≥80% (proficient). A GLM indicates that there are significant differences between S+ (orange) and S− (light blue), between time windows and between naive and proficient mice ($p < 0.001$, 5538 d.f., one mouse, *post-hoc ranksum/two-sided $t$ test $p$ value < pFDR = 0.043). **h**, **i** $\Delta F/F$ for four mice averaged (mean ± 95% CIs) over all the ROIs for all trials falling within different proficiency windows. $\Delta F/F$ average was calculated for 1 s before odorant application (**h**) and for the last second during odorant application (**i**). For the data during odorant application (**i**) GLM analysis indicated that there is a statistical significance for the interaction between percent correct and the identity of the odorant ($p < 0.001$, 43 d.f., $n = 4$ sessions, 4 mice, GLM F-statistic = 60, $p < 0.001$). GLM did not yield statistically significant differences for pre-odorant data (**h** $p > 0.05$, 43 d.f., $n = 4$ sessions, 4 mice, GLM F-statistic = 0.46, $p > 0.05$). *Post-hoc two-sided $t$ test, $p < $ pFDR = 0.018. Error bars are 95% CIs.

revealed that the accuracy for decoding the odorant identity from $\Delta F/F$ during the odorant period increases as the animal learns to differentiate between odorants.

**LDA decoding analysis with subsets of ROIs.** Since dimensionality of MLI neural activity is low (Supplementary Fig. 6) a question arose as to whether performing LDA analysis on a fraction of the ROIs in the FOV would result in accurate decoding of the stimulus. We performed the LDA decoding analysis for smaller numbers of ROIs ranging from 1 to 100. We used pseudorandom sampling of fifty unique subsets of ROIs. As expected, subsampling resulted in a decrease in accuracy of LDA decoding of the stimulus, but the decrease was

relatively small and even a single ROI yielded a decoding accuracy significantly different from the shuffled control. In the experiment shown in Fig. 3f decoding accuracy decreases from 94% with all ROIs in the FOV to 83% with a single ROI, while the shuffled trials yield 50% accuracy. Figure 3g shows the summary of this analysis for four sessions. A GLM analysis indicated that the decoding accuracy for the shuffled analysis was statistically different from accuracy with the subsets of ROIs ($p < 0.001$, 72 observations, 54 d.f., $n = 4$ sessions, 4 mice, GLM F-statistic = 23.6, $p < 0.001$). Thus, MLI activity encodes for the stimulus even when a single ROI is analyzed indicating that stimulus information encoded by MLI activity is highly redundant.

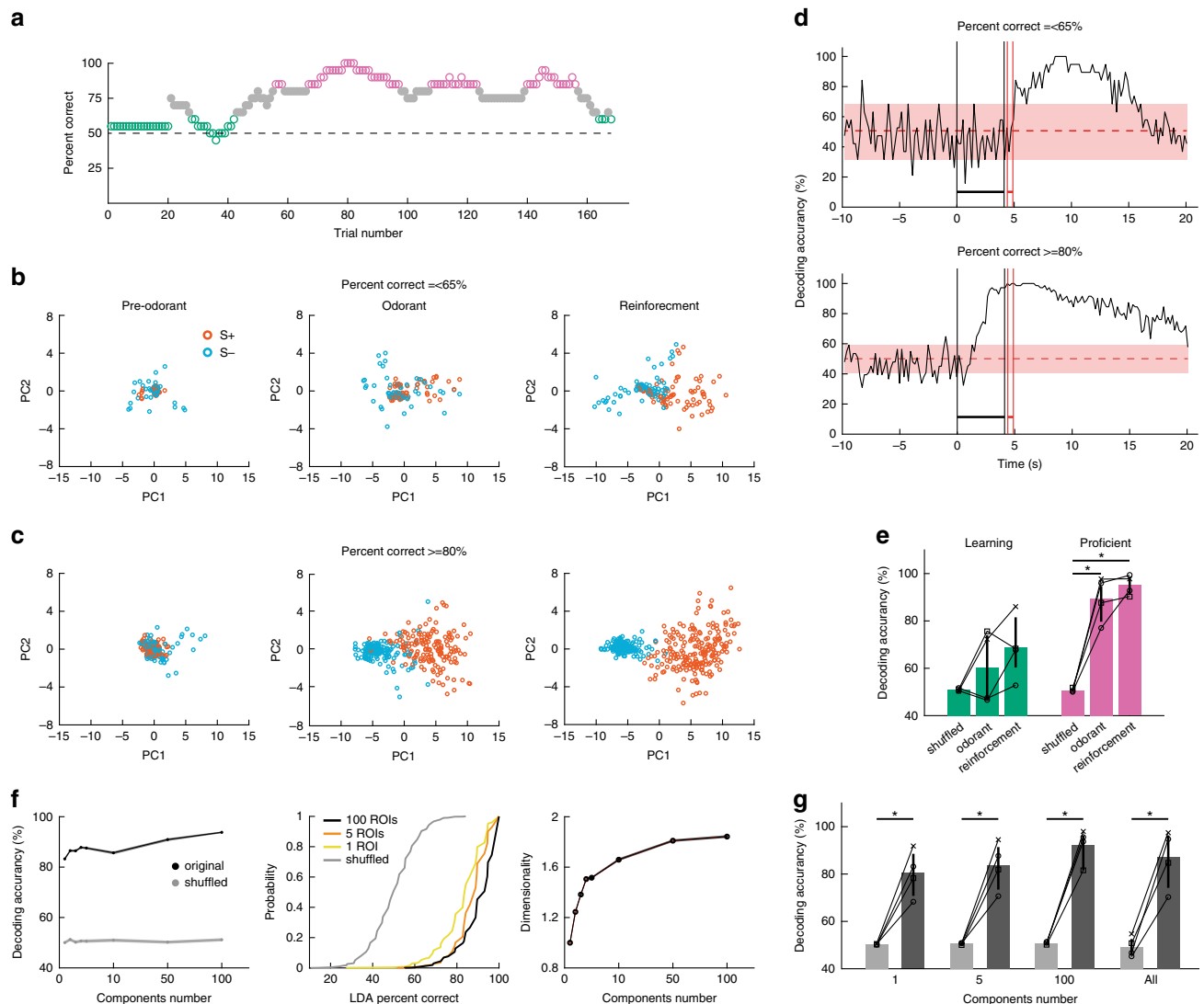

**Fig. 3 Stimulus decoding accuracy increases as the animal learns to discriminate odorants. a–d** Show behavioral performance (**a**), PCA analysis (**b, c**) and LDA analysis (**d**) for one session (one mouse). **a** Learning curve for a mouse performing the head-fixed go–no go task. Magenta dots: proficient level (≥80% percent correct), green dots: percent correct ≤65%. $n = 20$ trials within a sliding window. **b, c** First two principal components for the PCA for the changes in GCaMP6f fluorescence ($\Delta F/F$) in all the ROIs in the FOV. **b** The mouse was performing ≤65% correct, **c** the same mouse was performing ≥80% correct. The principal components are shown for three time periods: Pre-odorant: 1 s before odorant addition, odorant: 1 s before removal of the odorant, and Reinforcement: 1.5 s after reward. Orange circles: S+ trials, light blue circles: S− trials. **d** Decoding accuracy for the linear discriminant analysis (LDA) trained to predict odorant identity using $\Delta F/F$ for all ROIs in the FOV. Shade: 95% CI for LDA trained with shuffled odorant identity (line = 50%). The vertical black lines are odorant onset and removal and the red lines bound the reinforcement period. **e** Average LDA decoding accuracy (mean ± 95% CI) calculated from the ($\Delta F/F$) for all ROIs in the FOV from data recoded in four mice. A GLM analysis of decoding accuracy yielded statistically significant interactions between (shuffled vs. odorant) × (proficient vs. naive) and for (shuffled vs. reinforcement) × (proficient vs. naive) (GLM $p$ values <0.01 and <0.05, 24 observations, 18 d.f., $n = 4$ sessions, 4 mice, GLM F-statistic = 16.4, $p < 0.001$). *Post-hoc two-sided $t$ tests corrected for multiple comparison using the false discovery rate (FDR, $p < pFDR = 0.007$, $n = 4$ sessions, 4 mice). **f** Example of LDA decoding analysis performed with subsets of ROIs ranging from 100 to 1 ROI. The analysis was performed for one session for trials where the mouse was proficient (percent correct ≥ 80%). Left: Decoding accuracy (mean ± 95% CI) as a function of the number of ROIs. Center: Cumulative probability histograms for the decoding accuracy for the different number of ROIs and for the shuffled LDA. Right: Dimensionality as a function of the number of ROIs. **g** Summary graph showing the average decoding accuracy for different numbers of ROIs (mean ± 95% CI, 4 sessions, 4 mice). A GLM analysis indicated that the decoding accuracy for the shuffled analysis was statistically different from accuracy with the subsets of ROIs ($p < 0.001$, 72 observations, 54 d.f., $n = 4$ sessions, 4 mice, GLM F-statistic = 23.6, $p < 0.001$). *Post-hoc two-sided $t$ tests or ranksum $p < pFDR = 0.03$. Black: original data, light gray: shuffled. Error bars are 95% CIs.

**MLI odorant responses switch when valence is reversed.** In order to determine whether the MLIs responded to the chemical identity of the odorant, as opposed to responding to the valence (contextual identity: is the stimulus rewarded?), we reversed odorant reinforcement. When the reward was reversed for a proficient mouse the animal kept licking for the previous rewarded odorant resulting in a fall in percent correct below 50%,

and as the animal learned the new valence the percent correct raised back above 80% (Fig. 4a and Supplementary Fig. 5e, f). We analyzed how odorant-elicited changes in MLI $\Delta F/F$ varied in this reversal task. Figure 4b–d shows single trial examples for mean $\Delta F/F$ odorant responses when the reward is reversed. MLIs responded to Iso (S+) with an increase in $\Delta F/F$ before reversal (e.g. forward trial 52), maintained the odorant increase in $\Delta F/F$ to

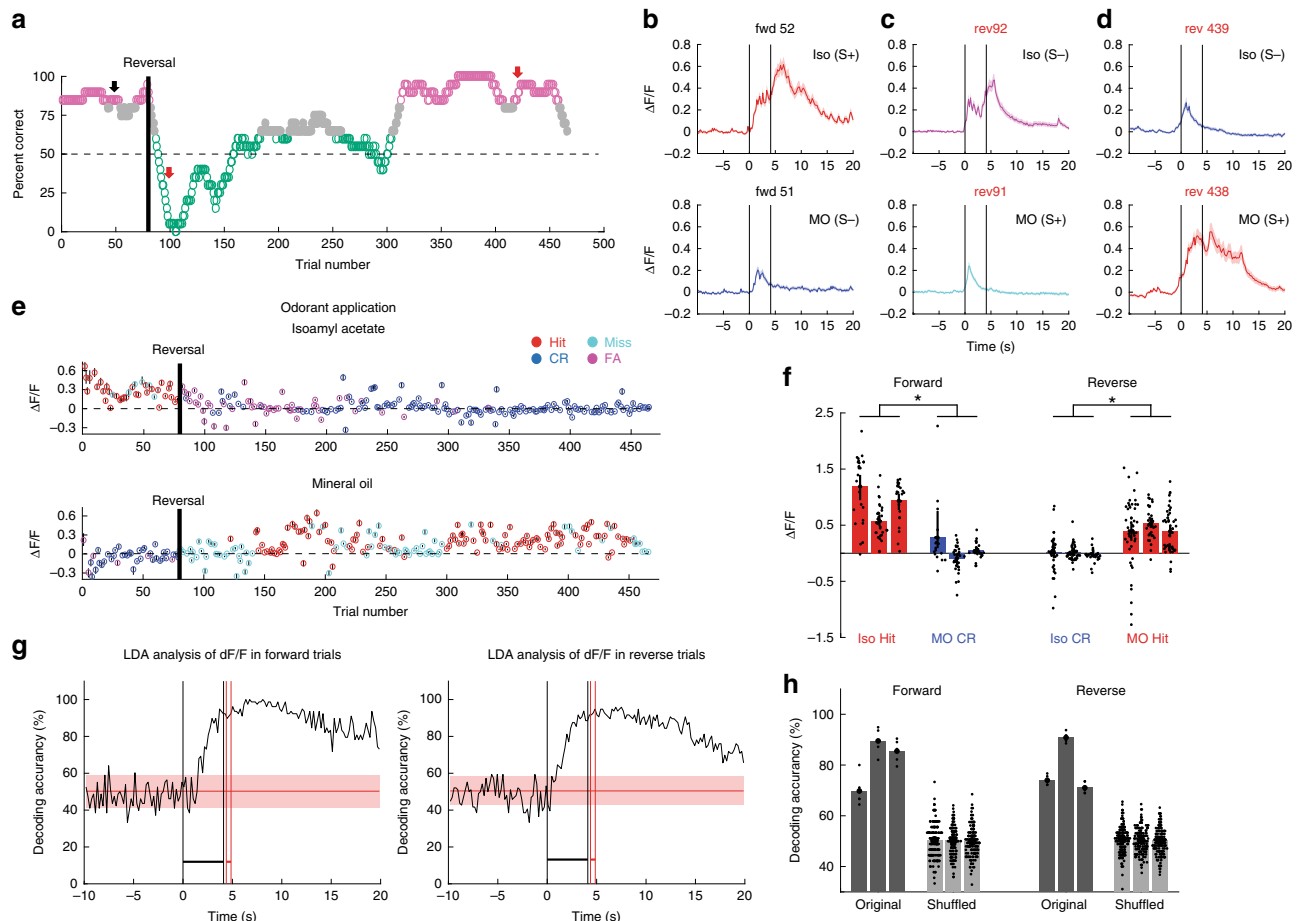

**Fig. 4 MLI odorant responses switched after reversal. a–e, g** Show behavioral performance (**a**), Δ*F/F* (**b–e**) and LDA analysis (**g**) for one session where odorant reinforcement was reversed (one mouse). **a** Learning curve showing the animal's behavior after reversal. Green ≤ 65%, magenta: ≥80% correct. Forward: S+ 1% Iso and S− MO. *n* = 20 trials within a sliding window. **b–d** Average Δ*F/F* time courses for example trials before (trials 51 and 52) and shortly after reversal (trials 91 and 92), and after the animal attained proficiency during reversal (trials 438 and 439). The vertical black lines are odorant onset and removal. **e** Per trial average Δ*F/F* during the last second of odorant application for the reversal task in Fig. 4a. The upper plot shows the responses to Iso and the lower plot shows responses to MO. *n* = 1 mouse, 185 ROIs. **f** Average odorant-induced changes (mean ± 95% CI) in Δ*F/F* recorded from three animals computed for trials when the animal was behaving ≥80% correct. Red: Hit, Blue: CR. A GLM analysis indicates that there is a significant difference in odorant-induced changes in Δ*F/F* for both odorant and reversal (*p* < 0.001, 12 observations, 8 d.f., *n* = 3 sessions, 3 mice, GLM F-statistic = 14.8, *p* < 0.01). **g** Time course for decoding accuracy computed with LDA analysis of Δ*F/F* data for all ROIs in the FOV. The shade is the 95% CI of decoding accuracy calculated with shuffled odorant identity. The vertical black lines are odorant onset and removal and the red lines bound the reinforcement period. **h** Average decoding accuracy (mean ± 95% CI) for three animals calculated with odorant-induced changes in Δ*F/F* from trials where the animal was performing ≥80% correct. Dark-gray: original, light gray: shuffled. A GLM analysis indicates that differences in decoding accuracy are statistically significant between original and shuffled (*P* < 0.001), but not between forward and reverse (*p* > 0.05, *n* = 12, 8 d.f., *n* = 3 sessions, 3 mice, GLM F-statistic = 16, *p* < 0.001).

what was then the unrewarded odorant (Iso, S−) immediately following reversal (reverse trial 92) and switched responses with increases to the new rewarded stimulus (MO, S+) when they became proficient in the reversal task (reverse trial 438) (Δ*F/F* for the last second of the odorant period is shown for all trials in Fig. 4e). When repeated for three different mice the odorant-induced Δ*F/F* changes reversed for proficient mice (≥80% correct) when the valence was reversed (Fig. 4f). GLM analysis indicated that there was a significant difference in odorant-induced changes in Δ*F/F* for the odorant and the reversal (*p* < 0.001, 12 observations, 8 d.f., *n* = 3 sessions, 3 mice, GLM F-statistic = 14.8, *p* < 0.01). Thus, after successful reversal the Δ*F/F* time course switched for the two odorants: the stimulus-induced increase in Δ*F/F* took place for the reinforced odorant, not for the chemical identity of the odorant.

Next we computed the accuracy for decoding the reinforced odorant for trials when the animal was proficient in either the

forward or reverse trials using LDA analysis. Figure 4g shows for the reversal session in Fig. 4a that for the proficient animal decoding accuracy rises above the 95% CI calculated with shuffled trials shortly after odorant addition for both forward and reverse trials. Finally, the results of the forward and reverse LDA analysis for three mice that are proficient showed that decoding accuracy for the stimuli differed from shuffled LDA (Fig. 4h). GLM analysis indicated that differences in decoding accuracy were statistically significant between original and shuffled (*P* < 0.001), but not between forward and reverse (*p* > 0.05, 12 observations, 8 d.f., *n* = 3 sessions, 3 mice, GLM F-statistic = 16, *p* < 0.001). These results indicated that for the proficient mouse it is possible to decode contextual identity, suggesting that MLI activity encodes for valence.

**MLI activity encodes the contextual identity in error trials.** To gain a better understanding of the information on odorant

**Table 1 Percent of ROIs displaying a change in ΔF/F during odorant application.**

| Calculation | Percent responsive | Number | Number of ROIs |
|---|---|---|---|
| Percent of ROIs responding in Hit trials | 99.9 ± 0.23 | 21 Time series, 5 mice | 74–199 |
| Percent of ROIs responding in CR trials | 82.3 ± 26 | 21 Time series, 5 mice | 74–199 |
| Percent of ROIs responding in Miss trials | 97.8 ± 6.5 | 13 Time series, 5 mice | 105–199 |
| Percent of ROIs responding in FA trials | 78 ± 29 | 15 Time series, 5 mice | 74–199 |
| Percent of ROIs responding in Miss trials that also respond in Hit trials | 99.96 ± 0.15 | 13 Time series, 5 mice | 105–199 |
| Percent of ROIs responding in FA trials that also respond in Hit trials | 99.8 ± 0.47 | 15 Time series, 5 mice | 74–199 |

This calculation was performed in time series that included at least one Miss or FA trial when the animal was proficient (percent correct ≥80%). The ROI was classified as responsive when ΔF/F increased above or decreased below baseline ΔF/F by 2.5 × SD. Baseline mean and SD were calculated for the time interval 10–2 s before odorant onset.

valence present in the responses of the MLI ensemble we asked whether stimulus decoding accuracy calculated with LDA for proficient mice differed between correct (Hits and CRs) and incorrect trials (Miss and FAs). If information in MLI activity reflects the outcome of the trial, we would expect that decoding accuracy would be lower for incorrect trials. On the other hand, if information encoded by MLI activity reflects the stimulus regardless of trial outcome decoding accuracy would not differ between correct and incorrect trials.

We performed this analysis for sessions that included at least one error trial (Miss or FA) when the animals were proficient. In these time series the majority of the ROIs exhibited changes in ΔF/F during the odorant application regardless whether the trial was a correct response (Hit or CR) or an error (Miss, FA) (Table 1). In addition, virtually all ROIs that responded with changes in ΔF/F during error trials also responded during Hit trials (Table 1, and see examples of ΔF/F time courses for single ROIs in Supplementary Fig. 8). As expected, in both the odorant application and reinforcement periods mean lick frequency for FA was higher than for CR and mean lick frequency for Miss trials was lower than Hits (Supplementary Fig. 9a, GLM analysis $p < 0.001$, 48 observations, 40 d.f., $n = 6$ sessions, 5 mice, GLM F-statistic = 68, $p < 0.001$, post-hoc two-sided $t$ test $p < $ pFDR = 0.036). In contrast, ΔF/F did not differ between Hits vs. Miss and CR vs. FA (Supplementary Fig. 9b, c). GLM analysis indicated that differences were not significant between Hits vs. Miss and CR vs. FA ($p > 0.05$), while Hits/Miss differ from CR/FA ($p < 0.01$, 48 observations, 40 d.f., $n = 6$ sessions, 5 mice, GLM F-statistic = 8.48, $p < 0.001$, post-hoc two-sided $t$ test $p < $ pFDR = 0.027). To survey the information encoded in MLI activity in trials with different outcomes in proficient mice, we utilized LDA analysis to decode the stimulus (Supplementary Fig. 9d: forward go–no go sessions, Supplementary Fig. 9e: reverse sessions). Decoding accuracy differed from shuffled for all outcomes and time periods (Supplementary Fig. 9d, two-sided $t$ test, $p < $ pFDR = 0.025, $n = 6$ sessions, 5 mice, Supplementary Fig. 9e two-sided $t$ test, $p < $ pFDR = 0.012, $n = 3$ sessions, 3 mice). In addition, GLM analysis did not find a significant difference between outcomes (Hit, Miss, CR or FA) or time period (odorant vs. reinforcement) indicating that MLI activity reflects the stimulus regardless of trial outcome (Supplementary Fig. 9d, $p > 0.05$, 48 observations, 40 d.f., $n = 6$ sessions, 5 mice, GLM F-statistic = 1.49, $p > 0.05$, Supplementary Fig. 9e, $p > 0.05$, 24 observations, 16 d.f., $n = 3$ sessions, 3 mice, GLM F-statistic = 1.12, $p > 0.05$). This analysis determined that odorant-induced MLI Ca$^{2+}$ changes carry information on the stimulus, as opposed to the outcome.

**LR correlates with MLI activity during reinforcement**. To explore whether the observed MLI ensemble activity recorded in vermis is directly related to licking, as found in Crus II[34,35], we examined the correlation between ΔF/F and the LR. When the animal was proficient the animal licked at least once in each of the two lick segments during application of the S+ odorant, increased licking after receiving the water reward and refrained from licking for the S− trials (Fig. 5a). For the proficient mouse the lick frequency diverged between S+ and S− trials shortly after addition of the odorant (Fig. 5b) and this divergence was evidenced by a large decrease in the $p$ value calculated with a ranksum test comparing licking between S+ and S− trials (Fig. 5c).

In order to explore the relationship of MLI activity to licking we proceeded to examine the correlation between ΔF/F time course and the LR and between the derivative of the ΔF/F time course (D$_t$ΔF/F) and the derivative of the lick rate (D$_t$LR)[34] and plotted their relationship during different time periods for proficient mice. Figure 5d–g shows examples for several trial outcomes for the time course for ΔF/F and the LR. Examples of these correlations for a single session for a proficient mouse are shown in Fig. 5h, i. In this example the correlation between ΔF/F and the LR was larger during the reinforcement period ($\rho = 0.73$, $p$ value < 0.001) compared to the odorant ($p = 0.19$, $p$ value < 0.001) and pre-odorant ($p = -0.06$, $p$ value < 0.05) periods (Fig. 5h). Similarly, the correlation between D$_t$ΔF/F and D$_t$LR was significant for the reinforcement period ($\rho = 0.28$, $p < 0.001$), and was smaller for both the pre-odorant ($\rho = -0.08$, $p < 0.05$) and odorant ($\rho = -0.045$, $p > 0.05$) periods (Fig. 5i). We proceeded to compare the correlations between these parameters in several mice. As found in Crus II[21], ΔF/F and the LR (Fig. 5j) and D$_t$ΔF/F and D$_t$LR (Fig. 5k) were positively correlated during the reinforcement period. Furthermore, the correlations were lower during the pre-odorant and odorant periods (Fig. 5j, k). A two-sided $t$ test indicated that there was a statistically significant difference in the correlation coefficient between the reinforced vs. the odorant or pre-odorant periods for ΔF/F vs. LR ($p < $ pFDR = 0.05, $n = 6$ sessions, 5 mice) and between the reinforced and odorant periods for D$_t$ΔF/F vs. D$_t$LR ($p < $ pFDR = 0.016, $n = 6$ sessions, 5 mice). These data suggested that changes in ΔF/F during the reinforcement period reflect changes in lick activity, while changes in ΔF/F during the odorant period are less dependent on licks, and maybe dependent on multiple variables.

**Relationship between changes in ΔF/F and lick frequency**. We performed complementary studies of ΔF/F in CR trials when the mouse did not lick and for time courses aligned to the beginning of ΔF/F changes after odorant addition. We compared ΔF/F and lick frequency during the odorant application period for CR trials when the animal did not lick during the two 2 s odorant response periods vs. CR trials when the animal licked (Supplementary Fig. 10). We did not find a difference between ΔF/F measured during the odorant period for the CR trials when the animal did not lick, compared to CR trials when the animal did lick during odorant application (Supplementary Fig. 10c, GLM yielded no significant difference for the two types of CR trials, or for the

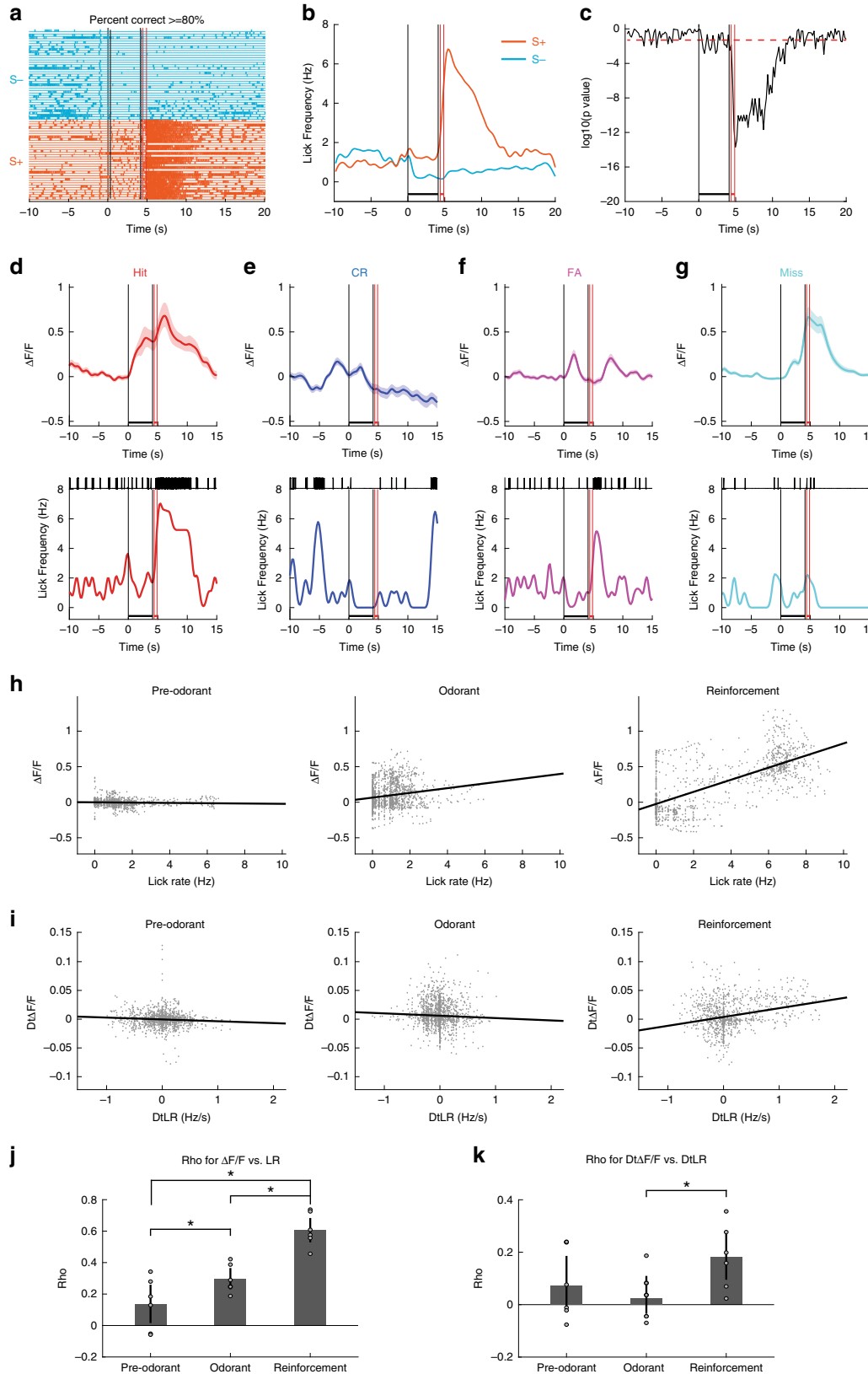

different time periods, $p > 0.05$, 36 observations, 30 d.f., $n = 6$ sessions, 5 mice, GLM F-statistic = 0.88, $p > 0.05$).

Furthermore, we analyzed the relationship between the time course of $\Delta F/F$ and lick frequency in the time period shortly after odorant application when $\Delta F/F$ increases for both S+ and S−, before $\Delta F/F$ decreases for S− (and keeps increasing for S+). In Crus II, where neural activity of MLIs is thought to reflect licks, $\Delta F/F$ increases whenever there is an increase in licking frequency[34,35]. We aligned the traces to the point where the time derivative for $\Delta F/F$ increased above 0.03. We found that $\Delta F/F$ increased for both S+ and S− (Supplementary Fig. 11b, GLM analysis yielded a significant change as a function of time,

**Fig. 5 Changes in the lick rate correlate with changes in the MLI activity during the reinforcement period. a–g** are examples of lick traces (**a**), lick frequency (**b**), lick ranksum $p$ value (**c**) and $\Delta F/F$ and lick traces (**d–g**) for one session (one mouse). **a** Examples of per trial lick traces when the mouse was learning to discriminate the odorants when the animal was proficient (≥80% correct, orange traces: S+ trials, light blue traces: S− trials). **b** Average lick frequency for S+ (orange) and S− (light blue) trials shown in (**a**). **c** $p$ value for a ranksum test estimating the difference in licks between the S+ and S− odorants for the example in (**a**). **d–g** Single trial examples of average $\Delta F/F$ (±95% CI, shade, $n = 105$ ROIs, upper panels) and the corresponding lick frequency and lick traces (lower panels). **h** Relationship between the per trial average $\Delta F/F$ and the lick frequency shown per time point for all trials in a go–no go session (one session, one mouse). Time points are segregated within the last second before odorant addition (Pre-odorant), during the last second of odorant addition (Odorant) and during the 1.5 s after reward (Reinforcement). Correlation coefficients and $p$ values for these time periods are: Pre-odorant: 0.058, $p < 0.05$. Odorant: 0.19, $p < 0.001$. Reinforcement: 0.73, $p < 0.001$. *$p < $ pFDR $= 0.05$, $n = 6$ sessions, 5 mice. **i** Relationship between the derivative of average $\Delta F/F$ and the derivative of lick frequency shown per time point for all trials in a go–no go session (one session, one mouse). Time points are segregated within the last second before odorant addition (Pre-odorant), during the last second of odorant addition (Odorant) and during 1.5 s after reward (Reinforcement). Correlation coefficients and $p$ values for these time periods are: Pre-odorant: $-0.076$, $p < 0.01$. Odorant: $-0.045$, $p > 0.05$. Reinforcement: 0.28, $p < 0.001$. *$p < $ pFDR $= 0.016$, $n = 6$ sessions, 5 mice. **j** Correlation coefficients for the relationship between the average $\Delta F/F$ and lick frequency for six sessions (five mice). The correlation coefficient is significantly different between reinforcement and odorant (*two-sided $t$ test $p < $ pFDR $= 0.05$, $n = 6$ sessions, 5 mice. **k** Correlation coefficients for the relationship between the derivative of average $\Delta F/F$ and the derivative of lick frequency for six sessions (five mice). The correlation coefficient is significantly different between reinforcement and odorant (*two-sided $t$ test $p < $ pFDR $= 0.016$, $n = 6$ sessions, 5 mice. In **b**, **c**, **d–g** the vertical black lines are odorant onset and removal and the vertical red lines bound the reinforcement period. All data shown in this figure are for proficient mice (percent correct ≥ 80%). Error bars are 95% CIs.

$p < 0.05$, and no significant difference between S+ and S−, $p > 0.05$, 132 observations, 128 d.f., 6 sessions, 5 mice, GLM F-statistic $= 3.1$, $p < 0.05$. In contrast, in this time period there was no increase in lick frequency (Supplementary Fig. 11a, GLM analysis yielded no significant difference as a function of time, $p > 0.05$ and a significant change between S+ and S−, $p < 0.001$, 132 observations, 128 d.f., 6 sessions, 5 mice, GLM F-statistic $= 21$, $p < 0.001$). The data on the relationship between lick frequency and $\Delta F/F$ indicate that although there is a dependence between these two variables, the dependence is not consistent with a direct relationship between $\Delta F/F$ and lick frequency, as found in Crus II.

**Do the MLIs respond to reward value?** The correlation between $\Delta F/F$ time course and the LR (Fig. 5h) raises the question whether $\Delta F/F$ reflects the reward value as opposed to the valence of the odorant. Here we define the valence of the odorant as indicating whether the stimulus is good or bad[36]. Thus, valence is a binary measure of an emotion that is reflected by the motivation to receive reward, whereas reward value, in the present paradigm, is related to the amount of sugar water delivered. To evaluate whether $\Delta F/F$ changes are dependent on reward value we recorded changes in $\Delta F/F$ from mice performing a go–go task where both odorants were rewarded equally and we varied the volume of sugar water delivered for successful trials. As expected, the mouse responded to both odorants (Supplementary Fig. 12a). Also, as expected, when the volume of reward was increased, the lick frequency increased during the delivery of sugar water (wet licks, Supplementary Fig. 12b), but not during dry licking before reward (dry licks, Supplementary Fig. 12b). A GLM analysis yielded a statistically significant difference for dry vs. wet licking and for the interaction between the volume of sugar water delivered and dry vs. wet ($p < 0.001$, 198 observations, 194 d.f., 1 session, 1 mouse, GLM F-statistic $= 21.8$, $p < 0.001$). In contrast, a GLM analysis of the changes in $\Delta F/F$ as a function of volume delivered in dry and wet lick conditions did not yield statistically significant changes (Supplementary Fig. 12c, $p > 0.05$, 198 observations, 194 d.f., 1 session, 1 mouse, GLM F-statistic $= 1.7$, $p > 0.05$). Finally, if MLI activity reflected reinforcement value $\Delta F/F$ should show a positive correlation with volume of sugar water delivered (see Fig. 3 of ref. [34]). We did not find a significant correlation between lick frequency and $\Delta F/F$ for either dry (Supplementary Fig. 12d, left panel, $\rho = 0.16$, $p > 0.05$) or wet licking (Supplementary Fig. 12d, right panel, $\rho = -0.66$, $p > 0.05$). This experiment indicated that MLI activity does not reflect

reward value, and is consistent with MLI activity reflecting valence.

**Contextual identity contributes to GLM fit of MLI activity.** In order to understand the contribution of different variables to MLI activity, we implemented a GLM to quantify the dependence of the average $\Delta F/F$ on the different behavioral and stimulus variables[37]. We included event variables, whole trial variables, and continuous variables (Fig. 6a, "Methods"). Continuous variables quantified kinematics including the LR and the derivative of LR (Fig. 5) and body velocity and body acceleration of movements made by the animal during the trial (Supplementary Fig. 3). The identity of the odorant (S+ /S− odorant) was an event variable that increased from zero to one during the time for odorant application. Finally, whole trial variables were accuracy (1 for correct and 0 for incorrect responses), reinforcement history (1 for reinforcement in the last trial, 0 otherwise) and percent correct behavior calculated in a window of 20 trials.

Black traces in Fig. 6b show examples of the fit of the GLM model to average $\Delta F/F$ time courses for four example trials with different outcomes. The fit traces in black largely overlap with the recorded data. When quantified over the different periods within a trial (pre-odorant, odorant and reinforcement), we found that GLM explained a substantial percent of the average $\Delta F/F$ variance ranging from 23.5 to 95% (Fig. 6c). We next quantified the relative contribution of the different variables to the ensemble activity in the three time periods. Figure 6d shows the contributions to the GLM fit by the variables that make the largest percent contribution to the fit (body kinematics, licks, reinforcement history, and odorants). Odorant (S+ vs. S−) was the dominant contributor to the activity during odorant application (Fig. 6d, lower left panel, asterisks show differences with ranksum or two-sided $t$ test $p < $ pFDR $= 0.033$, $n = 6$ sessions, 5 mice). Furthermore, variables describing the licks contributed to the GLM fit during the outcome period (Fig. 6d, upper right panel, *$p < $ pFDR $= 0.016$ for a two-sided $t$ test, $n = 6$ sessions, 5 mice). In contrast, the other two variables (body kinematics and reinforcement history) did not differ in contribution during the different periods ($p > $ pFDR $= 0.017$, ranksum test). These data indicate that odorant identity contributes to modeling MLI activity during the odorant application period while licks contribute to MLI activity during the reinforcement period.

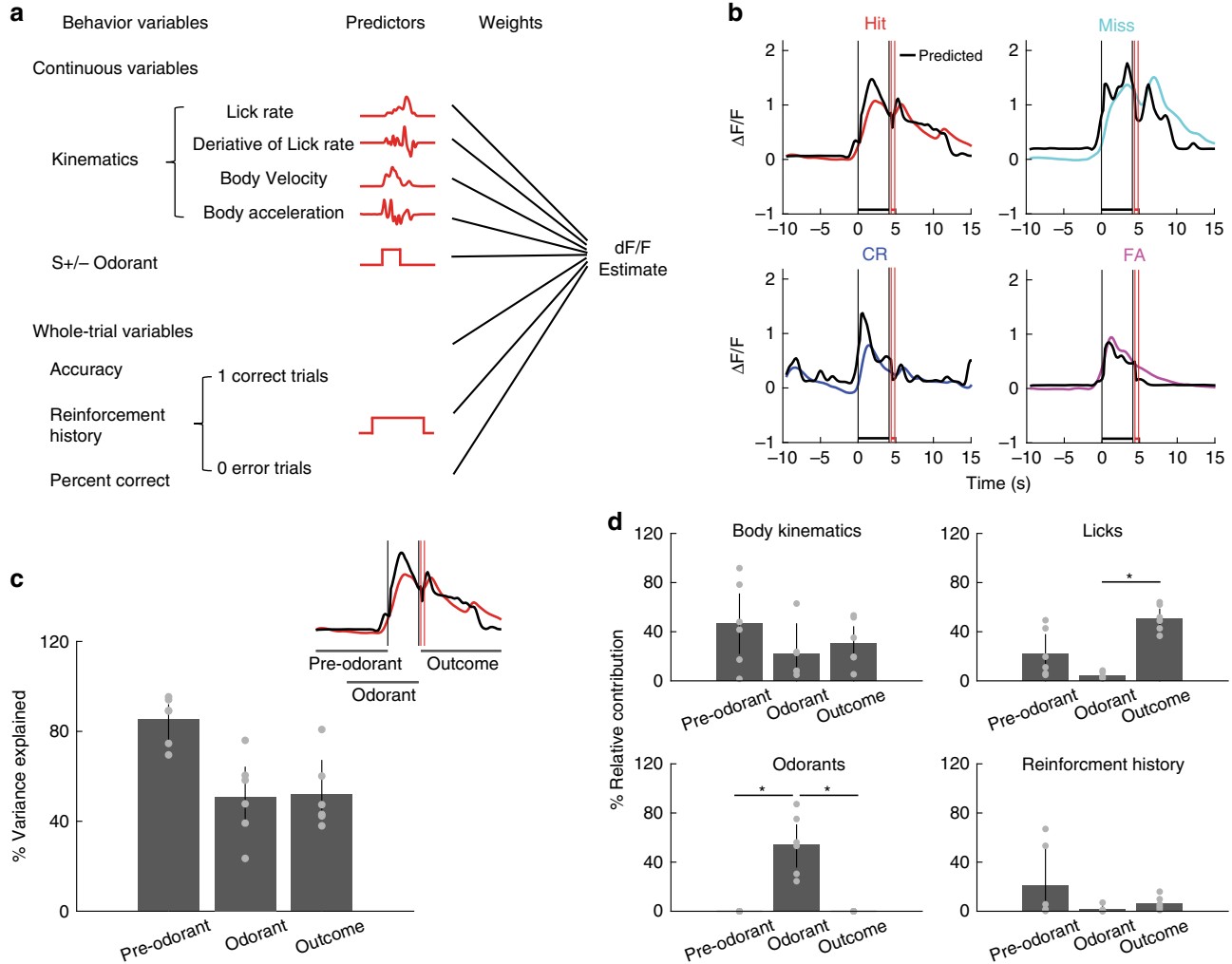

**Fig. 6 Fit of MLI activity with a generalized linear model. a** Schematic illustration of the variables used by the GLM model to quantify the relationship between average $\Delta F/F$ and behavioral variables and odorant valence on a per trial basis. **b** Examples of GLM best-fit (black) and actual $\Delta F/F$ $Ca^{2+}$ traces for Hit (red), Miss (cyan), CR (blue), and FA (magenta) trials. The vertical black lines are odorant onset and removal and the red lines bound the reinforcement period. **c** Percent of the variance explained by GLM in the different time periods (Pre-odorant, odorant and outcome, $n = 6$ sessions, 5 mice, error bars are 95% CIs). The inset shows the time intervals used for the different periods. **d** Relative contribution of kinematics, licks, odorants, and reinforcement history to the GLM fit for fits in the different time periods (pre-odorant, odorant, outcome). The results showed that odorants and licks play an important role in contributing to the activity during odorants and outcome periods, respectively, while kinematics contribute to the activity in all three time periods. Asterisks for licks (upper right panel) and odorants (lower left panel) denote significant differences determined by ranksum and two-sided $t$ test ($p < \text{pFDR} = 0.033$, $n = 6$ sessions, 5 mice). There are no significant differences for Kinematics or Reinforcement history ($p > \text{pFDR} = 0.017$, ranksum test). Error bars are 95% CIs.

**Chemogenetic inhibition leads to impaired go–no go learning.** In order to determine whether activity of MLIs plays a role in behavioral responses in the go–no go task we used a Cre-dependent AAV virus to express the inhibitory DREADDs receptor hM4Di in MLIs in six PV-Cre mice (hM4Di group)[38]. To control for off-target effects of clozapine-N-oxide (CNO)[39], we injected PV-Cre mice with Cre-dependent mCherry AAV virus in another group of six mice (control group) (see Supplementary Fig. 13a, b for virus expression). Animals from both groups were trained to differentiate two odorant mixtures (S+: 0.1% of 60% heptanal + 40% ethyl butyrate, S−: 0.1% of 40% heptanal+60% ethyl butyrate) for 3–4 sessions encompassing 400–500 trials in the go–no go task. The animals were injected intraperitoneally (IP) with saline 40 min before the start of the sessions (control-saline and hM4Di-saline). One to two weeks later, mice were trained again to differentiate between the same odorant mixtures, but they were injected IP with CNO (3 mg kg⁻¹)

40 min before the start of the session (control-CNO and hM4Di-CNO).

Mice in all groups with the exception of the hM4Di-CNO attained proficiency (≥80% percent correct) (Fig. 7b, d). GLM analysis indicated that there is a statistically significant interaction between CNO drug treatment and hM4Di expression ($p < 0.01$, 24 observations, 20 d.f., $n = 6$ mice, GLM F-statistic = 13.7, $p < 0.001$) and post-hoc tests indicated that the hM4Di expressing group differs between CNO and saline ($p < \text{pFDR} = 0.025$, $n = 6$ mice per group), while there are no significant differences between CNO and saline for control mice ($p > \text{pFDR}$, $n = 6$ mice per group), indicating an effect of CNO-induced inhibition of MLIs expressing hM4Di on behavioral output and the absence of off-target CNO effects.

We proceeded to compare the time course for lick frequency between different conditions for trials when the mouse was proficient (≥80% correct, Fig. 7e). The time course for lick

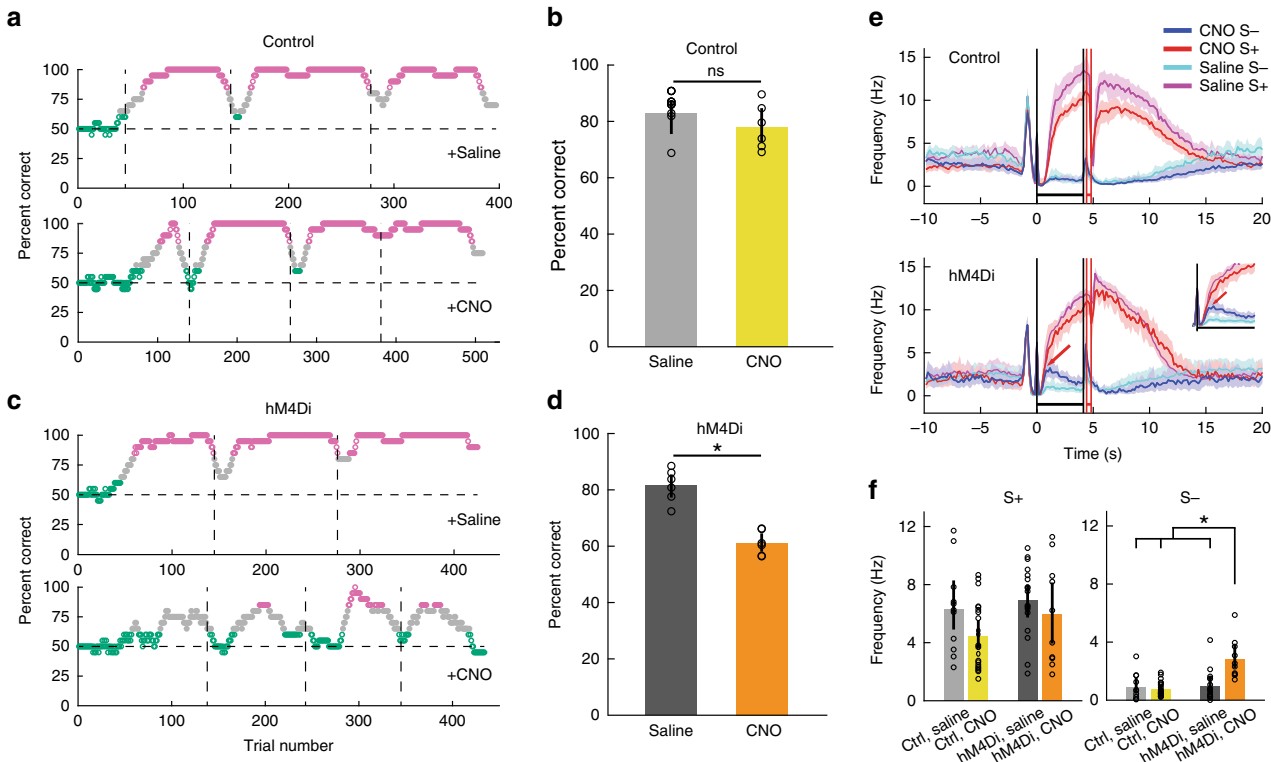

**Fig. 7 Chemogenetic inhibition of MLI activity impairs associative learning. a, c** Examples of behavioral performance in a go–no go task where mice learned to differentiate between two odorant mixtures (S+: 0.1% of 60% heptanal+40% ethyl butyrate, S−: 0.1% of 40% heptanal+60% ethyl butyrate). Green: ≤65% correct, magenta: ≥80% correct. The vertical lines are the boundaries between different daily sessions. $n = 20$ trials within a sliding window. **b, d** Behavioral performance (mean ± 95% CIs, $n = 6$). **b** Mice expressing mCherry in MLIs (control mice). **d** Mice expressing hM4Di in MLIs. A GLM analysis indicates that there is a statistically significant interaction between CNO drug treatment and hM4Di expression ($p < 0.01$, 24 observations, 20 d.f., $n = 6$ mice, GLM F-statistic = 13.7, $p < 0.001$) and post-hoc unpaired two-sided $t$ tests indicate that the hM4Di expressing group differs between CNO and saline (asterisks, $p < pFDR = 0.025$, $n = 12$ mice). **e** Average lick frequency time course for the control (upper panel) and hM4Di (lower panel) groups (mean ± 95% CI, shade). The arrow in the inset shows that for S− trials of CNO treated hM4Di animals (blue), the animal kept licking for a longer time than S− trials for mice injected with saline (cyan). The vertical black lines are odorant onset and removal and the red lines bound the reinforcement period. **f** Mean lick frequency (±95% CI) during the initial portion of the odorant application period (0.8–1.8 s) for mice performing ≥80% correct (left: S+, right: S−). GLM analysis did not find a statistically significant difference for treatment or hM4Di expression (or interactions) for S+ ($p > 0.05$, 60 observations, 56 d.f., $n = 6$ mice, GLM F-statistic = 2.96, $p > 0.05$), but found a difference for CNO treatment x hM4Di expression for S− ($p < 0.001$, 60 observations, 56 d.f., $n = 6$ mice, GLM F-statistic = 12.8, $p < 0.001$). *$p < pFDR = 0.03$ for post-hoc two-sided t test. Error bars are 95% CIs.

frequency differed between hM4Di mice injected with CNO and the other conditions. With the exception of hM4Di + CNO, shortly after odorant addition (~0.8 s) the lick frequency for S− did not increase beyond ~1 Hz while the lick frequency for S+ kept increasing beyond 8 Hz (Fig. 7e). In contrast, for hM4Di + CNO lick frequency for S− kept increasing beyond 1 Hz and did not diverge from the S+ time course until it reached 2.5 Hz at a later time point (~1 s, Fig. 7e, lower panel) likely reflecting slow decision-making. This would explain why the hM4Di + CNO mice accumulated errors in the go–no go task (Fig. 7d). In order to quantify this difference in lick frequency we calculated the lick frequency 0.8–1.8 s after odorant addition (Fig. 7f). For S+ GLM analysis did not find a statistically significant difference for treatment or hM4Di expression (or interactions) for the rewarded odorant trials (Fig. 7f, left panel, $p > 0.05$, 60 observations, 56 d.f., $n = 6$ mice, GLM F-statistic = 2.96, $p > 0.05$). Yet GLM found a statistically significant difference for S− for the interaction between CNO and hM4Di expression (Fig. 7f, right panel, $p < 0.001$, 60 observations, 56 d.f., $n = 6$ mice, GLM F-statistic = 12.8, $p < 0.001$). We obtained a similar impairment of performance in a separate set of experiments with two hM4Di mice and two controls where animals discriminated between 1% Iso and MO, and CNO was applied when the animal was naive, and we

reversed the reward (Supplementary Fig. 13). These data indicate that inhibition of MLIs causes a slower differential lick response to odorants and impaired behavioral performance.

**Modeling MLI modulation of Purkinje cell output.** In order to understand the circuit basis for chemogenetic interference in the go–no go associative learning olfactory discrimination task we generated a simple computational model of MLI-PC interactions. Our model has a PC synaptically connected with two stellate cells (SCs) (Supplementary Figs. 14a and 15). Both the SCs and the PC receive PF afferents. The odorant inputs increase the PF firing rate. The superficial SC inhibits the deep SC, and both SCs send inhibitory connections to the PC[40]. The dendrites of PCs receive excitatory CF inputs and the SCs receive excitatory CF input through glutamate spillover. The CF inputs convey information about water reward. As a consequence, there is activation of the CF input after the presentation of the S+ odorant, but not during S− trials. Analogous to modeling of eye blink conditioning[24], we assumed that a pause in PC firing causes an increase in the LR that we modeled by convolution of the PC spikes with a reversed Gaussian function (see "Methods" for details on the model). Simulation of S+ trials showed that odorant stimulus though PF inputs increases the activity of SCs that inhibit PC firing eliciting

an increase in the LR (Supplementary Fig. 14b, Saline S+). For the S− trials, we assumed the occurrence of a strong LTD at the PF–SC synapses and at the SC–PC synapses[41,42]. In this way, there is a reduction in the SC inhibition and an increase in PC excitation, yielding to a decrease in the LR (Supplementary Fig. 14b, Saline S−).

To model the effect of inhibitory chemogenetics (Fig. 7 and Supplementary Fig. 13), we considered the effects of a reduction of 40% in the activity of the SC population. To model learning of the S+ odorant reward in the hM4Di + CNO condition, we reduced by 40% the synaptic weight of the synapses between PF–SC and SC–PC. This partial reduction still allowed the SCs to inhibit the PC and thus maintain a high LR (Supplementary Fig. 14b, CNO S+). Regarding the effect of learning on lick behavior for S− trials during the treatment hM4Di + CNO, we deemed that a 40% reduction in the activity of the SC population induced a diminished occurrence of LTD between the PF–SC and SC–PC synapses. The diminished LTD decreased the inhibition strength of SCs to pause the PC firing, which impaired the typical reduction of the LR during the S− condition (Supplementary Fig. 14b, CNO S−). In summary, for the control condition we found an increase in lick strength for the S+ odorant that diverged from lick strength for S−. In contrast, for hM4Di + CNO lick strength increased slightly for both S+ and S−. A GLM analysis indicated that there were significant differences for lick strength for S+ vs S− ($p < 0.01$) and CNO ($p < 0.05$) and for the interactions between S+ vs. S− and CNO ($p < 0.01$, 88 observations, 80 d.f., GLM F-statistic 7.9, $p < 0.001$).

The model is simple. For example, it does not include basket cells. Furthermore, we did not model the large increase we find in the LR when the mouse receives the sugar water reward. In addition, we did not perform an exhaustive study of how the variables affect the changes in lick strength. Therefore, other explanations should be explored in future studies with alternate computational models and an exhaustive search of the input variables. Regardless, our model provides plausible mechanisms for the results that can be tested in future experiments with slice electrophysiology and awake behaving recording.

## Discussion

We found that vermal MLIs developed a differential response to odorants in the go–no go task that switched when the valence was reversed. Decoding analysis revealed that when the animal was proficient the contextual identity of the odorant could be decoded from MLI responses. GLM analysis revealed that contextual identity made a large contribution to the fit of MLI activity during the odorant application period. Chemogenetic inhibition of MLIs impaired achievement of proficient discrimination of odorants. These data indicate that MLIs play a role in associative learning by encoding valence.

The cerebellum has been implicated in mediating supervised learning through an iterative process whereby the response to an input is evaluated against a desired outcome, and errors are used to adjust adaptive elements within the system[8–10,43]. CFs carrying error signals make profuse synaptic connections on the dendrites of PCs and elicit powerful excitatory dendritic $Ca^{2+}$ spikelets[44–47]. Furthermore, CFs also signal reward prediction[11,12] or decision-making errors[13], and the cerebellum modulates association pathways in VTA enabling a cerebellar contribution to reward-based learning and social behavior[14]. The increase in $Ca^{2+}$ mediates LTD in subsets of synapses innervated by co-activated GC PFs carrying sensorimotor information relevant to learning[48,49]. However, recent studies by Rowan et al.[25] revealed that increasing feedforward inhibition by MLIs can switch the valence of plasticity from LTD to LTP (also, see ref. [50]). In addition, adaptive changes

in the vestibulo-ocular reflex elicited by CF optogenetic activation switched from increase to decrease depending on whether MLIs were co-activated[25]. Finally, MLIs gate supralinear CF-evoked $Ca^{2+}$ signaling in the PC dendrite[51]. These studies suggest that the valence of learning is graded by MLI activity.

Here we provide evidence for the involvement of MLIs in conveying information on contextual identity of a stimulus in associative learning. We do not find that the MLIs respond to odorants per se. Rather, the reversal experiment (Fig. 4) and the similar $\Delta F/F$ responses and stimulus decoding for correct and incorrect behavioral response trials (Supplementary Fig. 9) indicate that MLIs respond to contextual odorant identity: is this the rewarded odorant?, which is directly related to valence, a binary measure of an emotion reflected by the motivation to receive reward[36]. Furthermore, our results in the go–go task where both odorants are rewarded with varying volumes of sugar water (Supplementary Fig. 12) and the lack of a correlation between $\Delta F/F$ and the LR (Supplementary Fig. 12d) are consistent with the response reflecting valence (as opposed to value). Thus, we postulate that the MLI response during the odorant period is related to valence that reflects the sign of reward expectation (positive or negative), consistent with the fact that GCs in lobule VI were found to respond to reward expectation[29,30]. However, future experiments are necessary to fully disentangle whether MLIs encode for value vs. reward expectation.

A question that arises is which circuit mechanism is responsible for the decreased behavioral performance after chemogenetic inhibition of MLI activity (Fig. 7 and Supplementary Fig. 13). MLIs receive sensorimotor information from multiple GCs through PF input and in vivo studies have found remodeling of MLI receptive fields upon repeated electrical stimulation of the skin[52]. In addition, plasticity in PF–MLI synapses are postulated to increase the information capacity of the MLI–PC network and richness of PC output dynamics[10,16], and a model of PF–MLI plasticity has been proposed[41]. If long-term plasticity of PF–MLI synapses is responsible for the large change of MLI responsiveness found here upon reversal of stimulus valence, it is likely that the error signal would be provided by CF spillover resulting in highly redundant stimulation of SCs[40,53]. We developed a model of the MLI/PC circuit (Supplementary Figs. 14 and 15) that suggests that plasticity in SC–PC synapses[42] and in PF–SC synapses[41], complemented with CF spillover acting on the feedforward disinhibitory MLI circuit described recently by Arlt and Hausser[40], would explain the changes in behavior we find after chemogenetic inhibition of MLI activity. Finally, we found that the divergence between the time courses for lick frequency between S+ and S− took place at a later time when MLIs were inhibited by chemogenetics (Fig. 7e, Supplementary Fig. 13f) likely reflecting slow decision-making, consistent with a role for the CF/PC circuit in reward timing prediction[54]. Future studies are necessary to understand the role of plasticity in the PF-MLI-PC circuit in associative learning.

Interestingly, odorant responses have been reported in the cerebellum. Studies in sexually trained male rats found that female bedding or almond smell elicited increased cFos immunoreactive GCs in the vermis compared to rats exposed to clean air[55]. Furthermore, sexual experience increased the number of cFos positive GCs. In humans odorants induced significant activation of the cerebellum[56]. In addition, human cerebellar lesions caused olfactory impairments in the contralesional nostril, and elicited sniffs with lower overall airflow velocity compared to controls[57]. These findings implicate an olfactocerebellar pathway prominent in odorant identification and detection that functionally connects each nostril primarily to the contralateral

cerebellum. Our study provides further evidence for involvement of the cerebellum in olfactory tasks.

Here we found changes in MLI activity during learning in the go–no go associative learning task in lobule VI where GCs[29,30] and CF[58] activity was proposed to encode aspects of reward signaling. Recent work on the contribution of cerebellar processing to execution of reward-driven behaviors indicates that multiple cerebellar regions are involved, including central and lateral cerebellum[11,29,30,58,59]. As behavioral tests are refined, it is likely that differences in how these regions process reward will emerge, as suggested by the recent work on climbing-fiber signaling[11]. Furthermore, we focused our recordings on the more superficial regions of the molecular layer and therefore, most of our measurements are from SCs. Recordings from synaptically connected pairs of MLIs and PCs in slices showed that mean the amplitude of synaptic currents decreases with distance from the PC layer, suggesting a stronger impact of basket versus SC inhibition on PC firing[60]. Recent recordings of PC spikes in vivo following genetic deletion of MLIs confirm this prediction[50] which is in accord with the morphological diversity of MLIs[61]. Finally, the effect of chemogenetics (Fig. 7) should be on both stellate and basket cells, and future experiments are necessary to differentiate between the roles of the two cell types.

Our findings show that differential MLI activity develops during learning in an associative learning task and that MLI activity switches when the rewarded odorant is reversed. We find that inhibition of MLI activity elicits decreased behavioral performance in the go–no go task. Our data indicate that MLIs have a role in learning valence. This would likely increase the information capacity of the MLI-PC network and richness of PC output dynamics[10,62].

## Methods

**Animals**. All animal procedures were performed in accordance with protocols approved by the Institutional Animal Care and Use Committee of the University of Colorado Anschutz Medical Campus. Mice were bred in the animal facility. We used both male and female adult Parvalbumin-Cre (PV-Cre, Stock number 008069, Jackson Laboratory, USA) mice and wild-type C57BL/6J mice. The animals were housed in a vivarium with a 14/10 h light/dark cycle. Food was available ad libitum. Access to water was restricted in for the behavioral training sessions according to approved protocols, all mice were weighed daily and received sufficient water during behavioral training to maintain ≥80% of original body weight. Animal are housed at 72 ± 2 °F and a humidity of 40 ± 10%.

**Immunohistochemistry**. To perform immunostaining, mice were sacrificed and transcardially perfused with ice cold 4% paraformaldehyde (Electron Microscopy Sciences, USA), followed by incubation in 30% sucrose (Sigma-Aldrich, USA). After the brain was incubated in the sucrose solution, 60-μm-thick slices were cut with a cryostat. The slices were imaged using a confocal laser scanning microscope (Leica TCS SP5II, Germany or Nikon A1R, Japan) to determine the GCaMP expression patterns in the cerebellum. The slices were counterstained with DAPI (Thermo Fisher Scientific, USA).

**Window implantation**. Adult mice (8 weeks or older) were first exposed to isoflurane (2.5%) and then maintained anesthetized by intraperitoneal ketamine–xylazine injection (100 and 20 μg g$^{-1}$). A craniotomy was made over the vermis of cerebellum centered at midline 6.8 mm posterior to Bregma leaving the dura intact (lobule VI). A square glass window (2 mm × 2 mm) of No. 1 cover glass (0.13–0.17 mm thick, Thermo Fisher Scientific, USA) was placed over the craniotomy and the edges were sealed with cyanoacrylate glue (3M, USA). The window was further secured with Metabond (Parkell, USA), and a custom-made steel head bracket was glued to the skull.

**Virus expression of GCaMP**. In order to express GCaMP6f in cerebellar MLIs, we injected the vermis of the cerebellum in three adult PV-Cre mice with 2.0 μl of AAV1-Syn-Flex-GCaMP6f[53] (Addgene, USA) (6.8 mm posterior to Bregma, bilaterally ±0.5 mm lateral to midline and 200–400 μm below the brain surface). The viral infection method we use has been reported to express GCaMP in MLIs, and not in PCs[64,65]. Supplementary Fig. 2 shows that indeed expression of GCaMP6 takes place only in MLIs. Note the absence of fluorescence from GCs as well as from PC somata and dendrites. Expression in MLIs may reflect the differential activity of the hSyn promoter, and has been described for a different

genetically encoded Ca2+ indicator[65]. In one C57BL/6 animal each, we used AAV5-Syn-GCaMP6s or AAVrg-Syn-jGCaMP7f (Addgene, USA) with similar results to those obtained with GCaMP6f (Supplementary Fig. 16a) and therefore the data were pooled. After the injection, the animals were maintained for at least 3 weeks before behavior training and imaging.

**Go–no go training**. Mice were water deprived by restricting daily consumption of water to 1–1.5 ml. Mice were monitored for signs of dehydration or a decrease in body weight below 80% of the initial weight. If either condition occurred, the animals received water ad-lib until they recovered. When the animals were water-deprived, they were trained in a head-fixed olfactory go–no go task with 1% Iso vs MO odorant application (Sigma-Aldrich, USA)[26,66]. Licks were monitored by an electrical circuit monitoring the resistance between the lick spout and recorded with Intan RHD2000 software (Intan, USA). The floor in an olfactometer that controlled valves to deliver a 1:40 dilution of odorant at a rate of 2 L min$^{-1}$. The LR was calculated from the lick records and the time course was convolved with a 2 s Gaussian for the experiments where we performed multiphoton calcium imaging. We did not convolve the LR records for the experiments with chemogenetics. The water-deprived mice started the trial by licking on the water port. The odorant was delivered after a random time interval ranging from 1 to 1.5 s. In S+ trials, the mice needed to lick at least once in two 2 s lick segments to obtain a reward (0.1 g ml$^{-1}$ sucrose water) (Fig. 1a). The inter-trial interval (ITI) was 22.3–22.8 s. In S− trials, the mice need to refrain licking one of the two 2 s segments to avoid a longer ITI (22.3–22.8 + 10 s). The animal's behavior performance was evaluated in a sliding window of 20 trials and the calculated value was assigned to the last trial in the window. Therefore, it estimated the performance in the last 20 trials. The percent correct value represents the percent of trials in which the animal successfully performed appropriate actions, and we considered the animal proficient if percent correct performance is above 80%. In reverse go–no go training sessions, the rewarded and unrewarded odorants were switched.

Movement of the mouse was imaged in the infrared to prevent light interference with the non-descanned detection in the visible using a 1 Megapixel NIR security camera (ELP-USB100W05MT-DL36, Amazon.com, USA) at 30 frames/s. Velocity of body movement was estimated using the Farneback algorithm coded in Matlab[67] (Mathworks, USA). Measurement of body movement with a single camera gives limited information. Five mice were used for the go–no go experiments. Four mice were imaged when they were naive. The window became opaque for two of the mice preventing MLI imaging for the reversal experiment.

**Go–go training**. For the experiment in Supplementary Fig. 12, both odorants were rewarded with the same volume of sugar water, and the volume of sugar water reward was varied. The two odorants in this experiment were either 1% Iso and MO, or the same odorant 1% Iso and 1% Iso. The results were similar for both odorant pairs, and the analysis was performed for all trials regardless of odorant pair.

**Behavioral performance recorded after chemogenetic inhibition of MLI activity**. For chemogenetic inhibition of MLI activity[38,39], 1.2 μl of AAV8-hSyn-DIO-hM4D(Gi)-mCherry virus (Addgene, USA) was bilaterally injected into six PV-Cre animals at ±0.5 mm lateral to midline, 6.8 mm posterior to Bregma and 200–400 μm below the brain surface. For control, an AAV8-hSyn-DIO-mCherry virus (Addgene, USA) was injected in the same position in 6 PV-Cre animals. This viral infection method results in expression of the protein in MLIs, and not in PCs[64,65]. We performed two separate experiments: (1) For the experiments in Fig. 7, four weeks after injection the animals were trained to proficiency in the go–no go task with 1% Iso (S+) vs MO (S−). When they reached a proficient level, the animals rested for 1 to 2 weeks, and were then injected IP with saline 40 min before starting the session and were trained to discriminate between two odorant mixtures: 0.1% of 60% heptanal + 40% ethyl butyrate (S+) (Sigma-Aldrich, USA) and 0.1% of 40% heptanal + 60% ethyl butyrate (S−) for 3–4 sessions for a total of 400–500 trials. The animals then rested for 1 to 2 weeks and were subsequently trained to discriminate the same odorant mixtures in sessions that took place 40 min after IP injection of 3 mg kg$^{-1}$ CNO (Tocris, USA). (2) For the experiment in Supplementary Fig. 13, the animals were naive for the discrimination of Iso vs. MO. The animals were injected IP with 3 mg kg$^{-1}$ CNO (Tocris, USA) 40 min before starting the session and were trained to discriminate between 1% Iso (S+) and MO (S−) for 6 sessions for a total of 500–600 trials. The reinforcement was then reversed and the animals were again injected with CNO 40 min before the sessions and were trained to discriminate between MO (S+) and 1% Iso (S−) for 5 sessions for a total of 400–500 trials.

**Two-photon imaging of MLI activity in animals undergoing the Go–no go task**. All the animals were first habituated to the setup to minimize stress during the imaging experiments. All the imaging sessions started at least 10 min after mice had been head fixed. We searched for active MLIs while imaging zones in the vermis of lobule VI, between the midline and the paravermal vein, an area of the cerebellum where GCs acquire a predictive feedback signal or expectation reward[29,30]. The head-fixed two-photon imaging system consisted of a movable objective microscope (MOM, Sutter Instrument Company, USA) paired with a

80 MHz, ~100 fs laser (Mai-Tai HP DeepSee, Spectra Physics, USA) centered at 920 nm. The MOM was fitted with a single photon epifluorescence eGFP filter path (475 nm excitation/500–550 nm emission) used for initial field targeting followed by switching to the two-photon laser scanning path for imaging GCaMP at the depth of the MLIs. The galvometric laser scanning system was driven by SlideBook 6.0 (Intelligent Imaging Innovations, USA). The two-photon time lapses were acquired at $256 \times 256$ pixels using a 1.0 NA/20x water emersion objective (Zeiss, Germany) at 5.3 Hz. On the day of initial imaging, a FOV was selected to image a large number of active cerebellar neurons located in the most superficial planes of the molecular layer (within 50 μm beyond the dorsal surface of the molecular layer) including mostly SCs, and several batches of 6000 frames (a time series) were collected in each training session. After two-photon imaging a second image of the vasculature was captured with wide field epifluorescence to reconfirm the field.

**Data analysis.** Raw imaging data were first surveyed in ImageJ 1.52 (NIH, USA) to exclude image sequences exhibiting axial movement. We did not find evidence of axial movement while the animal was engaged in the go–no go task. In addition, we performed control imaging where we excited GCaMP6f at 820 nm, a two-photon excitation wavelength where fluorescence emission is $Ca^{2+}$-independent[68]. Supplementary Fig. 16b, c shows that we did not detect transient changes in GCaMP6 fluorescence in a mouse engaged in the go–no go task when the cells were excited at 820 nm. If we found horizontal drift due to motion, we applied cross correlation-based image alignment using the Turboreg, Image J plugin. The data were then analyzed with CaImAn Matlab software that uses constrained nonnegative matrix factorization to define independent spatial and temporal components corresponding to changes in GCaMP fluorescence in individual MLIs[31]. CaImAn identifies different spatial components (addressed here as ROIs) and a component representing the background and neuropil signals. Baseline of intensity ($F0$) was defined as the mean fluorescence intensity before trial start, defined as the time when the animal first licked. This was when fluorescence started to increase above baseline and the odorant was added at a random time 1–1.5 s after trial start. Intensity traces ($F$) were normalized according to the formula $\Delta F/F = (F - F0)/F0$. After CaImAn analysis, the $\Delta F/F$ traces of the spatial components were sorted and we assigned trial traces to different behavioral events (Hit, CR, Miss and CR) and aligned them to trial start, odorant onset or water delivery. Finally, the time course for the average $\Delta F/F$ did not differ greatly between the different GCaMP variants as would be expected for a fast firing interneuron with small increases in $Ca^{2+}$ per action potential (Supplementary Fig. 16a).

**Statistical analysis.** Statistical analysis was performed in Matlab 9.6 (Mathworks, USA). Statistical significance in measured parameters for factors such as learning and odorant identity (S+ vs. S−) was estimated using a GLM, with post-hoc tests for all data pairs corrected for multiple comparisons using FDR[33]. The post-hoc comparisons between pairs of data were performed either with a two-sided $t$ test, or a ranksum test, depending on the result of an Anderson–Darling test of normality. 95% CIs shown in the figures as vertical black lines or shading bounding the lines were estimated by bootstrap analysis of the mean by sampling with replacement 1000 times using the bootci function in MATLAB.

PCA was calculated using the Matlab Statistics Toolbox. Classification of trials using $\Delta F/F$ measured from all components in the FOV was accomplished via LDA in Matlab. $\Delta F/F$ for all components for every trial except one were used to train the LDA, and the missing trial was classified by its fit into the pre-existing dataset. This was repeated for all trials and was performed separately for analysis where the identity of the odorants was shuffled. Fluorescence intensity traces, LRs, and kinematics in Fig. 6 were low-pass filtered with a hamming window of a time constant of 0.59 s.

**Dimensionality.** Following Litwin-Kumar et al.[69], we defined the dimension of the system (dim) with $M$ inputs as the square of the sum of the eigenvalues of the covariance matrix of the measured $\Delta F/F$ for all ROIs in the FOV divided by the sum of each eigenvalue squared:

$$\dim\left(\frac{\Delta F}{F}\right) = \left(\sum_{i=1}^{M} \lambda_i\right)^2 \left(\sum_{i=1}^{M} \lambda_i^2\right)^{-1}, \quad (1)$$

where $\lambda_i$ are the eigenvalues of the covariance matrix of $\Delta F/F$ computed over the distribution of $\Delta F/F$ signals measured in the FOV. If the components of $\Delta F/F$ are independent and have the same variance, all the eigenvalues are equal and dim ($\Delta F/F$) = $M$. Conversely, if the $\Delta F/F$ components are correlated so that the data points are distributed equally in each dimension of an $m$-dimensional subspace of the full $M$-dimensional space, only $m$ eigenvalues will be nonzero and dim ($\Delta F/F$) = $m$.

**GLM estimate of contribution of different variables to changes in $\Delta F/F$.** In order to quantify the contribution of different variables to neural activity, we used GLM as described by Engelhard et al.[37]. We used the Matlab fitglm function to fit the per trial $\Delta F/F$ time course for mice proficient in the go–no go task with a GLM. We included event variables, whole trial variables and continuous variables. Continuous variables quantified kinematics including the LR, the derivative of the

LR and the velocity and acceleration of movements made by base of the tail of the head-fixed animal during the trial (body velocity and body acceleration, see Supplementary Fig. 3). The identity of the odorant (S+/S− odorant) was an event variable that increased from zero to one during the time for S+ or S− odorant application. Finally, whole trial variables were accuracy (1 for correct responses and 0 for incorrect responses), reinforcement history (1 for reinforcement in the last trial, 0 otherwise) and percent correct behavior calculated in a window of 20 trials.

**Model of Purkinje cell and MLIs.** For stellate cell simulation we used reconstructed mouse SC morphology available in Neuromorpho (http://neuromorpho.org/neuron_info.jsp?neuron_name = GlyT2_030_Slice3_Stellate_cell).

The morphology file was visualized using Blender 2.78 with the addon NeuroMorphoVis 1.4.0[70]: https://github.com/BlueBrain/NeuroMorphoVis.

We removed the axons from the original swc morphology file and exported the reconstructed morphology into a NEURON 7.5 hoc file (https://www.neuron.yale.edu/neuron/) using NLMorphologyViewer 0.3.0 (http://www.neuronland.org). We proceeded to create the electrical compartmental model with passive and active properties of the SC membrane. The passive parameters of the SC model were adapted mainly from Molineux et al.[71]. We set the specific membrane resistivity $R_m = 20$ kΩ cm$^2$, the specific membrane capacitance $C_m = 1.5$ μF cm$^{-2}$ [71], and the intracellular resistivity Ri = 115 Ω cm[72]. The input resistance Rin = 571.39 MΩ and membrane time constant $\tau_m = 40.30$ ms were obtained injecting a hyperpolarizing current into the soma (−1 pA, 500 ms). The time constant was obtained by a double exponential fit of membrane voltage decay. Those values are within the range of experimental values measured in SCs[71,73].

For modeling the active properties, we included voltage-dependent mechanisms for modeling the ionic channels at the soma of SCs. The firing patterns of SCs are regulated by fast sodium currents (Na), delayed rectifier potassium currents (KDR), A-type potassium currents (KA) and transient calcium currents (CaT)[71]. Since SCs and Golgi cells have similar firing properties[73], we adapted the voltage-dependent schemes of the conductances of Na, KDR, KA and CaT from a previous Golgi cell model[74] to reproduce the typical firing pattern of SCs[71] (Supplementary Fig. 15a).

For Purkinje cell simulation we adapted a previous two-compartment model that reproduces the typical spikes of PCs and is computationally efficient for constructing the cerebellum circuit model[75] (Supplementary Fig. 15b). The model was stimulated with background inhibitory inputs to present the typical curve of frequency versus current input from PCs[76].

For synaptic input simulation we used double exponential conductances to represent the synaptic inputs of the model with parameters taken from the literature.

PF–SC AMPARs[32]: $I_{max} = 96.42$ pA, $\bar{G} = 1.3774$ nS, $\tau_1 = 3.45$ ms, $\tau_2 = 3.17$ ms, $E_{rev} = 0$ mV.

PF–PC AMPARs[77]: $I_{max} = 20$ pA, $\bar{G} = 0.2857$ nS, $\tau_1 = 0.28$ ms, $\tau_2 = 1.23$ ms, $E_{rev} = 0$ mV.

SC–SC GABAaR[78,79]: $I_{max} = 75.5$ pA, $\bar{G} = 1.0786$ nS, $\tau_1 = 0.6$ ms, $\tau_2 = 5.9$ ms to 11.3 ms, $E_{rev} = −60$ mV

SC–PC GABAaR[80]: $\bar{G} = 15$ nS, $\tau_1 = 1.8$ ms, $\tau_2 = 8.5$ ms, $E_{rev} = −85$ mV, Delay = 2 ms.

CF–PC. The CF inputs were modeled as a strong activation of the PC AMPARs.

CF–SC. We assume that glutamate spillover from CF to SC has a delay of ~10 ms caused by glutamate diffusion[81] to stimulate the SC AMPARs. Based on the inferior olive neuron bursting patterns in response to CF inputs[82], we assumed a CF stimulus of 2 spikes at 300 Hz with an interval of 3.3 ms.

Synaptic plasticity was modeled by an increase or decrease of the maximum conductance of the synaptic channels. For modeling learning of S+ and S− tasks, we considered both the occurrence of LTD in SC-PC synapses[42], and the existence of LTD between the PF–SC synapses[41].

All simulations were performed on the NEURON 7.5 simulator[83]. The model is deposited in ModelDB (http://modeldb.yale.edu/266578).

**Reporting summary.** Further information on research design is available in the Nature Research Reporting Summary linked to this article.

## Data availability

The data supporting the findings of this study are available in GigaDB (https://doi.org/10.5524/100724)[84]. The model is deposited in ModelDB (http://modeldb.yale.edu/266578) and the data analysis code is available in https://github.com/restrepd/CaImAnDR.

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

## Acknowledgements

We thank Ms. Nicole Arevalo for animal husbandry, Ms. Dnate' Baxter for laboratory support, Ms. Arianna Gentile-Polese for help with the summary figure and Mr. Jesse Gilmer with help coding dimensionality. We thank Mr. Matt I. Becker, Mr. Jesse Gilmer and Dr. Abigail Person for discussions. This research was supported by NIDCD R01 DC000566 (D.R.), NINDS U01 NS099577 (D.R. and E.G.), NSF CBET-1631912 (E.G. and D.R.), NSF BIO-1926676 (D.R. and E.G.) and ANR-18-CE16-0010-01 (I.Ll. and D.R.).

## Author contributions

M.M., I.L. and D.R. designed all experiments aided with discussions with E.A.G., B.N.O., F.M.S. and G.L.F., while M.M. G.L.F. and D.R. performed them. E.A.G., G.L.F. and B.N.O. designed, setup and maintained the custom two-photon microscope, M.M. and D.R. set up the olfactometer. D.R. wrote all programs for data collection and analysis. F.M.S. performed circuit modeling, M.M. and D.R. analyzed the data. M.M. and D.R. generated the figures. All authors discussed the results. M.M. and D.R. wrote the manuscript, and all authors participated in commentary and revision of the manuscript.

## Competing interests

The authors declare no competing interests.
