## [Peer Review File · Nature Communications]

Reviewers' Comments:

Reviewer #1:

Remarks to the Author:

The study by Ma and colleagues investigates whether reward valence is encoded by molecular layer interneurons in lobule VI of the cerebellar cortex during an odour-based associative learning task. The authors provide compelling evidence to show that trial outcome can be decoded from neural activity changes during S+ vs S- trials and that switching the task contingency reverses MLI calcium responses. They then apply generalized linear models to show that odorant / task outcome can account for most of the variance in the MLI activity suggesting MLIs encode aspects of trial type and outcome, which they term valence. The experimental question regarding the importance of the cerebellum for cognitive control of behaviour is timely and important and the data are of high quality, and as a whole the manuscript will be of interest to the neuroscience community. However, I have some comments and suggestions that when addressed might help to improve the manuscript and further support the main conclusions of the manuscript.

Main comments:

- 1) The Go/NoGo task structure employed by Ma et al., is asymmetric in that Go trials are rewarded and in NoGo trials successful completion avoids punishment. Given that Wagner et al. 2017 have shown that a proportion of granule cells encode reward expectation and in this study the reward period immediately follows odour presentation, to what extent does reward expectation contribute to the dF changes observed during the olfactory odour presentation phase? The asymmetric nature of the task would suggest that in Go trials there would be a contribution while in NoGo trials there would be no reward expectancy after learning. The reverse would be true when the task contingency is switched, as seen in Figure 4. Experimentally or analytically disentangling the contribution of valence encoding, reward expectation and movement will be important to assess to what extent reward valence is specifically encoded during associative learning.
- 2) In Figure 1 and Supp. Fig. 3 it is clear that MLI dF changes correlate with licking during the odour presentation and reward delivery phases and this is probably not surprising given that lobule VI likely encodes some aspects of movement. The contribution of orofacial / tongue movement to dF changes during odour delivery is addressed later in the manuscript using GLMs but for transparency the average lick rate across time should be added to Fig. 2 panel b, Fig. 3 panel c, Fig. 4 panels b & e, and Fig. 5 panel a. Is the average lick rate different across pre-odorant, odorant and reinforcement phases and if so to what extent does that correlate with dF changes? It's clear in Fig. 7 that the GLM model suggests lick rate is not a strong determinant of S+ vs S- but given the functional data shown in Figs. 1, 6 and Supp. Fig 3 and the fact that lobule VI likely encodes orofacial/tongue movement, this seems rather surprising. This should be explored and discussed further.
- 3) It is clear that licking / lick rate is encoded in Crus II of the cerebellum but to fully appreciate the contribution of licking to dF changes in lobule VI MLIs, supplementary information should be provided to show how dF changes relate to lick frequency in isolation. These data could be generated by dispensing different reward volumes (3ul, 5ul, 7ul) to naïve mice which will generate a distribution of lick rates. This will provide the reader with a clear understanding of lick-related dF changes (perhaps including movement preparation). An intriguing finding is that although lobule VI likely encodes movement, during misses (when licking is absent), dF changes resemble Hit trial dF changes and during false alarms (when the animal licks when it shouldn't) dF changes are minimal (Fig. 5a). This highlights the importance of evaluating licking-evoked dF changes separately from dF changes related to other aspects of the task structure. Are there instances where there is a clear dF change during

odour presentation but licking is delayed? This might help to disentangle lick-related activity from reward-based information processing. Overall, addressing these points will help strengthen the main conclusions of the manuscript.

4) The main conclusion of the study is that MLIs encode reward valence but the analysis period (i.e. odour delivery) will include encoding of movement (licking), sensory feedback, reward expectation and valence. The authors use a binary readout of valence, either rewarded trails or non-rewarded trails to compare the underlying dF. The GLM modelling data in Figure 7 is convincing but does this suggest that MLIs encode reward valence or the learned presence of a reward vs no reward, which is somewhat different. To show that valence – the value that an individual places on the rewards of successfully completing a task – is specifically encoded, one would have to show that peak dF correlates with changes in reward valence. This could be achieved by training animals in blocks with different reward sizes (i.e. block one 3ul reward, block two 5ul reward, block three 7ul reward. If MLIs encode reward valence one would expect the peak dF to correlate with reward size / valence. This should be explored experimentally to support the main conclusion of the paper.

5) What is not clear from the main text is why the task structure changes in Figure 8. In Figs. 1-7 the task requires the mouse to detect an odour vs detect a lack of odour, whereas in Fig. 8 a true odour discrimination task is used. Why change the task structure? Do the changes affect the main findings of Figs 1-7? This comparison should be made and documented in Supplementary Information to ensure that the results of Fig. 8 can be used to interpret the findings of Figs. 1-7 and discussed in the main text.

6) The results of the hm4Di experiments in Figure 8 are rather confusing. In ci, expression of mCherry and addition of CNO during S- trials has no effect, as expected, due to MLIs not being active (see Fig. 1 & 2) and CNO not having any off-target effects. But in cii, expression of hm4Di and application of CNO now generates an increase in lick frequency in S- trials. But if in S- trials MLIs are not active why does inhibiting them increase the lick frequency, when silencing an already inactive population of neurons will have no net effect on feedforward inhibition to Purkinje cells or output from the cerebellum. The suggested cellular or network mechanisms underpinning these changes should be discussed in more detail in the main text. These findings are then used to support the conclusion that errors during the S- trials underpins the reduction in task success. It would also be helpful if the changes in Hits, misses, CR and FAs were shown to clearly illustrate which type of accumulated error leads to an overall reduction in task success.

Minor comments:

- 1) Figure 1 – include a description of the proportion of MLIs that are task active vs non-responsive in the main text.
- 2) Supp. Video 1 – the unfortunate camera angle makes it very difficult to observe mouse behaviour which involves orofacial movements and engagement with reward delivery apparatus that is presented to the mouth. The authors should instead image the behaviour from a front elevation.
- 3) Supp. Fig. 3 – reference is made to the lick 1-1.5s before odorant delivery onset. Why does this not trigger trial onset and odour delivery? Is there a pseudo randomised delay period from one trial to another or is the 'pre-odour' lick equivalent to the mouse counting the passage of time? This is not clear from the text, methods section or Supplementary figures.
- 4) Line 26 – 'restrain' should read 'refrain'?
- 5) Line 31 – 'licks diverged more slowly' is rather confusing for the reader.

- 6) Line 60 – to test sufficiency (which is rather difficult to achieve at the systems level) one would need to show that driving MLI activity alone is 'sufficient' to drive olfactory discrimination, this is not what is shown in the manuscript. Instead, the results appear to show that MLI activity is necessary for the task.
- 7) Line 81 – 'proficient discriminating' – reword.
- 8) Line 91 – the consistent, albeit small' dF change during CR trials should be discussed in the main text. Presumably this reflects the small amount of licking that occurs in CR trials (see Supp. Fig. 3b).
- 9) L221 – Add lick data regarding the average lick rate across all mice during Hit, CR, FA & miss trials.
- 10) L259/260 – the data suggest there is a correlation between licking and dF during odour presentation so the conclusion is a little overstated. The experiments suggested above will shed light on lick-related dF changes during the task.
- 11) L231 – the use of the term 'slower' has not been qualified. What is meant by slower?
- 12) L378 – the term 'slowed decision-making' is not supported by the data shown.
- 13) Fig. 1 / methods – the reason why mice consistently lick 1-1.5s before the lick that initiates each trial should be discussed in more detail. What in the task structure drives mice to show this behaviour?
- 14) Fig. 1. – the consistent increase in dF at the beginning of CR trials should be discussed in the text.
- 15) Fig. 2 panel e – requires labels to show 'pre-odorant' and 'odorant'.
- 16) Fig. 4 panel f – the decoding doesn't predict the behaviour it reflects licking vs withholding licking, change axis label and in main text.
- 17) L685 – label should read (c) not (d).
- 18) Supp. Fig. 3 panel a – from this elevation it is difficult to understand task structure or mouse engagement in the task.
- 19) Supp. Fig. 3 panel b – add details of what the two red lines represent, this is missing in the legend.

Reviewer #2:

Remarks to the Author:

The manuscript by Ma and colleagues provides imaging data that adds to the growing body of evidence for the role of cerebellar molecular layer interneurons (MLIs) in learning. Further, this is of interest to the more general neuroscience audience as the role of the interneurons is not very well understood.

The two major claims of the paper are:

1. MLIs encode the valence of the stimulus and they acquire this over time.
2. MLIs are necessary to reach proficiency in this associative learning task.

The first claim is justified, although there are quite a lot of methodological and statistical issues that need to be addressed to support this conclusion. The second claim requires additional experimental data. As of now the evidence presented by the authors does not give enough support to conclusion #2.

I have several suggestions that could improve the manuscript and strengthen the conclusions presented by the authors. However, before going into the details there are two major problems that severely impair the readability of the manuscript and need to be addressed in the revision.

1. The colors used in the figures need to be redesigned. Currently same colors indicate different things within the same figure.

Examples:

Figure 2: color blue in panels b, e and f indicate S- odorant (not rewarded) and red S+ odorant (rewarded). But in panel g the same colors have a different meaning as the dimensionality analysis

takes under consideration both S- and S+ trials. Here (I assume) blue depicts low learning (trials? sessions?) and red proficient (trials? sessions?)

Figure 3: color blue in panel b, indicate S- odorant (not rewarded) and color red S+ odorant (rewarded). Note: legend in that panel is missing. However, in panel d same colors do not mean the same thing. Just like in Figure 2 (I assume) blue depicts low learning (trials? sessions?) and red proficient (trials? sessions?)

Figure 7: possibly the worst color-coding of all figures. In panel b the four colors indicate Miss, CR, FA and Hit trials but in panel c THE SAME colors now depict different time epochs in trials. Colors in panels c and d are confusing and redundant.

Figure 8: The same colors that previously were used for Miss, CR, FA and Hit trials (and different time in trials in Figure 7) now stand for different drug conditions.

This list is not exhaustive. Almost every figure suffers from such poorly chosen color scheme. It is crucial to address this in the next submission. Moreover, the colors chosen for Miss and CR, and FA and Hit trials are very similar to each other making the data points in figure 2c, 2d and 4c very difficult to distinguish. Finally, panels 2a, 3a, 4a, 8a, 8b and supplementary figure 4 use colors very similar to the ones depicting trial types described above. Overall, the chosen color scheme makes the figures close to unreadable and hard to interpret.

2. Style of writing makes it difficult to follow the story. Instead of saying "Figure X shows...." authors should describe their findings and refer to the figure using brackets (Figure X). Describing each panel separately in a point-by-point manner is appropriate for a figure caption but does not suffice for the body of a manuscript.

Other major issues:

- Abstract: The abstract needs to be changed in order to resolve the following issues: a) Time-out is not really a "punishment". I would avoid using this terminology; b) Sentence in line 26/27 is confusing ("When the animal is naïve..."). Please do not use the brackets to explain terminology. This almost reads as a draft. Rather I'd suggest re-writing it. Perhaps something like "In naïve animals the MLI responses did not differ between the odorants. With learning, the rewarded odorant elicited a large increase in MLI calcium responses/transients/signals, and". Or something to that end. The last part of the abstract regarding chemogenetic inactivation of MLIs should be re-evaluated pending experimental input (see my comments below regarding Results).

- Results/Methods:

1. The number of animals used in each figure has to be explicitly stated, both in figure captions, results section and methods. When the n refers to sessions or trials, these should also be specified. Methods should explain why only 3 mice were reversed. Additionally, why do figure 2 and supplementary figure 4 only depict 4 mice while 5 mice were used according to methods section (lines 447-457), please explain.

2. Authors pool data obtained with GCaMP6f, GCaMP6s and GCaMP7f and justify this saying, in line 445, that "similar results were obtained" using those indicators. Given very different kinetics of these calcium indicators it is crucial for the authors to show the reader the comparisons, including statistics. This can be done in a supplementary figure.

3. What is the inter-trial interval? This should be mentioned in the methods section. If the ITI was fixed rather than random this could explain why the animals start licking and the calcium signal starts rising prior to the odorant exposure. If the ITI is indeed random is it possible that the animals are responding to the click or hiss of a valve?

4. Currently, correct rejection trials still contain lick responses in one segment (line 466-469 in Methods). Are there any trials where the animals show a full suppression of the lick response spanning both segments? If this is the case this should be analyzed as a different type of trial. This information should also be described in the Results section.

5. What bootstrapping technique was used to obtain Figure 2b? Explain in the results and methods.

6. The dimensionality calculation presented in Figure 2g should include a comparison between "learning" and "proficient" conditions. Currently, only within condition comparisons are indicated.
7. Results presented in Figure 5a seem to be in direct conflict with data presented in Figure 2c and 2d and Figure 4c. Specifically, the authors claim that as depicted in Figure 5a the calcium responses to Hits and Misses are virtually indistinguishable. However, the same type of trials in the other figures show much smaller calcium transients. For example, looking at Figure 4b middle bottom panel (trial rev91) the calcium response to a Miss trial is negligible and definitely different than a Hit response. Is the analysis presented in Figure 5 done on all trials or only at the trial of >80% accuracy. These issues have to be resolved in order to support the main conclusion of the paper. Moreover, the fact that the MLIs show similar responses after a reward is obtained (Hit trial) and withheld (Miss trial) should be further discussed in the Discussion section.
8. The decoding accuracy analysis presented in Figure 5 should also be applied to reversed trials.
9. Looking at data presented in Figure 6d it is unclear how the FA trials are classified. The lick responses for FA and Miss trials look almost the same. Is there a behavioral difference in the response? Please quantify and provide statistics.
10. The DREADD experiments do not support the conclusion that MLIs are necessary to learn the task as the chemogenetic inactivation is performed on trained animals. These experiments should be expanded by a) introducing CNO from the beginning of the training, or b) reversing the mice under CNO. Moreover, current results point to a motor impairment given a smaller lick response, rather than contextual learning. The authors state this themselves in lines 329-331. This conclusion is in stark contrast to the claims made in the abstract. Please provide supplementary figures with quantifications for the pattern of DREADD expression in the cerebella of all inactivated animals.
11. Behavioral analysis of body movement presented in Supplementary Figure 3 is insufficient. First, head-fixed mice cannot "lean" into the port. Second, Supplementary Movie 1 shows movements of different body parts (hind legs, front legs, tail) and overall shuffling of the body. I recommend using DeepLabCut, or similar, analysis to disentangle different movements.

- Introduction:

1. At some point the difference between stellate and basket cells should be mentioned, see Brown et al., 2019.
2. Authors should explicitly discuss why they chose to image Lobule VI over Crus I or simplex, which also show reward related signals. Please discuss papers by Kostadinov et al. 2019 and Heffley et al., 2018 and 2019.
3. When discussing plasticity in the cerebellar circuit the authors exclusively focus on LTD without mentioning LTP and intrinsic plasticity. This is a serious omission as it has been shown that LTD is not essential for motor learning, see Schonewille et al., 2011.

- Discussion:

1. Discussion should be expanded to address raised inconsistencies in the Results. However, it is advisable to rewrite the discussion pending suggested experiments and analyses.

Minor issues:

1. Figure panels 2e and 2f should have a title for quicker interpretation.
2. The text in lines 169-170 is inconsistent with the legend of figure 4b.
3. Figure 5b left bottom panel (fwd51) contains a mistake: MO(s+) should be MO(s-).
4. In figure 5, the legend states point d which should be c.
5. Why was LDA analysis chosen? Did authors try other classifiers? Some justification is needed (in Results or Methods).
6. Similar to Supplementary Figure 4, please show behavioral plots for all reversed mice.
7. Figure 6a is missing a color legend.
8. Figure 6b: the legend is swapped in the panel. Red should depict S+.

9. Figure 7c and 7d: what do individual dots refer to? Sessions? Trials? Animals?
10. Figure 7b: it would be helpful to provide goodness of fit analysis for data presented in this panel.
11. In line 463, please clarify what "were ready" stands for.
12. Reference number 5 is a review. It would be better to cite experimental papers, such as Tsai et al., 2013, Stoodley et al., 2017 or Badura et al., 2018.
13. When reporting statistics (example- line 110), include statistic value and error.
14. When figures are being described in the text, significances are often reported that do not have a corresponding star on the graph (example line 194). Consider adding these for easier interpretation of figures.
15. In the methods section, references to the used product/compounds should be added.

Reviewer #3:

Remarks to the Author:

In this manuscript, Ma et al. investigate how cerebellar molecular layer interneurons respond as a function of learning in an olfactory sensory discrimination task. The authors use in vivo calcium imaging to measure how these interneurons respond to odorants across learning, and conclude they develop preferential responses to whichever stimulus accurately predicts upcoming reward. Moreover, they find that responses are linked most strongly to the stimulus identity rather than the trial outcome, suggesting perhaps that responses reflect a context dependent stimulus representation rather than some form of reward prediction per se. Moreover, chemogenetic silencing of these interneurons impairs learning in this task, it seems by decreasing the animals' ability to restrict false alarms, implicating the cerebellum in a particular aspect of this associative learning task. Overall, these findings could be of considerable interest to the field of cerebellar research and those interested in associative learning, as recent work has begun to reveal new roles for the cerebellum in reward-based learning. However, I have several concerns related to the completeness of data presentation and analyses that limit my enthusiasm for the manuscript in its current form. These issues should be addressable if authors can enhance their analysis of the relationship between neural activity and behavior, and the presentation of neural data across cells and mice.

Major Concerns:

1) In general, the data are underrepresented by anecdotal examples rather than comprehensive depictions that span neurons and mice. To evaluate the veracity of claims that stem from high level dimensional reduction approaches and GLM output, it is necessary to see overall population responses and analyses for all the individual contributing ROIs. This should include, for example, mean DF/F responses across all contributing ROIs for each trial type (hits, misses, false alarms and correct rejects). Similarly, scatter plots or alternate depictions should be used to show variability across all the ROIs. As currently presented, it is not clear how many ROIs go into each experiment, let alone what the responses of all cells look like.

Along these lines, more detailed analyses of response types across neurons would be helpful. For example, is it the same MLIs that respond to Hits, misses, FAs and CRs? Or are these separate populations? What fraction of identified neurons in a FOV are responding on different trial types? Are the same neurons tracked across learning (this is not clear from the text)? This would be particularly useful for the reversal learning data. If the same neurons are tracked, is it the same or new cells that develop responses as a function of learning? Such information would be useful to understand potential contributions to learning (as is suggested by the chemogenetic manipulation)

2) The relationship between neural activity and behavior is under analyzed. Many of the changes in neural activity likely correlate with the changes in licking that occur across learning, but the authors only use the derivative of lick rate to test for a correlation with neural activity. Please show mean lick rates for all trial types for low and high proficiency learning conditions, and include these on the same timescale with the mean DF/F for all trial types across ROIs. Depending on the clarity with which such mean responses segregate behavior and neural activity, further analyses leveraging variability in licking across trials may be necessary (e.g. segregating within trial type based on the amount of licking). Without such information, it is difficult to interpret many of the results. For example, if licking best explains the neural activity in the 1 second after reward delivery as suggested by the GLM and in Figure 6, why is neural activity the same on average in this same period for Hits and Misses (Figure 5A)? One might conclude that licking is actually not different despite the categorization of these trials as 'misses', but without showing the licking is impossible to tell.

Related to this point, I am not confident in the criteria used to segregate trial types based on licking. Figure 6d, bottom shows very similar licking, specifically during the odorant period (between the black bars) for CR, FA and Miss trials. Based on these representative examples, the criteria for trial segregation requires more empirical justification.

3) The data show that DF/F begins increasing 1-1.5 seconds before odorant presentation (Figure 2, S+ trials, and S- trials also? Hard to tell for S- trials with the data behind the S+ trials). Is there some cue that indicates the start of the trial that is not described in the paper that lets the animal know when the odorant will be delivered? Perhaps the ITI is constant so that animals can predict the trial onset, and the animals know based on timing when the stimuli are delivered? Such task features are not described, but are essential for interpreting these results. If the task timing is predictable, it would be especially important to explicitly evaluate the role of such predictions in the neural data.

4) The interpretation of the results in this manuscript is limited. The discussion does not make clear whether and how these findings fit within a reward prediction framework, or specifically how they compare and contrast with what has been shown for climbing fibers and granule cells in other cerebellar reward-based learning tasks. What do these data tell us about reinforcement learning and the cerebellum?

Minor points

1) The curation of ROIs is not convincing. Figure 1B shows ROIs that vary much more widely in shape and size than MLIs. For example ROIs 43 and 79 are at least an order of magnitude larger than ROIs such as 3 and 4, and are also shaped much more irregularly than would be expected. This suggests that the Caiman algorithm may not be providing an accurate segmentation, and further data curation is necessary. Plots of ROI size distribution should be included, as well as justification for acceptance or rejection following automated segmentation. For example, for ROIs such as 43 and 79 that lie on the extremes of the pixels size distribution, what features of the response among grouped pixels indicate that these are single neurons?

2) Figure 1C shows the difference between Hits and CRs. This is the least informative comparison, as there is no licking in CRs. Please show all four conditions (including Hits, FAs, Misses and CRs).

3) Why are the first 20 trials in figures 2A and 3A pegged at the same, invariant percent correct? The methods indicate a sliding window of 20 trials, but shouldn't this window move forward for each trial (i.e. trials 1-20, then 2-21)? Perhaps I'm missing some feature of this analysis, but please clarify in the methods.

4) Figure 4, Panel B, bottom left (fwd 51) should be MO (S-)

5) Is the legend reversed in Figure 6B, or is this example animal licking more on S- trials?

6) It is confusing to use colors for figure 6E that overlap with the color coded trial types in 6A-D, but

do not actually reflect these trial types.

7) Are the DF/F traces in figure 6D averages across cells as indicated by the legend? If so, why no error bars here?

8) Line 464-5 indicates that an olfactometer was used to monitor licks. What does this mean? Was licking measured optically, with video, or something else?

9) Line 512: "were be collected"

10) Line 513: "was captured wide field epifluorescence"

11) Figure legends are often incomplete, missing details such as what asterisks indicate (Figure 5d), what dotted lines indicate (Figure 8a), etc. While it is possible to extrapolate in most cases, these details should be addressed. Please check the figure legends carefully, as well as for other errors (e.g. Figure 5 has no panel D).

Reply to reviewers

Reviewer #1 (Remarks to the Author):

The study by Ma and colleagues investigates whether reward valence is encoded by molecular layer interneurons in lobule VI of the cerebellar cortex during an odour-based associative learning task. The authors provide compelling evidence to show that trial outcome can be decoded from neural activity changes during S+ vs S- trials and that switching the task contingency reverses MLI calcium responses. They then apply generalized linear models to show that odour / task outcome can account for most of the variance in the MLI activity suggesting MLIs encode aspects of trial type and outcome, which they term valence. The experimental question regarding the importance of the cerebellum for cognitive control of behaviour is timely and important and the data are of high quality, and as a whole the manuscript will be of interest to the neuroscience community. However, I have some comments and suggestions that when addressed might help to improve the manuscript and further support the main conclusions of the manuscript.

Main comments:

We would like to thank the reviewer for the suggestions. Several of the comments below refer to our use of the term “valence”. We would like to preface our responses by clarifying what we mean by “valence” and “reward value”. As explained in an insightful review by Kay M. Tye¹ valence is whether a stimulus is good or bad. Thus, valence is a binary (or highly non-linear) measure of an emotion that is reflected by the motivation to receive reward (or avoid punishment). Thus as stated by Tye: “value may refer to any integer along a continuous spectrum, whereas valence refers only to the sign of that integer—working in a binary code.” Also, as discussed by Tye valence can be innate, or learned. Thus, learning valence is equivalent to learning that a stimulus is associated with reward. We now define what we mean by valence and reward value and we cite the Kay M. Tye review in lines 247 and 379.

1) The Go/NoGo task structure employed by Ma et al., is asymmetric in that Go trials are rewarded and in NoGo trials successful completion avoids punishment. Given that Wagner et al. 2017 have shown that a proportion of granule cells encode reward expectation and in this study the reward period immediately follows odour presentation, to what extent does reward expectation contribute to the dF changes observed during the olfactory odour presentation phase? The asymmetric nature of the task would suggest that in Go trials there would be a contribution while in NoGo trials there would be no reward expectancy after learning. The reverse would be true when the task contingency is switched, as seen in Figure 4. Experimentally or analytically disentangling the contribution of valence encoding, reward expectation and movement will be important to assess to what extent reward valence is specifically encoded during associative learning. We did not make clear the semantics of how we define the terms relevant to behavior in the go-no go task. The most important point is that we do not claim that during odour presentation the MLIs respond to the odourant per se. Rather, they respond to the valence represented by the contextual odourant identity: “is this the rewarded odourant?”. Yes, our data does show that there is a contribution of reward expectancy (equivalent to valence). Thus, our

conclusion that MLIs respond to valence is consistent with responses to reward expectation. We have added a paragraph explaining this in the discussion (line 373):

“Here we provide evidence for the involvement of MLIs in conveying information on contextual identity of a stimulus in associative learning. We do not find that the MLIs respond to odorants per se. Rather, the reversal experiment (Fig. 4) and the similar $\Delta F/F$ responses and stimulus decoding for correct and incorrect behavioral response trials (Supplementary Fig. 8) indicate that MLIs respond to contextual odorant identity: “is this the rewarded odorant?”, which is directly related to valence, a binary measure of an emotion reflected by the motivation to receive reward¹. Furthermore, our results in the go-go task where both odorants are rewarded with varying volumes of sugar water (Supplementary Fig. 12) are consistent with the response reflecting valence (as opposed to value). Thus, we postulate that the MLI response during the odorant period is related to valence that reflects reward expectation, consistent with the fact that GCs in lobe VI were found to respond to reward expectation^{2,3}. “

2) In Figure 1 and Supp. Fig. 3 it is clear that MLI dF changes correlate with licking during the odour presentation and reward delivery phases and this is probably not surprising given that lobule VI likely encodes some aspects of movement. The contribution of orofacial / tongue movement to dF changes during odour delivery is addressed later in the manuscript using GLMs but for transparency the average lick rate across time should be added to Fig. 2 panel b, Fig. 3 panel c, Fig. 4 panels b & e, and Fig. 5 panel a. Is the average lick rate different across pre-odorant, odorant and reinforcement phases and if so to what extent does that correlate with dF changes? It's clear in Fig. 7 that the GLM model suggests lick rate is not a strong determinant of S+ vs S- but given the functional data shown in Figs. 1, 6 and Supp. Fig 3 and the fact that lobule VI likely encodes orofacial/tongue movement, this seems rather surprising. This should be explored and discussed further. Yes, making a more thorough evaluation of the relationship between lick rate and dF/F makes sense. First, in the new Fig. 2b we show the licks corresponding to the dF/F time course. In addition, in the new Supplementary Fig. 5 we show the mean lick frequency time courses corresponding to Fig. 3 panel c and Fig. 4 panel e. Finally, the lick time course corresponding to what was Fig. 5a (now Supplementary Fig. 8b) are those shown in what is now Fig. 5b (and this is stated in the figure legend for Supplementary Fig. 8b).

These figures, and the additional example of lick frequency time courses already shown in Fig. 5b show that lick frequency in the odorant period tends to be low, sometimes decreases shortly after odorant addition, and increases drastically upon reinforcement. The low lick frequency during odorant stimulation is likely due to the fact that the rule for reinforcement is to lick at least once during each of the two 2 sec lick segments within the odorant application period (Fig. 1a). Thus, a lick frequency of 0.5 Hz would suffice. Because of this the animal can save on this dry lick effort by licking at a low frequency during the odorant application period, and increase the lick frequency when the sugar water is delivered.

Furthermore, we performed a new analysis of the correlation between $\Delta F/F$ and lick frequency during the different time periods (new Figs. 5e and g see lines 217 to 242 in the results). And, yes, as expected there is a correlation between these two variables during the odorant period, but this correlation is higher, and significantly different, during reinforcement (Fig. 5g). This is consistent with the fact that in the GLM analysis lick frequency is a stronger predictor of $\Delta F/F$

within the reinforcement period. As shown in Fig. 6 the contextual identity of the odorant contributes more strongly to the $\Delta F/F$ during the odorant period.

3) It is clear that licking / lick rate is encoded in Crus II of the cerebellum but to fully appreciate the contribution of licking to dF changes in lobule VI MLIs, supplementary information should be provided to show how dF changes relate to lick frequency in isolation. These data could be generated by dispensing different reward volumes (3ul, 5ul, 7ul) to naïve mice which will generate a distribution of lick rates. This will provide the reader with a clear understanding of lick-related dF changes (perhaps including movement preparation). The experiment suggested by the reviewer is relevant because Gaffield and Christie have performed a similar experiment in Crus II where MLIs respond to lick rate. Thus, Figure 3 of Gaffield and Christie⁴ shows that as the volume of water delivered is increased (increased reward value) lick rate increases and it is positively related to the change in fluorescence for GCaMP in Crus II MLIs. This is what is expected when dF/F reflects reward value (as opposed to valence, that would be expected to show no correlation between dF/F and reward volume).

We performed a go-go experiment where both stimuli are rewarded (Supplementary Figure 12, described in lines 244 to 267). As expected, during wet licking (defined as licking following sugar water reward) the frequency of licking increased linearly as a function of volume of reward (Supplementary Figure 12b). In contrast the frequency of licking did not change as a function of water reward during dry licking (when odorant was delivered, before reward delivery). When we compare dF/F vs. reward volume we do not find a positive correlation. These data are consistent with representation of valence (as opposed to reward value) by MLI activity in Lobule VI.

An intriguing finding is that although lobule VI likely encodes movement, during misses (when licking is absent), dF changes resemble Hit trial dF changes and during false alarms (when the animal licks when it shouldn't) dF changes are minimal (Fig. 5a). This highlights the importance of evaluating licking-evoked dF changes separately from dF changes related to other aspects of the task structure. Are there instances where there is a clear dF change during odour presentation but licking is delayed? This might help to disentangle lick-related activity from reward-based information processing. Overall, addressing these points will help strengthen the main conclusions of the manuscript. This is an important point. First, we present a more thorough analysis of licks and dF/F s in error trials, because, as the reviewer remarks, the fact that lick frequency and dF/F behave differently for error trials is relevant. Indeed, this is consistent with MLI activity reflecting valence, as opposed to licking per se (see Supplementary Note 3 and new analysis in Supplementary Fig. 8).

Second, with respect to the question whether there are instances where there is a clear dF change during odour presentation but licking is delayed, perhaps the best time period to address this question is at the beginning of the odorant period when the dF/F starts increasing, but does not differ between S+ and S- odorants. We include new Supplementary Fig. 11 where we analyzed the relationship of dF/F and lick frequency in this time period. In Crus II, where dF/F reflects lick rate (Gaffield and Christie, 2017 and Astorga et al. 2017) have shown that whenever there is a sharp increase in dF/F there is a corresponding increase in lick frequency. Supplementary Fig. 11 (Supplementary Note 4, Sup. lines 79 to 92) shows that this is not the case in our study. Together with the other experiments in the section that characterize thoroughly the dependence

of dF/F on lick frequency we find that, although there is a correlation between these two variables, the dependence differs depending on when it is evaluated (odorant period, shortly after odorant application, reward period). This indicates that for mice engaged in this go-no go task dF/F is not exclusively dependent on lick frequency and argues for the multivariate GLM approach that we used.

Also, see point 4 of reviewer 2 where we analyzed CR trials with no licks compared to CR trials with licks.

4) The main conclusion of the study is that MLIs encode reward valence but the analysis period (i.e. odour delivery) will include encoding of movement (licking), sensory feedback, reward expectation and valence. The authors use a binary readout of valence, either rewarded trials or non-rewarded trials to compare the underlying dF. The GLM modelling data in Figure 7 is convincing but does this suggest that MLIs encode reward valence or the learned presence of a reward vs no reward, which is somewhat different. To show that valence – the value that an individual places on the rewards of successfully completing a task – is specifically encoded, one would have to show that peak dF correlates with changes in reward valence. This could be achieved by training animals in blocks with different reward sizes (i.e. block one 3ul reward, block two 5ul reward, block three 7ul reward. If MLIs encode reward valence one would expect the peak dF to correlate with reward size / valence. This should be explored experimentally to support the main conclusion of the paper. This experiment is an interesting suggestion, and we did perform it. Please see the response to point 3 above where we describe the results of this experiment that yields data consistent with representation of valence (as opposed to reward value) by MLI activity. Also, please see the first response where we explain that our binary definition of valence is consistent with Kay M. Tye's widely accepted definition, and that a change in the volume of reward is a change in reward value, not valence, and that there is both innate *and learned* valence.

5) What is not clear from the main text is why the task structure changes in Figure 8. In Figs. 1-7 the task requires the mouse to detect an odour vs detect a lack of odour, whereas in Fig. 7 a true odour discrimination task is used. Why change the task structure? Do the changes affect the main findings of Figs 1-7? This comparison should be made and documented in Supplementary Information to ensure that the results of Fig. 7 can be used to interpret the findings of Figs. 1-7 and discussed in the main text. The reason why we did this is because we had decided to adopt the experimental design we use regularly to ensure detecting changes in olfactory discrimination behavior after circuit modification with drug treatment, optogenetics or chemogenetics. We first train the animals to discriminate between 1% Iso and MO (no CNO) so that they learn the go-no go task, and then we switch them to the discrimination experiment with the odorant mixtures (+/- CNO). The switch to a mixture odorants is used regularly in olfactory studies to make the discrimination task more difficult to ensure finding changes in behavior. This design is conservative in terms of obtaining reliably a behavioral result after circuit modification because it uses a difficult discrimination task.

However, the question raised by the reviewer is important: Is the result of Fig. 7 (previous Fig. 8) directly relevant to the results in Figs. 1-6? For the revised manuscript we performed a complementary experiment where CNO was applied before the first learning session with 1% Iso

and MO (Supplementary Fig. 13). We obtained similar results to the experiment with the odorant mixtures (Fig. 7). The revised chemogenetics experiments are described in lines 342 to 345 and the methods are updated in the revised manuscript (lines 557 to 564).

6) The results of the hM4Di experiments in Figure 8 are rather confusing. In ci, expression of mCherry and addition of CNO during S- trials has no effect, as expected, due to MLIs not being active (see Fig. 1 & 2) and CNO not having any off-target effects. But in cii, expression of hM4Di and application of CNO now generates an increase in lick frequency in S- trials. But if in S- trials MLIs are not active why does inhibiting them increase the lick frequency, when silencing an already inactive population of neurons will have no net effect on feedforward inhibition to Purkinje cells or output from the cerebellum. The suggested cellular or network mechanisms underpinning these changes should be discussed in more detail in the main text. These findings are then used to support the conclusion that errors during the S- trials underpins the reduction in task success. It would also be helpful if the changes in Hits, misses, CR and FAs were shown to clearly illustrate which type of accumulated error leads to an overall reduction in task success. The reviewer makes an interesting point. To explore the potential circuit basis of this chemogenetic effect we developed a simple computational model of the circuit. Because Arlt and Hausser⁵ published a manuscript showing that climbing fiber spillover stimulates with high probability superficial MLIs activating a feedforward *disinhibitory* circuit involving superficial MLIs inhibiting deep MLIs that control PC output we included two MLIs and one Purkinje cell in the model. In the model activity in MLIs inhibits PCs resulting in increased lick rates.

As discussed in Supplemental Note 5 (Supplemental Figs. 14, 15) in the model the increase in firing rate of the PF inputs to stellate cells (SCs) during odor stimulation induces an increase in SC firing for S+ in the control condition (Supplementary Fig. 14b). For S- stimulation we postulate that strong LTD at PF-SC decreases the odorant-induced increase in SC firing rate. For chemogenetic inhibition (hM4Di+CNO) we postulate that CNO is a partial inhibitor (it inhibits SCs by 50%). Therefore, the S+ induced increase in PF firing elicits an increase in firing in the remaining SCs. On the other hand, for S- the model suggests that there is an increase in SC firing in the hM4Di+CNO condition as a consequence of the occurrence of a weaker LTD.

Therefore, we use a simplified model of the PF-MLI-PC circuit to provide a plausible explanation of the behavioral outcome of the chemogenetics experiments. This is discussed in a new paragraph in the discussion (line 386): “A question that arises is which circuit mechanism is responsible for the decreased behavioral performance after chemogenetic inhibition of MLI activity (Fig. 7 and Supplementary Fig. 13). MLIs receive sensorimotor information from multiple GCs through PF input and *in vivo* studies have found remodeling of MLI receptive fields upon repeated electrical stimulation of the skin⁶. In addition, plasticity in PF-MLI synapses are postulated to increase the information capacity of the MLI-PC network and richness of PC output dynamics^{7,8}, and a model of PF-MLI plasticity has been proposed⁹. If long term plasticity of PF-MLI synapses is responsible for the large change of MLI responsiveness found here upon reversal of stimulus valence it is likely that the error signal would be provided by CF spillover resulting in highly redundant stimulation of stellate MLIs^{5,10}. We developed a model of the MLI/PC circuit described in Supplementary Note 5 (Supplementary Figs. 14,15) that suggests that plasticity in SC-PC synapses¹¹, PF-SC synapses⁹, complemented with CF spillover acting on the feedforward disinhibitory MLI circuit described recently by Arlt and Hausser⁵, would explain the changes in behavior we find after

chemogenetic inhibition of MLI activity. Finally, we found that the divergence between the time courses for lick frequency between S+ and S- took place at a later time when MLIs were inhibited by chemogenetics (Fig. 7c,ii, Supplementary Fig. 13g) likely reflecting slow decision-making, consistent with a role for the CF/PC circuit in reward timing prediction¹². Future studies are necessary to understand the role of plasticity in the PF-MLI-PC circuit in associative learning.”

Reviewer #1 Minor comments:

1) Figure 1 – include a description of the proportion of MLIs that are task active vs non-responsive in the main text. In line 103 we state: “the rewarded odorant elicited increases in $\Delta F/F$ in 89% of the ROIs among a total of 191”.

2) Supp. Video 1 – the unfortunate camera angle makes it very difficult to observe mouse behaviour which involves orofacial movements and engagement with reward delivery apparatus that is presented to the mouth. The authors should instead image the behaviour from a front elevation. We agree. Unfortunately the videos were all taken from this point of view. In the future we plan to use at least two fast frame rate cameras to allow more thorough analysis of tongue and body movement.

3) Supp. Fig. 3 – reference is made to the lick 1-1.5s before odorant delivery onset. Why does this not trigger trial onset and odour delivery? Is there a pseudo randomised delay period from one trial to another or is the ‘pre-odour’ lick equivalent to the mouse counting the passage of time? This is not clear from the text, methods section or Supplementary figures. In line 515 of the methods we added: “The odorant was delivered after a random time interval ranging from 1 to 1.5 seconds.”

Also, we modified the legend of Supplementary Figure 3 (Sup. line 226): “The mouse starts the trial by licking on the water spout and the odorant is delivered after a random delay of 1 to 1.5 sec.”

4) Line 26 – ‘restrain’ should read ‘refrain’? line 28 was changed to “refrain”

5) Line 31 – ‘licks diverged more slowly’ is rather confusing for the reader. This was changed to (line 32): “Finally, mice took a longer time to refrain from licking in the presence of the unrewarded odorant and had difficulty becoming proficient when MLIs were inhibited by chemogenetic intervention.”

6) Line 60 – to test sufficiency (which is rather difficult to achieve at the systems level) one would need to show that driving MLI activity alone is ‘sufficient’ to drive olfactory discrimination, this is not what is shown in the manuscript. Instead, the results appear to show that MLI activity is necessary for the task. Line 66 was changed to “Here we explored whether MLI activity plays a role in reward-associated learning in a go-no go task where the thirsty animal learns to lick to obtain a water reward^{13,14}.”

7) Line 81 – ‘proficient discriminating’ – reword. Line 87: “A proficient mouse (percent

correct $\geq 80\%$) licked at least once in each lick segment in the S+ trials and stopped licking during the segments for the S- trials, see lick trace at the bottom of Fig. 1d.”

8) Line 91 – the consistent, albeit small’ dF change during CR trials should be discussed in the main text. Presumably this reflects the small amount of licking that occurs in CR trials (see Supp. Fig. 3b). We changed the wording in line 100: “In contrast, the unrewarded odorant elicited smaller transient increases in $\Delta F/F$ (Fig. 1cii, Fig. 1d vertical light blue lines and Supplementary Figs. 4b,c,d), with some exceptions where the increases for $\Delta F/F$ for S- were larger (arrow in Fig. 1d).”

9) L221 – Add lick data regarding the average lick rate across all mice during Hit, CR, FA & miss trials. We added panel a to Supplementary Fig. 8 (presented in Supplementary Note 3) showing the mean lick frequency during Hit, CR, FA & Miss trials.

10) L259/260 – the data suggest there is a correlation between licking and dF during odour presentation so the conclusion is a little overstated. The experiments suggested above will shed light on lick-related dF changes during the task. We have changed the wording (line 235) to: “These data suggest that changes in $\Delta F/F$ during the reinforcement period reflect changes in lick activity, while changes in $\Delta F/F$ during the odorant period are less dependent on licks, and maybe dependent on multiple variables.”

11) L231 – the use of the term ‘slower’ has not been qualified. What is meant by slower? We did not find the term “slower” in L231. We assume this comment is on the use of this term in what was L321. We have changed the wording in line 327 to: “Interestingly, the time course for lick frequency differs between hM4Di mice injected with CNO and the other conditions (Figs. 7c,i and 7c,ii). With the exception of hM4Di + CNO, shortly after odorant addition (~ 0.8 sec) the lick frequency for S- does not increase beyond ~ 1 Hz while the lick frequency for S+ keeps increasing beyond 8 Hz (Fig. 7c). In contrast, for hM4Di + CNO lick frequency for S- keeps increasing beyond 1 Hz and does not diverge from the S+ time course until it reaches 2.5 Hz at a later time point (~ 1 sec, Fig. 7c,ii) likely reflecting slow decision-making.”

12) L378 – the term ‘slowed decision-making’ is not supported by the data shown. We have deleted this sentence in the discussion.

13) Fig. 1 / methods – the reason why mice consistently lick 1-1.5s before the lick that initiates each trial should be discussed in more detail. What in the task structure drives mice to show this behaviour? As was already stated in the methods (line 515) in this task the mouse must lick to initiate a trial. This is now also explicitly stated in the beginning of the results (line 79): “We used the go-no go task where thirsty mice initiate the trial by licking on the spout to elicit odorant delivery 1-1.5 sec after the first lick. Mice received a water reward when they licked at least once in two lick segments during rewarded odorant delivery (1% isoamyl acetate, Iso, termed S+) (Fig. 1a, Hit trial, mouse movement shown in Supplementary Movie 1, quantified in Supplementary Fig. 3).”

14) Fig. 1. – the consistent increase in dF at the beginning of CR trials should be discussed in the

text. In line 100 of the results we now state: “In contrast, the unrewarded odorant elicited smaller transient increases in $\Delta F/F$ (Fig. 1cii, Fig. 1d vertical light blue lines and Supplementary Figs. 4b,c,d), with some exceptions where the increases for $\Delta F/F$ for S- were larger (arrow in Fig. 1d).”

15) Fig. 2 panel e – requires labels to show ‘pre-odorant’ and ‘odorant’. These panels were labeled as requested.

16) Fig. 4 panel f – the decoding doesn’t predict the behaviour it reflects licking vs withholding licking, change axis label and in main text. The title for this panel and references to “prediction” were removed.

17) L685 – label should read (c) not (d). Fixed.

18) Supp. Fig. 3 panel a – from this elevation it is difficult to understand task structure or mouse engagement in the task. In future experiments we will change the angle of view.

19) Supp. Fig. 3 panel b – add details of what the two red lines represent, this is missing in the legend. We added: “The two vertical lines denote times for reinforcement on and off.”

Reviewer #2 (Remarks to the Author):

The manuscript by Ma and colleagues provides imaging data that adds to the growing body of evidence for the role of cerebellar molecular layer interneurons (MLIs) in learning. Further, this is of interest to the more general neuroscience audience as the role of the interneurons is not very well understood.

The two major claims of the paper are:

1. MLIs encode the valance of the stimulus and they acquire this over time.
2. MLIs are necessary to reach proficiency in this associative learning task.

The first claim is justified, although there are quite a lot of methodological and statistical issues that need to be addressed to support this conclusion. The second claim requires additional experimental data. As of now the evidence presented by the authors does not give enough support to conclusion #2. Thank you, we address the specific points below.

I have several suggestions that could improve the manuscript and strengthen the conclusions presented by the authors. However, before going into the details there are two major problems that severely impair the readability of the manuscript and need to be addressed in the revision.

1. The colors used in the figures need to be redesigned. Currently same colors indicate different things within the same figure.

Examples:

Figure 2: color blue in panels b, e and f indicate S- odorant (not rewarded) and red S+ odorant (rewarded). But in panel g the same colors have a different meaning as the dimensionality analysis takes under consideration both S- and S+ trials. Here (I assume) blue depicts low

learning (trials? sessions?) and red proficient (trials? sessions?)

Figure 3: color blue in panel b, indicate S- odorant (not rewarded) and color red S+ odorant (rewarded). Note: legend in that panel is missing. However, in panel d same colors do not mean the same thing. Just like in Figure 2 (I assume) blue depicts low learning (trials? sessions?) and red proficient (trials? sessions?)

Figure 7: possibly the worst color-coding of all figures. In panel b the four colors indicate Miss, CR, FA and Hit trials but in panel c THE SAME colors now depict different time epochs in trials. Colors in panels c and d are confusing and redundant.

Figure 8: The same colors that previously were used for Miss, CR, FA and Hit trials (and different time in trials in Figure 7) now stand for different drug conditions.

This list is not exhaustive. Almost every figure suffers from such poorly chosen color scheme. It is crucial to address this in the next submission. Moreover, the colors chosen for Miss and CR, and FA and Hit trials are very similar to each other making the data points in figure 2c, 2d and 4c very difficult to distinguish. Finally, panels 2a, 3a, 4a, 8a, 8b and supplementary figure 4 use colors very similar to the ones depicting trial types described above. Overall, the chosen color scheme makes the figures close to unreadable and hard to interpret. Thank you for the comment. We have modified the color representation scheme and we have used colors acceptable for color blind individuals. <http://mkweb.bcgsc.ca/colorblind/img/colorblindness.palettes.trivial.pdf>

2. Style of writing makes it difficult to follow the story. Instead of saying “Figure X shows....” authors should describe their findings and refer to the figure using brackets (Figure X).

Describing each panel separately in a point-by-point manner is appropriate for a figure caption but does not suffice for the body of a manuscript. We have thoroughly revised the manuscript accordingly.

“Other major issues:

- Abstract: The abstract needs to be changed in order to resolve the following issues: a) Time-out is not really a “punishment”. I would avoid using this terminology;” The wording was changed to (line 27): “...and avoid a time out when they restrain from licking for the unrewarded odorant”.

b) Sentence in line 26/27 is confusing (“When the animal is naïve...”). Please do not use the brackets to explain terminology. This almost reads as a draft. Rather I’d suggest re-writing it. Perhaps something like “In naïve animals the MLI responses did not differ between the odorants. With learning, the rewarded odorant elicited a large increase in MLI calcium responses/transients/signals, and”. Or something to that end. Thank you, we used the wording you suggested (line 28): “In naïve animals the MLI responses did not differ between the odorants.”

The last part of the abstract regarding chemogenetic inactivation of MLIs should be re-evaluated pending experimental input (see my comments below regarding Results). Please see the answer below where we describe a new experiment making the chemogenetic inhibition experiments stronger. The line was re-worded to (line 32): “Finally, mice took a longer time to refrain from licking in the presence of the unrewarded odorant and had difficulty becoming proficient when MLIs were inhibited by chemogenetic intervention.”

- Results/Methods:

1. The number of animals used in each figure has to be explicitly stated, both in figure captions, results section and methods. When the n refers to sessions or trials, these should also be specified. In the revised manuscript we specify the number of sessions and mice throughout.

Methods should explain why only 3 mice were reversed. For the go-no go dF/F imaging experiments we used 5 mice. However, for two mice a cloud developed under the window making it impossible to image the MLIs for the reversal go-no go session. This is now stated in the methods (line 529): “Five mice were used for the go-no go experiments. Four mice were imaged when they were naïve. The window became opaque for two of the mice preventing MLI imaging for the reversal experiment.”

Additionally, why do figure 2 and supplementary figure 4 only depict 4 mice while 5 mice were used according to methods section (lines 447-457), please explain. The reason is that when we recorded from the first mouse, the mouse learned to differentiate the odorants in a session where we were learning what was the optimal odorant application period. In this session we changed the odorant application period from 2 seconds to 4.2 seconds. Because of that, we only used data for subsequent sessions for that mouse that was already proficient. For all other mice we used the 4.2 sec odorant application period. The first mouse was not used for Figure 2, but it was used for analysis of proficient mice. In the methods we state (line 531): “Five mice were used for the go-no go experiments. Four mice were imaged when they were naïve. The window became opaque for two of the mice preventing MLI imaging for the reversal experiment.”

2. Authors pool data obtained with GCaMP6f, GCaMP6s and GCaMP7f and justify this saying, in line 445, that “similar results were obtained” using those indicators. Given very different kinetics of these calcium indicators it is crucial for the authors to show the reader the comparisons, including statistics. This can be done in a supplementary figure. This study started as a side project when Diego was going to Isabel’s lab to deliver a miniature fiber coupled two-photon microscope. Since we had not performed studies in the cerebellum in Denver we decided to do a few head-fixed imaging experiments and we tested several GCaMPs to attain optimal photon emission. However, when it became clear that the results were interesting, and that the kinetics of the GCaMPs did not differ greatly, we decided to do a full study with the animals we had at hand. In the future we should do all experiments with the same GCaMP.

In the revised manuscript Supplementary Fig. 16a (mentioned in lines 605 to 607) shows that the kinetics for MLI dF/F do not differ greatly between the GCaMPs used. Furthermore, in the original manuscript we showed the mean \pm CI for dF/F measurements, but we did not show the results for each mouse/session. In the revised manuscript we show all the individual data points giving evidence that the different mice yield similar results (no outliers) (e.g. Figs. 2 f and g, Figs. 3d and f, Figs. 4d and, Figs. 7 c and d, Supplementary Fig. 7 and Supplementary Fig. 8). The results for individual points do not suggest a difference due to the different calcium sensors.

The kinetics of the dF/F time courses we record in MLIs are slow, with increases in dF/F that take place and decay during several seconds. These changes in dF/F are much slower than the $t_{1/2}$ for any of the GCaMP calcium sensors used¹⁵. The slow kinetics are likely due to the fact

that for interneurons the calcium changes reflect an increase in firing frequency to a high firing rate. In this case the kinetics of Ca changes due to single spikes cannot be discerned. On purpose we did not analyze the fast calcium changes that could reflect single spikes. If we had done an analysis of fast calcium changes underlying high frequency burst the choice of GCaMP would have made a difference.

3. What is the inter-trial interval? This should be mentioned in the methods section. If the ITI was fixed rather than random this could explain why the animals start licking and the calcium signal starts rising prior to the odorant exposure. If the ITI is indeed random is it possible that the animals are responding to the click or hiss of a valve? As stated in line 79, these are self-initiated trials started when the mouse licks the spout. This is likely why the dF/F starts rising when the animal starts the trial. Also, in the revised manuscript we state in line 518 that the minimum ITI was 22.3-22.8 sec. Because this is a not an instrument-initiated trial where the animal reacts to a start cue (e.g. sound of valves) it is unlikely that the animals are responding to the sound of the valve. We did not explore whether the noise elicited neural activity.

4. Currently, correct rejection trials still contain lick responses in one segment (line 466-469 in Methods). Are there any trials where the animals show a full suppression of the lick response spanning both segments? If this is the case this should be analyzed as a different type of trial. This information should also be described in the Results section. We analyzed the relationship between dF/F and lick frequency between CR trials when the animal shows full suppression of licking and those when the animal licks. We did not find differences for dF/F between these two kinds of CR trials (new Supplementary Fig. 10, Sup. lines 70 to 77 in Supplementary Note 4).

5. What bootstrapping technique was used to obtain Figure 2b? Explain in the results and methods. In line 613 of the methods we state that “95% CIs shown in the figures as vertical black lines or shading bounding the lines were estimated by bootstrap analysis of the mean by sampling with replacement 1000 times using the bootci function in MATLAB.”

6. The dimensionality calculation presented in Figure 2g should include a comparison between “learning” and “proficient” conditions. Currently, only within condition comparisons are indicated. In the revised manuscript (Sup line 14) we state that: “The dimensionality ranged from 2 to 6 and did not differ significantly between time periods or between naïve and proficient (GLM $p > 0.05$, 24 observations, 18 d.f., $n = 4$ sessions, 4 mice, GLM F-statistic = 2.42, $p > 0.05$).”

7. Results presented in Figure 5a seem to be in direct conflict with data presented in Figure 2c and 2d and Figure 4c. Specifically, the authors claim that as depicted in Figure 5a the calcium responses to Hits and Misses are virtually indistinguishable. However, the same type of trials in the other figures show much smaller calcium transients. For example, looking at Figure 4b middle bottom panel (trial rev91) the calcium response to a Miss trial is negligible and definitely different than a Hit response. Is the analysis presented in Figure 5 done on all trials or only at the trial of $> 80\%$ accuracy. These issues have to be resolved in order to support the main conclusion of the paper. There is no conflict between these data. First, as was stated in the results and the figure legend, the data for what was Fig. 5a (now Supplementary Fig. 8b) is for trials when the mouse was proficient ($> 80\%$ correct). These data are consistent with the trials in Fig. 2d for trials > 70 , when the animal is proficient (and with Fig. 2b, where a subset of the trials in Fig. 2d are

shown). The data in what was Fig. 5a (now Supplementary Fig. 8b) are different from the data in Fig. 4b middle bottom panel (trial rev91) because that trial takes place shortly after reversal. The dF/F for a Miss, shortly after reversal, looks like a CR before reversal, as expected because the animal is still responding to this stimulus as if it were unreinforced. The text describing the reversal has been modified to make this clear (Supplementary Note 3).

Moreover, the fact that the MLIs show similar responses after a reward is obtained (Hit trial) and withheld (Miss trial) should be further discussed in the Discussion section. This is an excellent suggestion, and we have modified the results to highlight this finding that is consistent with MLIs carrying information on valence (new Supplementary Note 3). We have added a panel to what is now Supplementary Fig. 8 (previously Fig. 5) showing the mean dFF for the different events for all the experiments (new Supplementary Fig. 8c). We have revised the text in Supplementary Note 3 that describes the data showing that MLIs show similar responses after a reward is obtained (Hit trial) and withheld (Miss trial). In addition we have added a reference to these data in the discussion (line 374): “We do not find that the MLIs respond to odorants per se. Rather, the reversal experiment (Fig. 4) and the similar $\Delta F/F$ responses and stimulus decoding for correct and incorrect behavioral response trials (Supplementary Fig. 8) indicate that MLIs respond to contextual odorant identity: “is this the rewarded odorant?”, which is directly related to valence, a binary measure of an emotion reflected by the motivation to receive reward¹.”

8. The decoding accuracy analysis presented in Figure 5 should also be applied to reversed trials. We show this analysis in Supplementary Fig. 8e. The results of LDA analysis for the different events are similar for the reversed and forward sessions.

9. Looking at data presented in Figure 6d it is unclear how the FA trials are classified. The lick responses for FA and Miss trials look almost the same. Is there a behavioral difference in the response? Please quantify and provide statistics. We had not stated in the beginning of the results what the rule is for classifying the events. As was already stated in the methods, we now state in the beginning of the results (line 80) that: “Mice received a water reward when they licked at least once in two lick segments during rewarded odorant delivery (1% isoamyl acetate, Iso, termed S+) (Fig. 1a, Hit trial, mouse movement shown in Supplementary Movie 1, quantified in Supplementary Fig. 3). Mice did not receive the reward if they failed to lick in one of the two lick segments (Miss trial).”

Furthermore, Fig. 6d (now Fig. 5d) now shows the lick trace (black lines above the lick frequency). This makes it evident that for Fig. 5d for the FA there was one lick at the very beginning of the odorant period (within the first two second lick segment), and a couple of licks during the second two second lick segment. This would classify the trial as a FA.

10. The DREADD experiments do not support the conclusion that MLIs are necessary to learn the task as the chemogenetic inactivation is performed on trained animals. These experiments should be expanded by a) introducing CNO from the beginning of the training, or b) reversing the mice under CNO. We performed the experiment suggested by the reviewer where CNO was applied at the beginning of training and reversing the mice under CNO (Supplementary Fig. 13). We obtained similar results to the experiment with the odorant mixtures (Fig. 7). The revised chemogenetics experiments are described in lines 342 to 345 and the methods are updated in the

revised manuscript (lines 557 to 564). This strengthens our claim that MLIs play a functional role in associative learning. In addition, we generated a simplified model of PF-MLI-PC to describe the circuit basis for the behavior found with chemogenetics (see point 6 of reviewer 1).

Moreover, current results point to a motor impairment given a smaller lick response, rather than contextual learning. The authors state this themselves in lines 329-331. This conclusion is in stark contrast to the claims made in the abstract. Please provide supplementary figures with quantifications for the pattern of DREADD expression in the cerebella of all inactivated animals. We did not find a smaller lick response in the presence of CNO. The wording in the results was not clear. What we find is that in the presence of CNO lick frequency increases above the value in the absence of CNO. We have changed the description of the results to: “Interestingly, the time course for lick frequency differs between hM4Di mice injected with CNO and the other conditions (Figs. 7c,i and 7c,ii). With the exception of hM4Di + CNO, shortly after odorant addition (~0.8 sec) the lick frequency for S- does not increase beyond ~1 Hz while the lick frequency for S+ keeps increasing beyond 8 Hz (Fig. 7c). In contrast, for hM4Di + CNO lick frequency for S- keeps increasing beyond 1 Hz and does not diverge from the S+ time course until it reaches 2.5 Hz at a later time point (~1 sec, Fig. 7c,ii) likely reflecting slow decision-making. This would explain why the hM4Di + CNO mice accumulated errors in the go-no go task (Fig. 7bii).”

11. Behavioral analysis of body movement presented in Supplementary Figure 3 is insufficient. First, head-fixed mice cannot “lean” into the port. Second, Supplementary Movie 1 shows movements of different body parts (hind legs, front legs, tail) and overall shuffling of the body. I recommend using DeepLabCut, or similar, analysis to disentangle different movements. Yes, the mouse does not “lean” into the port. We have changed the title of Supplementary Fig. 3 (Sup line 223) to: “Supplementary Figure 3. The mouse moves the body during the trial”.

Furthermore, we performed DeepLabCut analysis of the movements of the root of the tail and the knee of the hind leg. For the tail the velocity measured with DeepLabCut was highly correlated with the velocity measured with the optic flow algorithm ($\rho=0.98$). This is expected because optic flow is a well-validated algorithm that is often used in engineering studies (although it is not used regularly in neuroscience). We added DeepLabCut measurements of velocity to Supplementary Fig. 3. Finally, because the DeepLabCut and optical flow measurements were highly correlated we did not revise the GLM analysis in Figure 7.

The velocity for the movement of the knee measured with DeepLabCut was correlated with the movement of the tail ($\rho=0.38$, $p<0.05$) as expected for a coordinated movement of the body. Unfortunately, the knee hid behind the cylinder making the likelihood of the knee measurement with DeepLabCut decrease often during the time course (compare panels c and d below; likelihood is a measure of position is for each frame in DeepLabCut). As a result, the movement of the knee measured with DeepLabCut is not reliable. We agree with the reviewer that further detailed study of body movements could provide useful information. Nevertheless, using DeepLabCut with a single camera at the angle used in this study, with a plastic cylinder hiding the knee for a substantial amount of the time provides limited information. In line 530 of the methods we state that our study of the body kinematics of the mouse in the go-no go task yields limited information.

- Introduction:

1. At some point the difference between stellate and basket cells should be mentioned, see Brown et al., 2019. Yes, this is an important point. We have chosen to discuss stellate and basket cells in the discussion (line 425): “Furthermore, we focused our recordings on the more superficial regions of the molecular layer and therefore, most of our measurements are from stellate cells. Recordings from synaptically connected pairs of MLIs and PCs in slices showed that mean the amplitude of synaptic currents decreases with distance from the PC layer, suggesting a stronger impact of basket versus stellate cell inhibition on PC firing¹⁶. Recent recordings of PC spikes in vivo following genetic deletion of MLIs confirm this prediction¹⁷ which is in accord with the morphological diversity of MLIs¹⁸. Finally, the effect of chemogenetics (Fig. 7) should be on both stellate and basket cells, and future experiments are necessary to differentiate between the role of the two cell types.”

2. Authors should explicitly discuss why they chose to image Lobule VI over Crus I or simplex, which also show reward related signals. Please discuss papers by Kostadinov et al. 2019 and Heffley et al., 2018 and 2019. Initially we chose to study lobule VI because of the work from Giovannucci et al. and Wagner et al. We found interesting changes in the activity in MLIs in the go-no go task and decided to do the entire study in this region. We chose not to study regional differences in this manuscript.

We have added these sentences to the discussion (line 419): “Here we found changes in MLI activity during learning in the go-no go associative learning task in lobule VI where GCs^{2,3} and CF¹⁹ activity was proposed to encode aspects of reward signaling. Recent work on the contribution of cerebellar processing to execution of reward-driven behaviors indicates that multiple cerebellar regions are involved, including central and lateral cerebellum^{2,3,19-21}. As behavioral tests are refined, it is likely that differences in how these regions process reward will emerge, as suggested by the recent work on climbing-fiber signaling²⁰.”

“3. When discussing plasticity in the cerebellar circuit the authors exclusively focus on LTD without mentioning LTP and intrinsic plasticity. This is a serious omission as it has been shown that LTD is not essential for motor learning, see Schonewille et al., 2011.” In line 52 of the

introduction we have added text discussing LTP and intrinsic plasticity. We reference Schonewille et al., 2011, and other appropriate references: “Furthermore, although LTD at the PF-PC synapse is often considered as the substrate for cerebellar dependent learning^{7,22-24}, such learning can occur in the absence of LTD and may therefore involve other forms of plasticity²⁵. A potential substrate for plasticity is the PF-MLI synapse^{8,26} where LTP can be induced in slices by pairing MLI depolarization with PF stimulation²⁷ and *in vivo* by conjunctive stimulation of PFs and CFs²⁸, believed to underlie changes in the size of cutaneous receptive fields^{29,30}). Additionally, high frequency stimulation of PFs alters subunit composition of AMPA receptors, rendering them calcium-impermeable²⁷, a long-lasting change linked to behavioral modifications³¹.”

- Discussion:

1. Discussion should be expanded to address raised inconsistencies in the Results. However, it is advisable to rewrite the discussion pending suggested experiments and analyses. We have updated the discussion taking on account the reviewer’s comments and the new data we present.

Minor issues, Reviewer #2:

1. Figure panels 2e and 2f should have a title for quicker interpretation. These panels are now labeled “Pre-odorant” and “Odorant”

2. The text in lines 169-170 is inconsistent with the legend of figure 4b. We revised the sentence.

3. Figure 5b left bottom panel (fwd51) contains a mistake: MO(s+) should be MO(s-). We have changed the label to MO(S-) (note: this refers to Fig. 4b, not what was Fig. 5b, now Supplementary Fig. 8b).

4. In figure 5, the legend states point d which should be c. This was corrected.

5. Why was LDA analysis chosen? Did authors try other classifiers? Some justification is needed (in Results or Methods). Yes, we also tried perceptron analysis and we obtained similar results. However, LDA was faster. In addition, the fact that decoding accuracy increased as the animal learned is informative, regardless of whether we optimize the classifier. Thus, for this study we did not deem it essential to optimize the classifier.

6. Similar to Supplementary Figure 4, please show behavioral plots for all reversed mice. The percent correct for the other two reversals are shown in Supplementary Figs. 6e,f.

7. Figure 6a is missing a color legend. We added color legends to what used to be Fig. 6a (now Fig. 5a).

8. Figure 6b: the legend is swapped in the panel. Red should depict S+. The legends were swapped.

9. Figure 7c and 7d: what do individual dots refer to? Sessions? Trials? Animals? These are 6 sessions (5 animals). This is now stated in the legend.

10. Figure 7b: it would be helpful to provide goodness of fit analysis for data presented in this panel. The percent variance explained by the fit is shown in Fig. 6c.

11. In line 463, please clarify what “were ready” stands for. The wording was changed to: “When the animals were thirsty...”

12. Reference number 5 is a review. It would be better to cite experimental papers, such as Tsai et al., 2013, Stoodley et al., 2017 or Badura et al., 2018. We replaced reference 5 with Badura et al., 2018.

13. When reporting statistics (example- line 110), include statistic value and error. We have added the F-statistic and p value to the GLM statistics.

14. When figures are being described in the text, significances are often reported that do not have a corresponding star on the graph (example line 194). Consider adding these for easier interpretation of figures. We have added stars to the figures.

15. In the methods section, references to the used product/compounds should be added. We have added references to the products/compounds used.

Reviewer #3 (Remarks to the Author):

“In this manuscript, Ma et al. investigate how cerebellar molecular layer interneurons respond as a function of learning in an olfactory sensory discrimination task. The authors use in vivo calcium imaging to measure how these interneurons respond to odorants across learning, and conclude they develop preferential responses to whichever stimulus accurately predicts upcoming reward. Moreover, they find that responses are linked most strongly to the stimulus identity rather than the trial outcome, suggesting perhaps that responses reflect a context dependent stimulus representation rather than some form of reward prediction per se. Moreover, chemogenetic silencing of these interneurons impairs learning in this task, it seems by decreasing the animals’ ability to restrict false alarms, implicating the cerebellum in a particular aspect of this associative learning task. Overall, these findings could be of considerable interest to the field of cerebellar research and those interested in associative learning, as recent work has begun to reveal new roles for the cerebellum in reward-based learning. However, I have several concerns related to the completeness of data presentation and analyses that limit my enthusiasm for the manuscript in its current form. These issues should be addressable if authors can enhance their analysis of the relationship between neural activity and behavior, and the presentation of neural data across cells and mice.” We are pleased that the reviewer finds that this study could be of considerable interest to the field of cerebellar research and those interested in associative learning. The reviewer raises several interesting issues that we have addressed by enhancing our analysis and modifying the data presentation (below).

Major Concerns:

1. “1) In general, the data are underrepresented by anecdotal examples rather than comprehensive depictions that span neurons and mice. To evaluate the veracity of claims that stem from high level dimensional reduction approaches and GLM output, it is necessary to see overall population responses and analyses for all the individual contributing ROIs. This should include, for example, mean DF/F responses across all contributing ROIs for each trial type (hits, misses, false alarms and correct rejects). Similarly, scatter plots or alternate depictions should be used to show variability across all the ROIs. As currently presented, it is not clear how many ROIs go into each experiment, let alone what the responses of all cells look like.” In the revised version of the manuscript we provide a thorough illustration of the per-ROI changes in DF/F and, more importantly, we show a decoding analysis with randomly chosen subsets of ROIs. We show that large numbers of ROIs respond to the rewarded odorant in mice proficient for the odorant discrimination. Specifically, the new Supplementary Fig. 4 shows that all ROIs for the data in the example in Fig. 1 respond with a larger change in DF/F to the rewarded odorant. Furthermore, Fig. 2e shows that DF/F is larger in the presence of the odorant for proficient mice for the majority of ROIs. Finally, Supplementary Fig. 9 shows an example of the per-ROI responses for each trial type (Hits, Miss, CRs and FAs).

More importantly, we performed LDA odorant decoding analysis for different ROI numbers where we chose for each number of ROIs 50 random non-overlapping subsets of ROIs. The LDA subset analysis in Figs. 3e and f shows that, even if we choose one ROI, the decoding accuracy is significantly higher than shuffled data (described in Supplementary Note 2). Therefore, as shown

in the example Supplementary Movie 2 most ROIs respond simultaneously explaining the low dimensionality of the ensemble response. Furthermore, we provide a new table (Table 1) showing the number of ROIs for each experiment. Finally, we calculate dimensionality, a method to quantify redundancy in responses of multiple components. Supplementary Fig. 7 shows that the dimensionality in the presence of the odorant is low (ranging from 1 to 4) documenting that ROI responses are highly redundant.

Along these lines, more detailed analyses of response types across neurons would be helpful. For example, is it the same MLIs that respond to Hits, misses, FAs and CRs? Or are these separate populations? What fraction of identified neurons in a FOV are responding on different trial types? Are the same neurons tracked across learning (this is not clear from the text)? This would be particularly useful for the reversal learning data. If the same neurons are tracked, is it the same or new cells that develop responses as a function of learning? Such information would be useful to understand potential contributions to learning (as is suggested by the chemogenetic manipulation). The new Table 1, discussed in Sup, line 46 shows that virtually every ROI (>99%) that responds with an error (Miss or FA) also responds with a Hit. Furthermore, virtually every ROI identified by CaImAn responds with a Hit (>99%). Finally, regardless of the trial type a vast majority of ROIs respond. Because of this it was not necessary to track ROIs across the experiment. It is now stated in the methods that MLI recording in each go-no go session was performed in several 6000 image time series (line 583). ROIs were tracked within each of the time series. Finally, we did not track ROIs across reversal, and as a result we cannot claim that the same ROIs reverse. However, given that most ROIs respond this is likely the case.

2) The relationship between neural activity and behavior is under analyzed. Many of the changes in neural activity likely correlate with the changes in licking that occur across learning, but the authors only use the derivative of lick rate to test for a correlation with neural activity. Please show mean lick rates for all trial types for low and high proficiency learning conditions, and include these on the same timescale with the mean DF/F for all trial types across ROIs. Depending on the clarity with which such mean responses segregate behavior and neural activity, further analyses leveraging variability in licking across trials may be necessary (e.g. segregating within trial type based on the amount of licking). Without such information, it is difficult to interpret many of the results. We agree. While in the original manuscript we used both lick rate and the derivative of the lick rate as predictive variables in the GLM analysis, we did not show the DF/F vs lick rate analysis in Fig. 6 (this is now Fig. 5). We have added this analysis in Figs. 5e and g and we have revised the text in lines 217 to 242. The new analysis shows that, as expected, there is a correlation between DF/F and lick rate for both the odorant and reinforcement windows. However, consistent with the larger contribution of licks to GLM during the reinforcement period, the correlation between DF/F and lick rate is larger for reinforcement compared to the odorant window (Fig. 5g).

Furthermore, reviewers 1 and 2 suggested two other related analyses of the relationship between dF/F and lick rate: Following up on point 4 of reviewer 2 we analyzed the relationship between dF/F and lick frequency between CR trials when the animal shows full suppression of licking and those when the animal licks. We did not find differences for dF/F between these two kinds of CR trials (new Supplementary Fig. 10, Sup. lines 70 to 77 in the results). Furthermore, in response to point 3 of reviewer 1 we asked the question whether there are instances where there

is a clear dF change during odour presentation but licking is delayed. In Crus II, where dF/F reflects lick rate (Gaffield and Christie, 2017 and Astorga et al. 2017) have shown that whenever there is a sharp increase in dF/F there is a corresponding increase in lick frequency. Supplementary Figure 11 (Sup lines 79 to 92) shows that this is not the case in our study.

These results indicate that for mice engaged in this go-no go task dF/F is not exclusively dependent on lick rate and argues for the multivariate GLM approach that we used. In our opinion, a more detailed analysis of the relationship between licks and DF/F is interesting, but it is beyond the scope of this manuscript.

For example, if licking best explains the neural activity in the 1 second after reward delivery as suggested by the GLM and in Figure 6, why is neural activity the same on average in this same period for Hits and Misses (Figure 5A)? One might conclude that licking is actually not different despite the categorization of these trials as ‘misses’, but without showing the licking is impossible to tell. First, GLM does not show that licking explains all of the variance for dF/F in either the odorant or reinforcement periods (Fig. 6c shows that ~50% of the variance is explained by the GLM). However, the point that the reviewer is important. We should show the dependence of lick rate We performed an analysis of lick rate and dF/F for correct responses (Hits and CRs) and errors (Misses and FAs). In Supplementary Fig. 8a we show the lick rate for each event and in Supplementary Fig. 8c we show dF/F per event. We find that while there are statistically significant differences in lick rate between correct responses and errors, there is no statistically significant difference for dF/F (Supplementary Note 3). Again, dF/F is not exclusively dependent on lick rate. This is consistent with MLI activity representing valence.

Related to this point, I am not confident in the criteria used to segregate trial types based on licking. Figure 6d, bottom shows very similar licking, specifically during the odorant period (between the black bars) for CR, FA and Miss trials. Based on these representative examples, the criteria for trial segregation requires more empirical justification. The criterion for water reward is licking at least once in each of two 2-sec periods during odorant application (Fig. 1a). We have modified the examples in Fig. 5d (that used to be Fig. 6d) to show the lick onsets (vertical black lines above the lick rate). The lick onsets are consistent with the trial classification. We have modified wording in a couple of places where we had stated that “sustained licking” was required; that was incorrect. Lines 80-84: “Mice received a water reward when they licked at least once in two lick segments during rewarded odorant delivery (1% isoamyl acetate, Iso, termed S+) (Fig. 1a, Hit trial, mouse movement shown in Supplementary Movie 1, quantified in Supplementary Fig. 3). Mice did not receive the reward if they failed to lick in one of the two lick segments (Miss trial).”

3) The data show that DF/F begins increasing 1-1.5 seconds before odorant presentation (Figure 2, S+ trials, and S- trials also? Hard to tell for S- trials with the data behind the S+ trials). Is there some cue that indicates the start of the trial that is not described in the paper that lets the animal know when the odorant will be delivered? Perhaps the ITI is constant so that animals can predict the trial onset, and the animals know based on timing when the stimuli are delivered? Such task features are not described, but are essential for interpreting these results. If the task timing is predictable, it would be especially important to explicitly evaluate the role of such predictions in the neural data. As stated in line 79, these are self-initiated trials started when the mouse licks the

spout. This is likely why the dF/F starts rising when the animal starts the trial. Because the task is self-initiated the ITI varies from trial to trial, but there is a minimum ITI imposed by the computer that varies randomly from 22.3-22.8 sec. In the revised manuscript we state in line 518 that the minimum ITI was 22.3-22.8 sec. This is important because mice tend to use the shortest ITI possible for the first few trials (the ITI becomes longer as the animal becomes satiated). The reviewer is correct that it is important to evaluate the role of predictions and that is why we included reinforcement history as one of the variables in the GLM in Fig. 6. However, the percent of variance explained by reinforcement history was low.

4) The interpretation of the results in this manuscript is limited. The discussion does not make clear whether and how these findings fit within a reward prediction framework, or specifically how they compare and contrast with what has been shown for climbing fibers and granule cells in other cerebellar reward-based learning tasks. What do these data tell us about reinforcement learning and the cerebellum? In order to address this point and point 6 of reviewer 1 we have developed a computational model of the interaction of MLIs with PCs to explain the relevance of our findings to circuit function and behavioral output in the cerebellum. Interestingly, Arlt and Hausser⁵ published a manuscript showing that climbing fiber spillover stimulates with high probability superficial MLIs activating a feedforward disinhibitory circuit involving superficial MLIs inhibiting deep MLIs that control PC output. In order to understand how this finding would alter our understanding of the effect of increased MLI activity for S+ and decreased activity for S- on PC output we developed a model of the superficial MLI->deep MLI->PC circuit.

As discussed in Supplemental Note 5 (Supplemental Figs. 14, 15) in the model the increase in firing rate of the PF inputs to stellate cells (SCs) during odor stimulation induces an increase in SC firing for S+ in the control condition (Supplementary Fig. 14b). For S- stimulation we postulate that strong LTD at PF-SC decreases the odorant-induced increase in SC firing rate. For chemogenetic inhibition (hM4Di+CNO) we postulate that CNO is a partial inhibitor (it inhibits SCs by 50%). Therefore, the S+ induced increase in PF firing elicits an increase in firing in the remaining SCs. On the other hand, for S- the model suggests that there is an increase in SC firing in the hM4Di+CNO condition as a consequence of the occurrence of a weaker LTD.

Therefore, we use a simplified model of the PF-MLI-PC circuit to provide a plausible explanation of the behavioral outcome of the chemogenetics experiments. This is discussed in a new paragraph in the discussion (line 386): “A question that arises is which circuit mechanism is responsible for the decreased behavioral performance after chemogenetic inhibition of MLI activity (Fig. 7 and Supplementary Fig. 13). MLIs receive sensorimotor information from multiple GCs through PF input and *in vivo* studies have found remodeling of MLI receptive fields upon repeated electrical stimulation of the skin⁶. In addition, plasticity in PF-MLI synapses are postulated to increase the information capacity of the MLI-PC network and richness of PC output dynamics^{7,8}, and a model of PF-MLI plasticity has been proposed⁹. If long term plasticity of PF-MLI synapses is responsible for the large change of MLI responsiveness found here upon reversal of stimulus valence it is likely that the error signal would be provided by CF spillover resulting in highly redundant stimulation of stellate MLIs^{5,10}. We developed a model of the MLI/PC circuit described in Supplementary Note 5 (Supplementary Figs. 14,15) that suggests that plasticity in SC-PC synapses¹¹, PF-SC synapses⁹, complemented with CF spillover acting on the feedforward disinhibitory MLI circuit described recently by Arlt and Hausser⁵, would explain the changes in behavior we find after

chemogenetic inhibition of MLI activity. Finally, we found that the divergence between the time courses for lick frequency between S+ and S- took place at a later time when MLIs were inhibited by chemogenetics (Fig. 7c,ii, Supplementary Fig. 13g) likely reflecting slow decision-making, consistent with a role for the CF/PC circuit in reward timing prediction¹². Future studies are necessary to understand the role of plasticity in the PF-MLI-PC circuit in associative learning.”

Minor points Reviewer #3:

1. “1) The curation of ROIs is not convincing. Figure 1B shows ROIs that vary much more widely in shape and size than MLIs. For example ROIs 43 and 79 are at least an order of magnitude larger than ROIs such as 3 and 4, and are also shaped much more irregularly than would be expected. This suggests that the Caiman algorithm may not be providing an accurate segmentation, and further data curation is necessary. Plots of ROI size distribution should be included, as well as justification for acceptance or rejection following automated segmentation. For example, for ROIs such as 43 and 79 that lie on the extremes of the pixels size distribution, what features of the response among grouped pixels indicate that these are single neurons?” We have modified the CaImAn parameters to yield a median diameter falling within three standard deviations of the diameter reported for stellate MLIs in ³². Supplementary Fig. 4a shows a histogram of the diameters of the ROIs shown in Figure 1c. The ROIs have a median diameter of 9.95 μm , as expected for stellate MLIs that, according to Chu et al., are $9.13 \pm 0.8 \mu\text{m}$ (mean \pm SD).

We have re-processed all imaging data. The results of the analysis are similar to the results presented in the original manuscript. This is likely due to the fact that there is a low dimensionality in the response of the MLIs, and the constrained nonnegative matrix factorization (CNMF) analysis of CaImAn is designed to find ROIs with pixels that share the same temporal component. However, we appreciate the feedback from the reviewer, and it is best to make the ROIs geometrically as close as possible to single cells.

2) Figure 1C shows the difference between Hits and CRs. This is the least informative comparison, as there is no licking in CRs. Please show all four conditions (including Hits, FAs, Misses and CRs). We are showing Miss and FA in Supplementary Fig. 4e. More important, Supplementary Fig. 8 shows a thorough analysis of MLI responsiveness for all events (Hit, Miss, CR and FA).

3) Why are the first 20 trials in figures 2A and 3A pegged at the same, invariant percent correct? The methods indicate a sliding window of 20 trials, but shouldn't this window move forward for each trial (i.e. trials 1-20, then 2-21)? Perhaps I'm missing some feature of this analysis, but please clarify in the methods. On purpose we assigned the value calculated within the 20 trial window to the last point. This percent correct value estimates the performance in the last 20 trials. This is now stated in the methods in lines 520 to 522. For the first 20 trials we assign the percent correct of the 20th trial to all preceding trials.

4) Figure 4, Panel B, bottom left (fwd 51) should be MO (S-). We fixed this error.

5) Is the legend reversed in Figure 6B, or is this example animal licking more on S- trials? Yes, we swapped the labels.

6) It is confusing to use colors for figure 6E that overlap with the color coded trial types in 6A-D, but do not actually reflect these trial types. We have revised the color scheme.

7) Are the DF/F traces in figure 6D averages across cells as indicated by the legend? If so, why no error bars here? In the updated manuscript we show the 95% confidence interval.

8) Line 464-5 indicates that an olfactometer was used to monitor licks. What does this mean? Was licking measured optically, with video, or something else? We have changed this sentence to (line 512): “Licks were monitored by an electrical circuit monitoring the resistance between the lick spout and the floor in an olfactometer that controlled valves to deliver a 1:40 dilution of odorant at a rate of 2 lt/min.”

9) Line 512: “were be collected” Line 583: “...and several batches of 6000 frames (“time series”) were collected in each training session.”

10) Line 513: “was captured wide field epifluorescence”. Changed to “was captured with wide field epifluorescence”

11) Figure legends are often incomplete, missing details such as what asterisks indicate (Figure 5d), what dotted lines indicate (Figure 8a), etc. While it is possible to extrapolate in most cases, these details should be addressed. Please check the figure legends carefully, as well as for other errors (e.g. Figure 5 has no panel D). We have proofed the figures and figure legends.

1 Tye, K. M. Neural Circuit Motifs in Valence Processing. *Neuron* **100**, 436-452, doi:10.1016/j.neuron.2018.10.001 (2018).

2 Wagner, M. J., Kim, T. H., Savall, J., Schnitzer, M. J. & Luo, L. Cerebellar granule cells encode the expectation of reward. *Nature* **544**, 96-100, doi:10.1038/nature21726 (2017).

3 Giovannucci, A. *et al.* Cerebellar granule cells acquire a widespread predictive feedback signal during motor learning. *Nat Neurosci* **20**, 727-734, doi:10.1038/nn.4531 (2017).

4 Gaffield, M. A. & Christie, J. M. Movement Rate Is Encoded and Influenced by Widespread, Coherent Activity of Cerebellar Molecular Layer Interneurons. *J Neurosci* **37**, 4751-4765, doi:10.1523/JNEUROSCI.0534-17.2017 (2017).

5 Arlt, C. & Hausser, M. Microcircuit Rules Governing Impact of Single Interneurons on Purkinje Cell Output In Vivo. *Cell Rep* **30**, 3020-3035 e3023, doi:10.1016/j.celrep.2020.02.009 (2020).

- 6 Jorntell, H. & Ekerot, C. F. Receptive Field Remodeling Induced by Skin Stimulation in Cerebellar Neurons in vivo. *Front Neural Circuits* **5**, 3, doi:10.3389/fncir.2011.00003 (2011).
- 7 Albus, J. S. A theory of cerebellar function. *Mathematical Biosciences* **10**, 25-61 (1971).
- 8 Dean, P., Porrill, J., Ekerot, C. F. & Jorntell, H. The cerebellar microcircuit as an adaptive filter: experimental and computational evidence. *Nat Rev Neurosci* **11**, 30-43, doi:10.1038/nrn2756 (2010).
- 9 Lennon, W., Yamazaki, T. & Hecht-Nielsen, R. A Model of In vitro Plasticity at the Parallel Fiber-Molecular Layer Interneuron Synapses. *Front Comput Neurosci* **9**, 150, doi:10.3389/fncom.2015.00150 (2015).
- 10 Szapiro, G. & Barbour, B. Multiple climbing fibers signal to molecular layer interneurons exclusively via glutamate spillover. *Nat Neurosci* **10**, 735-742, doi:10.1038/nn1907 (2007).
- 11 Bing, Y. H., Wu, M. C., Chu, C. P. & Qiu, D. L. Facial stimulation induces long-term depression at cerebellar molecular layer interneuron-Purkinje cell synapses in vivo in mice. *Front Cell Neurosci* **9**, 214, doi:10.3389/fncel.2015.00214 (2015).
- 12 Chabrol, F. P., Blot, A. & Mrcic-Flogel, T. D. Cerebellar Contribution to Preparatory Activity in Motor Neocortex. *Neuron* **103**, 506-519 e504, doi:10.1016/j.neuron.2019.05.022 (2019).
- 13 Gire, D. H., Whitesell, J. D., Doucette, W. & Restrepo, D. Information for decision-making and stimulus identification is multiplexed in sensory cortex. *Nat Neurosci* **16**, 991-993, doi:10.1038/nn.3432 (2013).
- 14 Doucette, W. & Restrepo, D. Profound context-dependent plasticity of mitral cell responses in olfactory bulb. *PLoS Biol* **6**, e258, doi:08-PLBI-RA-1962 [pii] 10.1371/journal.pbio.0060258 (2008).
- 15 Chen, T. W. *et al.* Ultrasensitive fluorescent proteins for imaging neuronal activity. *Nature* **499**, 295-300, doi:10.1038/nature12354 (2013).
- 16 Vincent, P. & Marty, A. Fluctuations of inhibitory postsynaptic currents in Purkinje cells from rat cerebellar slices. *J Physiol* **494** (Pt 1), 183-199, doi:10.1113/jphysiol.1996.sp021484 (1996).
- 17 Brown, A. M. *et al.* Molecular layer interneurons shape the spike activity of cerebellar Purkinje cells. *Sci Rep* **9**, 1742, doi:10.1038/s41598-018-38264-1 (2019).
- 18 Palay, S. L. & Chan-Palay, V. *Cerebellar Cortex: Cytology and Organization*. (Springer Berlin Heidelberg, 1973).

- 19 Kostadinov, D., Beau, M., Blanco-Pozo, M. & Hausser, M. Predictive and reactive reward signals conveyed by climbing fiber inputs to cerebellar Purkinje cells. *Nat Neurosci* **22**, 950-962, doi:10.1038/s41593-019-0381-8 (2019).
- 20 Heffley, W. & Hull, C. Classical conditioning drives learned reward prediction signals in climbing fibers across the lateral cerebellum. *Elife* **8**, doi:10.7554/eLife.46764 (2019).
- 21 Heffley, W. *et al.* Coordinated cerebellar climbing fiber activity signals learned sensorimotor predictions. *Nat Neurosci* **21**, 1431-1441, doi:10.1038/s41593-018-0228-8 (2018).
- 22 Gao, Z., van Beugen, B. J. & De Zeeuw, C. I. Distributed synergistic plasticity and cerebellar learning. *Nat Rev Neurosci* **13**, 619-635, doi:10.1038/nrn3312 (2012).
- 23 Marr, D. A theory of cerebellar cortex. *J Physiol* **202**, 437-470 (1969).
- 24 Ito, M. Neural design of the cerebellar motor control system. *Brain Res* **40**, 81-84, doi:10.1016/0006-8993(72)90110-2 (1972).
- 25 Schonewille, M. *et al.* Reevaluating the role of LTD in cerebellar motor learning. *Neuron* **70**, 43-50, doi:10.1016/j.neuron.2011.02.044 (2011).
- 26 Rancillac, A. & Crepel, F. Synapses between parallel fibres and stellate cells express long-term changes in synaptic efficacy in rat cerebellum. *J Physiol* **554**, 707-720, doi:10.1113/jphysiol.2003.055871 (2004).
- 27 Liu, S. Q. & Cull-Candy, S. G. Synaptic activity at calcium-permeable AMPA receptors induces a switch in receptor subtype. *Nature* **405**, 454-458, doi:10.1038/35013064 (2000).
- 28 Jorntell, H. & Ekerot, C. F. Receptive field plasticity profoundly alters the cutaneous parallel fiber synaptic input to cerebellar interneurons in vivo. *J Neurosci* **23**, 9620-9631 (2003).
- 29 Jorntell, H. & Ekerot, C. F. Reciprocal bidirectional plasticity of parallel fiber receptive fields in cerebellar Purkinje cells and their afferent interneurons. *Neuron* **34**, 797-806, doi:10.1016/s0896-6273(02)00713-4 (2002).
- 30 Jorntell, H., Bengtsson, F., Schonewille, M. & De Zeeuw, C. I. Cerebellar molecular layer interneurons - computational properties and roles in learning. *Trends Neurosci* **33**, 524-532, doi:10.1016/j.tins.2010.08.004 (2010).
- 31 Liu, Y. *et al.* A single fear-inducing stimulus induces a transcription-dependent switch in synaptic AMPAR phenotype. *Nat Neurosci* **13**, 223-231, doi:10.1038/nn.2474 (2010).
- 32 Chu, C. P., Bing, Y. H., Liu, H. & Qiu, D. L. Roles of molecular layer interneurons in sensory information processing in mouse cerebellar cortex Crus II in vivo. *PLoS One* **7**, e37031, doi:10.1371/journal.pone.0037031 (2012).

Reviewers' Comments:

Reviewer #1:

Remarks to the Author:

The authors have addressed all of my previous concerns by providing extensive new analysis, supplementary information and additional data. For me the modelling aspect of the study is perhaps the weakest addition to the revised manuscript given the 'parameter selection' based on best guesses rather than recorded activity in the described behavior. That said, the model highlights a plausible mechanism for how hM4Di affects PC output during behavior, which could be tested in follow-on studies.

Ian Duguid

Reviewer #2:

Remarks to the Author:

The revised manuscript by Ma and colleagues makes substantial improvements to the original version. The authors took time and care to address most of the points raised by all 3 reviewers, which is very much appreciated. The current version presents the results in a much more convincing way and is therefore much easier to follow. I also appreciate the change of the color scheme of the Figures and indicating the N in all experiments. However, I still have a few concerns and questions that I would like to see addressed before publication.

Major:

1) Data in new Figure 2e now shows the individual dF/F for all ROIs, which is a good improvement and is appreciated. However, many ROIs seem to have negative $\Delta F/F$, which indicates that the way the baseline is calculated may not be accurate. GCaMPs can only depict increase in the Ca^{2+} concentration. Since $dF/F = (F(t) - F_0)/(F_0 - F_b)$, where F_0 is the baseline and F_b the background, the dF/F_0 can only be negative if F_0 is greater than the $F(t)$. In general negative dF/F is hard to interpret and in fact some 2photon experts, like David Tank, suggest that negative events are false, and therefore the distribution of negative event amplitudes and durations can be used to predict the corresponding distribution of positive events. In line 598 the authors write: "Baseline of intensity (F_0) was defined as the mean fluorescence intensity before trial start, defined as the time when the animal first licked." Negative dF/F suggests that in fact the MLIs were very active during that period. Since movement can increase MLI activity the authors could define F_0 based on the same time-period but only for trials where the animal was very still. Maybe the window should be shorter since the trials were self-initiated? The authors could also just take a small percentile of the average fluorescence signal over that time window as their F_0 . Of course it could reflect the true inhibition but then it's better to calculate the z-score instead of the dF/F . This issue does not change the conclusions but the authors should address the accuracy of their F_0 estimation.

2) Why is there a significant, short-latency peak in the licking frequency in the DREADD experiment that can be seen in Figure 7c? It is occurring $\sim 1s$ before the onset of the odorant and reaches between 5-10Hz. It is not there during the imaging experiments (Figure 5b and Supplementary Figure 5). In fact in these figures we can see a small ($\sim 2Hz$) lick response which peak coincides with the onset of the odorant. The sharp, short peak in Figure 7c looks almost like an artifact. Please check this carefully and provide an explanation to why we can see such response in these experiments.

Minor:

1) The choice to present new analyses and results thereof as "Supplementary Notes" is a bit odd as often the data and text of the notes are very relevant to the interpretation of the results and figures. I

would strongly urge the authors to incorporate these notes into the main results section or (where appropriate) to the Methods. Leaving these details out of the main body of the manuscript decreases readability. For example: Note 4 is crucial to support the authors' conclusion regarding the licking behavior.

2) The same applies to the model. I find it odd that it's not presented under Results but in the Discussion.

3) Abstract could use a sentence or two of an introduction before the authors proceed to the description of the experiments so that the readers can understand what the manuscript is about. I suggest something like: "The cerebellum plays a crucial role in sensorimotor and associative learning. However, the contribution of molecular layer interneurons (MLIs) to these processes is not well understood."

4) Where possible, remove "Interestingly," when results are described. The results will suffice. The reader can decide on their own which part they find interesting.

5) line 27 - "time out" should be spelled as "timeout";

6) line 67 - Please remove "the thirsty animal learns" and replace with: "mice learn";

7) line 79 - Please replace "thirsty" with "water-deprived"

8) Sometimes the authors switch from describing the results using the past tense to the present. For example: Lines 107 to 115 are written using the past tense but then all of the sudden in line 115 the narration switches to the present tense. This is unnecessary. Please be consistent.

9) Order of Supplementary Figures: Supplementary Figure 6 appears in text before Supplementary Figure 5 (line 126 vs line 150). Please switch the figures because as I was reading I thought I missed something.

10) Add explanation to Fig. 3f legend to indicate what light and dark grey bars are. I assume light grey histograms depict the shuffled trials and dark grey original data but this should be explicit.

11) Please remove the first sentence from line 315 ("Figs. 7a,i ..."). It is redundant since you refer to it later on.

12) Please change "leading to" in line 346 to "and". "Leading to" is an interpretation that should be reserved for the discussion.

Reviewer #3:

Remarks to the Author:

The authors have significantly revised their manuscript, greatly improving the presentation of data and clarifying several important points. Most notably, they have improved their analysis of the relationship between licking and neural activity, demonstrating that there is not a fixed relationship between the two across phases of the trial (strongly suggesting that motor output per se cannot explain the main results). The authors have also addressed the majority of my other concerns.

Notably, I am still confused about what these data do tell us. Disentangling reward expectation and valence is challenging. These are not the same, but the authors' response seems to suggest that they believe otherwise (response to reviewer one states "our data does show that there is a contribution of reward expectancy (equivalent to valence)". Expectation is not binary, and should scale with the degree to which the animal anticipates reward. The experiment rewarding both odors and randomly varying reward size does not address this. Typically one would instead vary the task difficulty, changing for example the odorant concentration to manipulate expectation based on uncertainty.

However, licking is also a measure of expectation, and the authors do show a positive relationship between licking and dF/F in the anticipation period (though it is weaker than during the reinforcement period). This might suggest a role of expectancy. On the other hand, given that misses and hits are encoded the same, it seems that this could be a learned stimulus response independent of

expectation. In my view, it remains unclear how to reconcile different results such as these within the manuscript. Ideally, the revision would have had experiments targeting this question. For example, if this is truly just a learned stimulus encoding according to the binary definition of valence the authors describe, one should be able to remove the reward tube after learning and measure the same stimulus-driven responses (likely even in the anesthetized animal) regardless of expectation.

I realize that these issues are challenging to disentangle, and I do think the manuscript as currently presented contains useful data and many reasonable claims. I am on the fence about whether further effort should be made to explain the nature of these responses. If there is consensus from other reviewers that this revision addresses their points sufficiently, I am content with this version. I would also, however, support more effort to understand or clearly describe the nature of these responses (even if only with a more convincing discussion) if I am not alone in this thinking.

Reply to reviewers

We would like to thank the reviewers for constructive comments. Below is a point-by-point reply.

Reviewer #1 (Remarks to the Author):

The authors have addressed all of my previous concerns by providing extensive new analysis, supplementary information and additional data. For me the modelling aspect of the study is perhaps the weakest addition to the revised manuscript given the 'parameter selection' based on best guesses rather than recorded activity in the described behavior. That said, the model highlights a plausible mechanism for how hM4Di affects PC output during behavior, which could be tested in follow-on studies.

Ian Duguid

Thank you Ian. We do agree about the relevance of the model. However, the model makes a contribution because it shows a plausible explanation for the results of the DREADDs experiment.

Reviewer #2 (Remarks to the Author):

The revised manuscript by Ma and colleagues makes substantial improvements to the original version. The authors took time and care to address most of the points raised by all 3 reviewers, which is very much appreciated. The current version presents the results in a much more convincing way and is therefore much easier to follow. I also appreciate the change of the color scheme of the Figures and indicating the N in all experiments. However, I still have a few concerns and questions that I would like to see addressed before publication.

Major:

“1) Data in new Figure 2e now shows the individual dF/F for all ROIs, which is a good improvement and is appreciated. However, many ROIs seem to have negative $\Delta F/F$, which indicates that the way the baseline is calculated may not be accurate.” Several issues are raised by the reviewer in point 1. Because of this, we have separated the answer in different sections:

“Since $dF/F = (F(t) - F_0)/(F_0 - F_b)$, where F_0 is the baseline and F_b the background, the dF/F_0 can only be negative if F_0 is greater than the $F(t)$.” Yes, for the estimation of dF/F it is key to estimate a background and neuropil fluorescence. The CaImAn software we use does estimate a background component, but we had not explicitly indicated that this is the case. We added a sentence to the methods indicating that a separate background component was estimated by CaImAn (line 720):

“CaImAn identifies different spatial components (addressed here as regions of interest, or ROIs) and estimates a separate component representing the background and neuropil signals.”

“GCaMPs can only depict increase in the Ca²⁺ concentration.” “In general negative dF/F is hard to interpret and in fact some 2photon experts, like David Tank, suggest that negative events are false, and therefore the distribution of negative event amplitudes and durations can be used to predict the corresponding distribution of positive events.” For neurons whose basal firing rate is low, such as pyramidal cells in neocortex GCaMP only reports increases in calcium from baseline. However, the basal rate of MLIs is not zero, and earlier studies by Isabel Llano showed that addition of muscimol inhibiting the MLIs elicits a decrease in calcium¹.

In line 598 the authors write: “Baseline of intensity (F₀) was defined as the mean fluorescence intensity before trial start, defined as the time when the animal first licked.” Negative dF/F suggests that in fact the MLIs were very active during that period. Since movement can increase MLI activity the authors could define F₀ based on the same time-period but only for trials where the animal was very still. Maybe the window should be shorter since the trials were self-initiated? The authors could also just take a small percentile of the average fluorescence signal over that time window as their F₀. Of course it could reflect the true inhibition but then it’s better to calculate the z-score instead of the dF/F. This issue does not change the conclusions but the authors should address the accuracy of their F₀ estimation. We beg to differ with the reviewer. The reviewer suggests to calculate the baseline “for trials where the animal was very still”. However, there is no accurate way to know at which point in time the MLI is not firing, thus the window cannot be properly chosen using this suggestion.

We have chosen to calculate the baseline interval during a behaviorally well-defined epoch. Under these circumstances it is likely that the basal calcium does not correspond to a resting calcium level since MLIs can be active even before the licking periods. Because the baseline interval is well-defined the decreases in dF/F likely carry biological meaning. Ultimately, whether decreases in dF/F reflect decreases in the firing rate would need to be determined by parallel calcium and patch clamp recording, but these experiments are beyond the scope of this manuscript.

2) Why is there a significant, short-latency peak in the licking frequency in the DREADD experiment that can be seen in Figure 7c? It is occurring ~1s before the onset of the odorant and reaches between 5-10Hz. It is not there during the imaging experiments (Figure 5b and Supplementary Figure 5). In fact in these figures we can see a small (~2Hz) lick response which peak coincides with the onset of the odorant. The sharp, short peak in Figure 7c looks almost like an artifact. Please check this carefully and provide an explanation to why we can see such response in these experiments. We double-checked the lick records for Fig. 7c, and this is not an artifact, there is always at least one lick 1-1.5 sec before odorant onset in these experiments (as well as in all other behavioral experiments). As far as to why there is a difference between the DREADDs and GCaMP experiments: Lick rate records for the GCaMP experiments were convolved with a two second Gaussian. In contrast, lick rate records for the DREADDs experiments were not convolved because we were particularly interested in determining differences shortly after odorant addition (see inset in Fig. 7c, the differences are quantified in Fig. 7d).

In the manuscript we had not stated that the lick rate records were convolved with the two second Gaussian. This is now stated in line 632 of the methods:

“Lick rate was calculated from the lick records and the time course was convolved with a two second Gaussian for the experiments where we performed multiphoton calcium imaging. We did not convolve the lick rate records for the experiments with chemogenetics.”

Minor:

1) The choice to present new analyses and results thereof as “Supplementary Notes” is a bit odd as often the data and text of the notes are very relevant to the interpretation of the results and figures. I would strongly urge the authors to incorporate these notes into the main results section or (where appropriate) to the Methods. Leaving these details out of the main body of the manuscript decreases readability. For example: Note 4 is crucial to support the authors’ conclusion regarding the licking behavior. We agree with the reviewer. We had relegated substantial relevant text to the supplementary notes because the text for the introduction, results and discussion was close to the 5,000 word limit. Fortunately, Dr. Ranade allowed us to increase the text to 6,500 words allowing us to move all notes but one to the results.

2) The same applies to the model. I find it odd that it’s not presented under Results but in the Discussion. The model is now presented in the results.

3) Abstract could use a sentence or two of an introduction before the authors proceed to the description of the experiments so that the readers can understand what the manuscript is about. I suggest something like: “ The cerebellum plays a crucial role in sensorimotor and associative learning. However, the contribution of molecular layer interneurons (MLIs) to these processes is not well understood.” Thank you, we added the two sentences and we were allowed to increase the abstract length to 165 words.

4) Where possible, remove “Interestingly,” when results are described. The results will suffice. The reader can decide on their own which part they find interesting. We deleted or replaced “interestingly” in all instances with the exception of this sentence in the discussion: “Interestingly, odorant responses have been reported in the cerebellum.”

5) line 27 - “time out” should be spelled as “timeout”; Done

6) line 67 - Please remove “the thirsty animal learns” and replace with: ”mice learn”; Replaced

7) line 79 – Please replace “thirsty” with “water-deprived”. Replaced two instances of “thirsty”

8) Sometimes the authors switch from describing the results using the past tense to the present. For example: Lines 107 to 115 are written using the past tense but then all of the sudden in line 115 the narration switches to the present tense. This is unnecessary. Please be consistent. In the revision we use past tense.

9) Order of Supplementary Figures: Supplementary Figure 6 appears in text before Supplementary Figure 5 (line 126 vs line 150). Please switch the figures because as I was reading I thought I missed something. The supplementary figures were reordered.

10) Add explanation to Fig. 3f legend to indicate what light and dark grey bars are. I assume light grey histograms depict the shuffled trials and dark grey original data but this should be explicit. We have added an explanation to the figure legend.

11) Please remove the first sentence from line 315 (“Figs. 7a,i ...”). It is redundant since you refer to it later on. **Removed.**

12) Please change “leading to” in line 346 to “and”. “Leading to” is an interpretation that should be reserved for the discussion. **Edited**

Reviewer #3 (Remarks to the Author):

The authors have significantly revised their manuscript, greatly improving the presentation of data and clarifying several important points. Most notably, they have improved their analysis of the relationship between licking and neural activity, demonstrating that there is not a fixed relationship between the two across phases of the trial (strongly suggesting that motor output per se cannot explain the main results). The authors have also addressed the majority of my other concerns.

Notably, I am still confused about what these data do tell us. Disentangling reward expectation and valence is challenging. These are not the same, but the authors’ response seems to suggest that they believe otherwise (response to reviewer one states “our data does show that there is a contribution of reward expectancy (equivalent to valence)”). Expectation is not binary, and should scale with the degree to which the animal anticipates reward. The experiment rewarding both odors and randomly varying reward size does not address this. Typically one would instead vary the task difficulty, changing for example the odorant concentration to manipulate expectation based on uncertainty.

However, licking is also a measure of expectation, and the authors do show a positive relationship between licking and dF/F in the anticipation period (though it is weaker than during the reinforcement period). This might suggest a role of expectancy. On the other hand, given that misses and hits are encoded the same, it seems that this could be a learned stimulus response independent of expectation. In my view, it remains unclear how to reconcile different results such as these within the manuscript. Ideally, the revision would have had experiments targeting this question. For example, if this is truly just a learned stimulus encoding according to the binary definition of valence the authors describe, one should be able to remove the reward tube after learning and measure the same stimulus-driven responses (likely even in the anesthetized animal) regardless of expectation.

I realize that these issues are challenging to disentangle, and I do think the manuscript as currently presented contains useful data and many reasonable claims. I am on the fence about whether further effort should be made to explain the nature of these responses. If there is consensus from other reviewers that this revision addresses their points sufficiently, I am content with this version. I would also, however, support more effort to understand or clearly describe

the nature of these responses (even if only with a more convincing discussion) if I am not alone in this thinking.

First, we reviewed the wording of the manuscript, and we agree that the term “reward expectation” was not being used appropriately in the discussion. We changed the wording of the sentence in line 498 to:

“Thus, we postulate that the MLI response during the odorant period is related to valence that reflects the sign of reward expectation (positive or negative), consistent with the fact that GCs in lobule VI were found to respond to reward expectation^{2,3}.”

However, we would argue that the go-go data does indicate that the activity of the MLIs reflects valence as opposed to reward expectation. The reviewer states that “However, licking is also a measure of expectation, and the authors do show a positive relationship between licking and dF/F in the anticipation period (though it is weaker than during the reinforcement period).” This statement is incorrect. The correlation between these two variables (Supplementary Fig. 12d) is not statistically significant arguing for encoding of value, not reward expectation. Furthermore, the reviewer suggested to remove the spout to show that this is not reward expectation. We did not do this, but we did remove the reward (delivered zero microliters), and the dF/F did not decrease substantially (Supplementary Fig. 12c). Therefore, our data makes an important contribution suggesting that MLIs encode for valence. However, we do agree with the reviewer that future experiments are necessary to fully disentangle whether MLIs encode for value vs. reward expectation. We have modified the paragraph dealing with this issue:

“Here we provide evidence for the involvement of MLIs in conveying information on contextual identity of a stimulus in associative learning. We do not find that the MLIs respond to odorants per se. Rather, the reversal experiment (Fig. 4) and the similar $\Delta F/F$ responses and stimulus decoding for correct and incorrect behavioral response trials (Supplementary Fig. 9) indicate that MLIs respond to contextual odorant identity: “is this the rewarded odorant?”, which is directly related to valence, a binary measure of an emotion reflected by the motivation to receive reward⁴. Furthermore, our results in the go-go task where both odorants are rewarded with varying volumes of sugar water (Supplementary Fig. 12) and the lack of a correlation between $\Delta F/F$ and lick rate (Supplementary Fig. 12d) are consistent with the response reflecting valence (as opposed to value). Thus, we postulate that the MLI response during the odorant period is related to valence that reflects the sign of reward expectation (positive or negative), consistent with the fact that GCs in lobule VI were found to respond to reward expectation^{2,3}. However, future experiments are necessary to fully disentangle whether MLIs encode for value vs. reward expectation.”

References

- 1 Franconville, R., Revet, G., Astorga, G., Schwaller, B. & Llano, I. Somatic calcium level reports integrated spiking activity of cerebellar interneurons in vitro and in vivo. *J Neurophysiol* **106**, 1793-1805, doi:10.1152/jn.00133.2011 (2011).

- 2 Wagner, M. J., Kim, T. H., Savall, J., Schnitzer, M. J. & Luo, L. Cerebellar granule cells encode the expectation of reward. *Nature* **544**, 96-100, doi:10.1038/nature21726 (2017).
- 3 Giovannucci, A. *et al.* Cerebellar granule cells acquire a widespread predictive feedback signal during motor learning. *Nat Neurosci* **20**, 727-734, doi:10.1038/nn.4531 (2017).
- 4 Tye, K. M. Neural Circuit Motifs in Valence Processing. *Neuron* **100**, 436-452, doi:10.1016/j.neuron.2018.10.001 (2018).

Reviewers' Comments:

Reviewer #2:

Remarks to the Author:

The authors have addressed all of my previous concerns by providing extensive analysis, supplementary information and additional notes and data. I agree that the simultaneous electrophysiological and two-photon measurements are beyond the scope of this manuscript. As previously indicated, the negative values do not change the interpretation of the overall results. I would suggest that in the future work the dF/F analysis could benefit from estimating the baseline as 10th percentile of the F (taking the average of the entire raw fluorescence signal from a given neuron over the whole period of the recording and using the 10th percentile of it for F_0) or using a z-score instead of the dF/F . I would like to express my appreciation to the authors for taking time to address all of reviewers' concerns.

REVIEWERS' COMMENTS:

Reviewer #2 (Remarks to the Author):

The authors have addressed all of my previous concerns by providing extensive analysis, supplementary information and additional notes and data. I agree that the simultaneous electrophysiological and two-photon measurements are beyond the scope of this manuscript. As previously indicated, the negative values do not change the interpretation of the overall results. I would suggest that in the future work the dF/F analysis could benefit from estimating the baseline as 10th percentile of the F (taking the average of the entire raw fluorescence signal from a given neuron over the whole period of the recording and using the 10th percentile of it for F_0) or using a z-score instead of the dF/F . I would like to express my appreciation to the authors for taking time to address all of reviewers' concerns.

We would like to thank the reviewer for the constructive comments. We will consider the suggestions in our future studies.